# `ProbTS`: Benchmarking Point and Distributional Forecasting across Diverse Prediction Horizons

**Jiawen Zhang**[*]
HKUST (GZ)
Guangzhou, China
jiawe.zh@gmail.com

**Xumeng Wen**
Microsoft Research Asia
Beijing, China
xumengwen@microsoft.com

**Zhenwei Zhang**[*]
Tsinghua University
Beijing, China
zzw20@mails.tsinghua.edu.cn

**Shun Zheng**
Microsoft Research Asia
Beijing, China
shun.zheng@microsoft.com

**Jia Li**
HKUST (GZ)
Guangzhou, China
jialee@ust.hk

**Jiang Bian**
Microsoft Research Asia
Beijing, China
jiang.bian@microsoft.com

## Abstract

Delivering precise point and distributional forecasts across a spectrum of prediction horizons represents a significant and enduring challenge in the application of time-series forecasting within various industries. Prior research on developing deep learning models for time-series forecasting has often concentrated on isolated aspects, such as long-term point forecasting or short-term probabilistic estimations. This narrow focus may result in skewed methodological choices and hinder the adaptability of these models to uncharted scenarios. While there is a rising trend in developing universal forecasting models, a thorough understanding of their advantages and drawbacks, especially regarding essential forecasting needs like point and distributional forecasts across short and long horizons, is still lacking. In this paper, we present ProbTS, a benchmark tool designed as a unified platform to evaluate these fundamental forecasting needs and to conduct a rigorous comparative analysis of numerous cutting-edge studies from recent years. We dissect the distinctive data characteristics arising from disparate forecasting requirements and elucidate how these characteristics can skew methodological preferences in typical research trajectories, which often fail to fully accommodate essential forecasting needs. Building on this, we examine the latest models for universal time-series forecasting and discover that our analyses of methodological strengths and weaknesses are also applicable to these universal models. Finally, we outline the limitations inherent in current research and underscore several avenues for future exploration. [1]

## 1 Introduction

Time-series forecasting has extensive applications in various industries, including traffic flow forecasting [43], renewable energy forecasting [67], and diverse forecasting demands in retail [8], finance [29], physical system [39], and climate [48]. It is crucial to provide forecasts across different prediction horizons, addressing both short- and long-term planning needs [13, 26, 4, 61]. Moreover, modern decision-making processes typically require not only point forecasts to quantify planning efficiency but also robust distributional estimations to manage uncertainty effectively [24, 30]. The fundamental need to produce accurate point and distributional forecasts across various horizons presents significant challenges to existing forecasting approaches.

---

[*]This work was done during the internship at Microsoft Research Asia.
[1]Project repository: https://github.com/microsoft/ProbTS

38th Conference on Neural Information Processing Systems (NeurIPS 2024) Track on Datasets and Benchmarks.

Nevertheless, much of the previous research on developing deep learning models for time-series forecasting has often focused on isolated aspects, such as long-term point forecasting or short-term distribution estimations. This narrow focus may result in skewed methodological choices and hinder the adaptability of these models to rarely evaluated scenarios. For example, studies such as [78, 72, 38, 76, 11, 71, 49, 73, 40] have primarily explored neural architecture designs tailored for long-term point forecasting with strong trending and seasonal patterns. However, it remains unclear how these advancements can be effectively extended to capture complicated distributions and whether these designs maintain their effectiveness in short-term scenarios. Conversely, research such as [58, 57, 62, 7, 33] adapts deep generative models [17, 28] for probabilistic forecasting, specializing in characterizing complex data distributions. Yet, these models have mainly been developed and evaluated in short-term scenarios, leaving questions about their effectiveness in long-term forecasting and their ability to preserve point forecasting performance.

Despite the recent surge in building time-series foundation models over the past year [18, 56, 14, 10, 15, 25, 41, 20, 70, 2, 23, 74], our understanding of their advantages and limitations, especially regarding essential forecasting needs like point and distributional forecasts across various horizons, is still limited. Many of these models claim to support arbitrary prediction horizons, employing different mechanisms that come with their own set of advantages and drawbacks. Among them, a select few offer capabilities for distributional forecasting, which, however, are typically confined to predefined closed-form distributions [56, 70] or discrete distributions with value quantization [2]. The emergence of these foundation models has brought about unprecedented zero-shot forecasting capabilities. Consequently, it is both timely and crucial to delve into an evaluation of their strengths and weaknesses, especially in relation to the fundamental forecasting needs mentioned earlier.

In this study, we present `ProbTS`, a benchmark tool crafted to serve as a comprehensive platform for assessing those key forecasting needs and for performing a detailed comparison of several state-of-the-art models developed in recent years. To address the core forecasting requirements, `ProbTS` includes a broad array of datasets and spans various forecasting horizons. It also utilizes both point and distributional metrics to facilitate a thorough performance evaluation.

Our research reveals that the specific data characteristics inherent to different forecasting requirements often play a crucial role in guiding the selection of model designs. As a result, it is crucial to have a comprehensive view of the essential forecasting needs. To aid in the analysis and interpretation of performance, we measure three essential data characteristics in `ProbTS`: the strength of trends and seasonality, and the complexity of the data distribution. Moreover, we have explicitly distinguished three fundamental methodological aspects within `ProbTS` that differentiate the existing forecasting models, largely influencing their pros and cons. The first aspect involves the approach to distributional forecasting, ranging from models focused on point forecasts [49, 40] to those using pre-defined distribution heads based on specific data assumptions [56, 70]. The second aspect is the decoding scheme used to generate multi-step forecasts, which can be either autoregressive (AR) or non-autoregressive (NAR). The third aspect pertains to the normalization choice, where the long-term point forecasting models typically employ reversible instance normalization (RevIN) [32] while short-term probabilistic ones often use mean scaling strategies [58, 57].

By utilizing `ProbTS`, we conduct a systematic comparison between studies that focus on long-term point forecasting and those aimed at short-term distributional estimation, employing various forecasting horizons and evaluation metrics. Our overarching finding is that the strengths of these methods tend to diminish in scenarios they are rarely evaluated in, highlighting several important but unresolved research questions. Notably, while recent probabilistic forecasting approaches have shown proficiency in short-term distribution estimation, we find that long-term distributional forecasting remains a significant challenge. This challenge stems from achieving distribution estimation that remains both efficient and effective as the prediction horizon extends—a topic that has not been thoroughly investigated in existing literature. Additionally, our analysis uncovers a clear divide in the choice of decoding schemes: most long-term point forecasting methods opt for NAR, whereas choices in short-term forecasting studies are more evenly split. Further investigation suggests that the preference for NAR methods stems from existing AR models' difficulty in managing error accumulation, particularly over extended horizons with strong trends. However, we observe that a proper normalization strategy can significantly improve AR models in long-term forecasting, opening new possibilities for AR-based approaches. Moreover, AR decoding performs better in scenarios with pronounced seasonality, indicating potential for refining these strategies, particularly for long-

term forecasts. Given the inefficiency of existing NAR-based probabilistic methods like CSDI, our comparison highlights the need for further exploration of decoding strategies in future research.

Furthermore, we have expanded the analytical framework of ProbTS to include an examination of several very recently developed time-series foundation models, which has allowed us to re-validate some of our earlier findings. Interestingly, there appears to be a relatively even split in their preference for AR and NAR decoding schemes. Our analysis reaffirms the limitation of AR in handling time-series data, as we observe that AR-based foundation models tend to excel at shorter horizons. However, their performance advantages often significantly diminish over longer forecasting periods. This underscores the critical need for future research to focus on addressing the issue of error accumulation in AR-based foundation models. Besides, our exploration reveals that current probabilistic foundation models may face challenges when dealing with complex data distributions. This observation suggests that the integration of more sophisticated distribution estimation techniques could enhance the development of time-series foundation models.

In summary, we have made the following contributions.

- Introduction of ProbTS, a benchmark tool designed for a thorough evaluation of essential forecasting needs, towards precise point and probabilistic forecasting across varied horizons.

- Comprehensive analysis of methodological variations within forecasting models, especially regarding distributional estimation methods and decoding schemes (AR vs. NAR), which illuminates significant yet previously underexplored research challenges.

- Extension of our analytical framework to include the latest time-series foundation models, providing insights into the implications of their methodological choices and underscoring important directions for future research endeavors.

## 2   Related Work

**Classical Time-series Forecasting Models**   In recent years, classical research in time-series forecasting has bifurcated into two distinct but complementary streams. The first stream has concentrated on refining neural architecture designs for long-term forecasting, primarily employing non-autoregressive decoding schemes to address scenarios with pronounced trend and seasonality. This stream has evolved from enhancing multi-layer perceptrons [53, 76] to developing specialized recurrent or convolutional neural networks [34, 37], and introducing Transformer-based models [66, 49, 40]. Despite achieving advancements in point forecasts, these efforts mainly capture average future changes, with only a few adopting approaches like quantile regression to partially overcome this limitation [69, 36]. On the other hand, the second stream, probabilistic time-series forecasting specializes in capturing the intricate data distribution of future time series. It encompasses a spectrum of techniques, from utilizing predefined likelihood functions [55, 60] and Gaussian copulas [59, 19] to exploring advanced deep generative models [58, 7]. Unlike the first stream, this branch employs both AR [58, 57] and NAR decoding schemes [62, 7, 33], often utilizing standard neural network architectures to represent time series [16, 58, 57, 7, 19], though some studies propose customized designs [62, 35, 5]. Together, these streams highlight the diverse approaches to forecasting, ranging from point predictions focusing on the mean future variations to probabilistic forecasts that capture the full distribution of future values. In Appendix A.1, we summarize a comparison of these models on the coverage of essential forecasting needs and their methodological preferences.

**Universal Time-series Foundation Models**   Over the past year, the development of time-series foundation models has greatly accelerated, driven by the success of language foundation models [9]. This wave has seen models such as Lag-Llama [56], TimesFM [15], Timer [41], and Chronos [2] adopting the decoder-only Transformer architecture with an AR decoding scheme. Conversely, models like ForecastPFN [18], MOIRAI [70], TTM [20], and UniTS [23] employ the NAR decoding, often using variable-length placeholders to indicate prediction positions for different horizons. Probabilistic forecasting is less common, with MOIRAI and Lag-Llama integrating pre-defined distribution heads (Student-t for Lag-Llama and a mixture for MOIRAI) while Chronos uses quantized bins to accommodate time-series values and adopts Softmax outputs for distribution approximation. The strategic choice between AR and NAR decoding and the method for distributional estimation highlight distinct trade-offs. For an extensive comparison, see Appendix A.2.

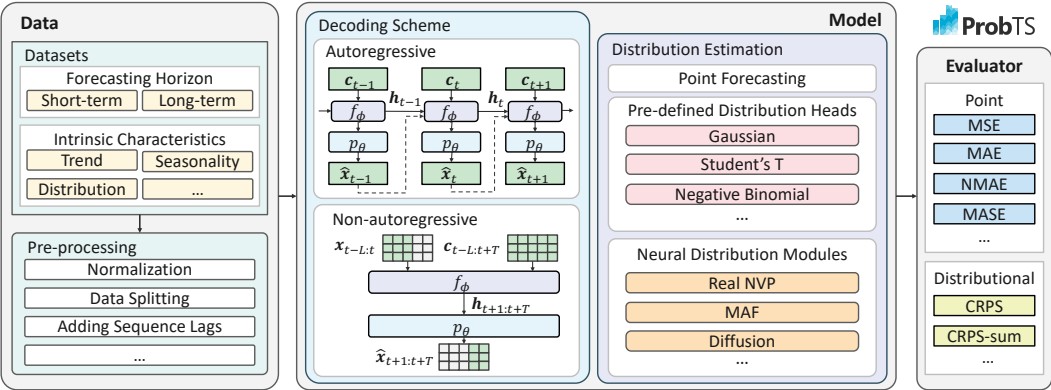

Figure 1: An overview of `ProbTS`.

**Toolkits for Time-series Forecasting.** We observe a plethora of toolkits that have been developed for time-series forecasting. These range from those primarily designed for point forecasting, such as Prophet [63], sktime [42], tsai [52], and TSlib [71], to others that incorporate probabilistic forecasting, including GluonTS [1], PyTorchTS [58], PyTorchForecasting[2], and NeuralForecast[3]. In creating `ProbTS`, we built upon the foundations laid by tools like PyTorchTS, GluonTS, and TSlib. Our unique contribution is a detailed approach that supports both precise point and probabilistic forecasting over various horizons, and examines methodological differences in forecasting models, especially regarding distributional estimation and decoding schemes (AR vs. NAR). Additionally, `ProbTS` integrates cutting-edge time-series foundation models, making it a comprehensive benchmark tool for tackling current and future challenges in time-series forecasting. A comparison of `ProbTS` with existing toolkits, focusing on functionalities and features, is provided in Appendix A.3.

## 3  The `ProbTS` Tool

This section offers a concise overview of the `ProbTS` tool's design and implementation. The core modules and the primary pipeline of `ProbTS` are depicted in Figure 1.

**Data** We aggregate publicly accessible datasets used for both short-term and long-term forecasting. Initial data visualization analyses reveal that the data domains and forecasting horizons significantly influence specific data characteristics within a given forecasting horizon. For instance, many long-term forecasting scenarios exhibit clear trend and seasonality patterns within a forecasting window, while numerous short-term forecasting cases display irregular variations within a short sliding window. Consequently, we have developed quantified indicators, such as trend and seasonality strengths, along with *non-Gaussianity* to indicate the complexity of data distribution within a forecasting window. Detailed information about dataset statistics, visualization analyses, and quantified measures can be found in Appendix B.1.1, B.1.2, B.1.3, and B.1.4. The quantified measurements for all forecasting scenarios are compiled in Table 1.

**Metrics** `ProbTS` incorporates a broad range of evaluation metrics to enable a thorough assessment of both point and distributional forecasts. These metrics are elaborated in detail in Appendix B.2. In this paper, we primarily use the normalized mean absolute error (NMAE) for point forecasts and the continuous ranked probability score (CRPS) for distributional forecasts to succinctly communicate the critical insights discovered. It is noteworthy that some methods reproduced in `ProbTS`, their original papers reported certain point forecast metrics before de-normalizing forecasts to the initial scale [75, 71, 49] or primarily reveal aggregated distributional metrics over all time-series variates, namely CRPS-sum [59, 57, 58]. We have verified our reproduced results align with their reported results and utilized the unified metrics in this study to offer a comprehensive and fair comparison of these studies from different research threads.

---

[2]github.com/jdb78/pytorch-forecasting
[3]github.com/Nixtla/neuralforecast

Table 1: This table includes a quantitative assessment of the inherent characteristics for all forecasting scenarios, each corresponding to a dataset with a specific forecasting horizon. We use the suffixes "-S" and "-L" to differentiate between short-term and long-term scenarios. Quantified indicators encompass trend and seasonality strengths, as well as non-Gaussianity, where a higher value signifies a greater deviation from a Gaussian distribution.

| Dataset-Horizon | Exchange-S | Solar-S | Electricity-S | Traffic-S | Wikipedia-S | ETTm1-L | ETTm2-L |
|---|---|---|---|---|---|---|---|
| Trend $F_T$ | 0.9982 | 0.1688 | 0.6443 | 0.2880 | 0.5253 | 0.9462 | 0.9770 |
| Seasonality $F_S$ | 0.1256 | 0.8592 | 0.8323 | 0.6656 | 0.2234 | 0.0105 | 0.0612 |
| Non-Gaussianity | 0.2967 | 0.5004 | 0.3579 | 0.2991 | 0.2751 | 0.0833 | 0.1701 |

| Dataset-Horizon | ETTh1-L | ETTh2-L | Electricity-L | Traffic-L | Weather-L | Exchange-L | ILI-L |
|---|---|---|---|---|---|---|---|
| Trend $F_T$ | 0.7728 | 0.9412 | 0.6476 | 0.1632 | 0.9612 | 0.9978 | 0.5438 |
| Seasonality $F_S$ | 0.4772 | 0.3608 | 0.8344 | 0.6798 | 0.2657 | 0.1349 | 0.6075 |
| Non-Gaussianity | 0.0719 | 0.1422 | 0.1533 | 0.1378 | 0.1727 | 0.1082 | 0.1112 |

**Model**   The model module in `ProbTS` explicitly differentiates critical methodological decisions, especially the decoding scheme (AR vs NAR) and the distributional estimation approach. Specifically, we employ the following mathematical formulation. We denote an element of a multivariate time series as $x_t^k \in \mathbb{R}$, where $k$ represents the variate index and $t$ denotes the time index. At time step $t$, we have a multivariate vector $\boldsymbol{x}_t \in \mathbb{R}^K$. Each $x_t^k$ is associated with covariates $\boldsymbol{c}_t^k \in \mathbb{R}^N$, which encapsulates auxiliary information about the observations. Given a length-$T$ forecast horizon, a length-$L$ observation history $\boldsymbol{x}_{t-L:t}$ and corresponding covariates $\boldsymbol{c}_{t-L:t}$, the objective in time series forecasting is to generate the vector of future values $\boldsymbol{x}_{t+1:t+T}$. Based on established conventions, we categorize forecast as short-term if the horizon $T \leq \mathcal{T}$ [57, 62], and long-term if $T \gg \mathcal{T}$ [75, 49, 40], where $\mathcal{T}$ represents the primary periodicity of the data (e.g., 24 for hourly frequency). To represent point and distributional forecasting in a unified way, here we divide a model into an encoder $f_\phi$ and a forecaster $p_\theta$. An encoder is tasked with generating expressive hidden states $\boldsymbol{h} \in \mathbb{R}^D$. Under *autoregressive* decoding scheme, encoder forecasts variates using their past values: $\boldsymbol{h}_t = f_\phi(\boldsymbol{x}_{t-1}, \boldsymbol{c}_t, \boldsymbol{h}_{t-1})$. Under the *non-autoregressive* scheme, the encoder generates all the forecasts in one step: $\boldsymbol{h}_{t+1:t+T} = f_\phi(\boldsymbol{x}_{t-L:t}, \boldsymbol{c}_{t-L:t+T})$. A forecaster $p_\theta$ is employed either to directly estimate *point forecasts* as $\hat{\boldsymbol{x}}_t = p_\theta(\boldsymbol{h}_t)$, or to perform sampling based on the estimated *probabilistic distributions* as $\hat{\boldsymbol{x}}_t \sim p_\theta(\boldsymbol{x}_t|\boldsymbol{h}_t)$. In addition, the normalization choices utilized by different research branches vary, with a detailed analysis provided in Appendix D.1.

## 4   Results and Analyses

Utilizing `ProbTS`, we conducted a comprehensive benchmarking and analysis of a diverse range of state-of-the-art models from different strands of research. We mainly assessed these models using NAME and CRPS metrics across multiple forecasting horizons, repeating each experiment five times with different seeds to ensure result reliability.

**Selected Models for Comparison.**   Our selection criteria for models focused on a balance of performance, reproducibility, and simplicity. For long-term point forecasting, we included models like iTransformer [40], PatchTST [49], TimesNet [71], N-HiTS [11], and LTSF-Linear [75]. Probabilistic forecasting methods selected include GRU NVP, GRU MAF, Trans MAF [58], TimeGrad [57], and CSDI [62]. Additionally, general architectures like Linear, GRU [12], and Transformer [66], along with simple non-parametric baselines, were evaluated as a reference. For foundation models, reproducible methods such as Lag-Llama [56], TimesFM [15], Timer [41], MOIRAI [70], Chronos [2], and UniTS [23] were included. Detailed implementation specifics are in Appendix B.3.

Due to space constraints, comprehensive comparison results are placed in Appendix C, with detailed results for short-term and long-term forecasting in Tables 9 and 10, respectively. Zero-shot evaluations of pre-trained time-series foundation models are detailed in Tables 11 and 12. Our evaluation highlights the critical relationship between forecasting requirements, data properties, and modeling strategies. It aims to shed light on the strengths and limitations of current approaches, paving the way for uncovering novel research avenues.

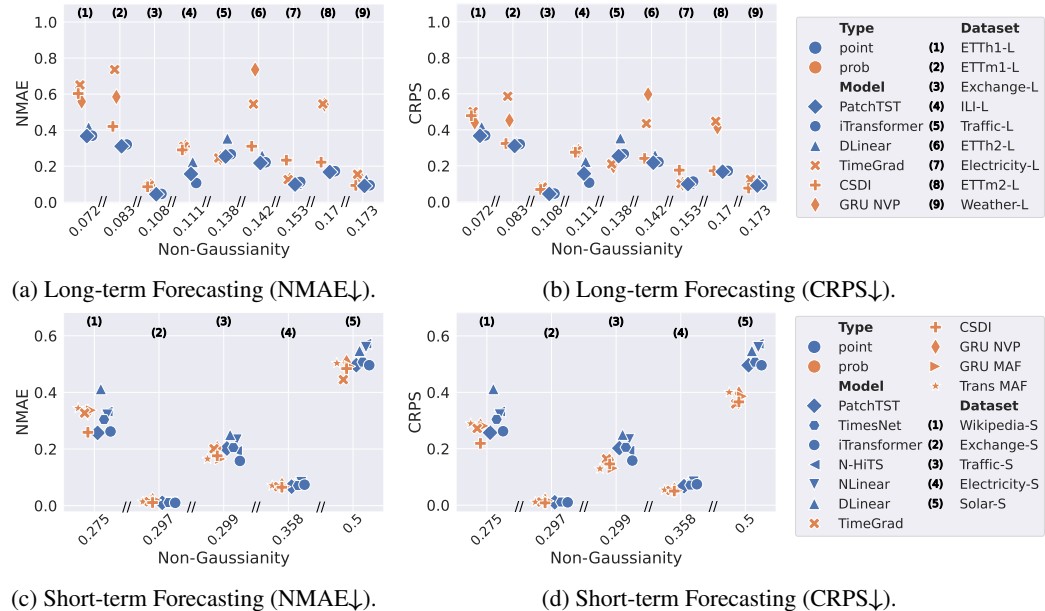

Figure 2: We present a comprehensive comparison between classical models designed for long-term point forecasting and short-term distributional forecasting across various prediction horizons. It utilizes a non-Gaussianity score to highlight the complexity of the data distribution across different datasets. The aggregated performance metrics are derived from Tables 9 and 10.

## 4.1 Analyzing Classical Models for Time-series Forecasting

We examine traditional non-universal time-series models from distinct research branches: one branch focuses on developing customized neural architectures tailored for long-term point forecasting, while the other branch concentrates on creating advanced probabilistic methods for short-term distributional forecasting. Our investigation confirms the effectiveness of these models for their intended purposes. However, we observe a notable trend: the strengths of these methods tend to diminish in scenarios where they are seldom tested.

**Diminishing Advantages of Customized Architectures in Short-term Forecasting Scenarios**
The comparative analysis presented in Figures 2a and 2c showcases the performance of point and probabilistic forecasting methods with respect to the NMAE metric. These figures also illustrate how NMAE values correlate with non-Gaussianity, a measure we employ to evaluate the complexity of data distributions. It becomes evident that customized architectures, originally crafted for long-term forecasting, tend to lose their competitive performance in short-term scenarios. This phenomenon could be attributed to the increased importance of accurately characterizing complex data distributions within shorter forecasting windows, where higher non-Gaussianity scores are indicative of this necessity. A closer look at Figures 2c and 2d further reveals that the performance disparity measured with CRPS becomes even more pronounced for datasets characterized by significant non-Gaussianity, such as Solar-S. This observation underscores the critical need for incorporating short-term patterns and distributional estimation capabilities into the design of new forecasting architectures.

**Significant Performance Degradation for Existing Probabilistic Methods in Long-term Distributional Forecasting**   The performance of current probabilistic forecasting models in long-term scenarios, even when assessed using distributional metrics such as CRPS, reveals notable limitations. This is highlighted by the comparison between Figures 2b and 2d, which shows a significant drop in performance for models like TimeGrad on ETTm1-L, GRU NVP on ETTh2-L, and Weather-L datasets. The decline in performance can be attributed to the fact that these probabilistic models were not specifically designed with the unique challenges of long-term forecasting in mind. This oversight has mixed influences. On the positive side, the design of these methods has led to a more balanced approach in the choice between AR and NAR decoding schemes, providing a versatile

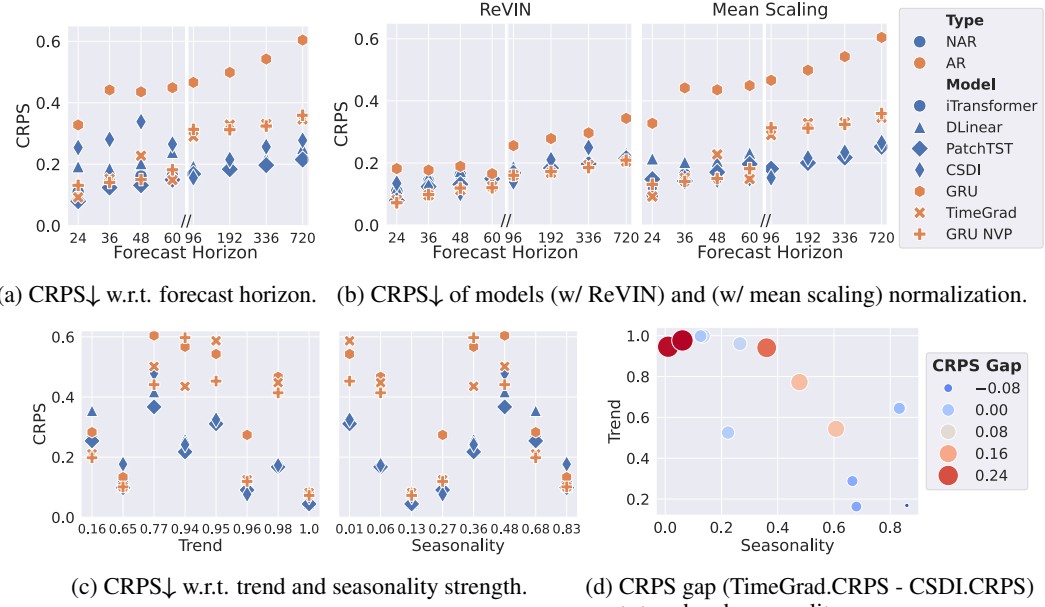

(a) CRPS↓ w.r.t. forecast horizon.

(b) CRPS↓ of models (w/ ReVIN) and (w/ mean scaling) normalization.

(c) CRPS↓ w.r.t. trend and seasonality strength.

(d) CRPS gap (TimeGrad.CRPS - CSDI.CRPS) w.r.t. trend and seasonality.

Figure 3: We explore the challenges faced by current models in conducting long-term distributional forecasting, with insights drawn from Table 10 and Table 16. Subplot (a) shows significant error increases in AR-based models, averaged across all datasets except Traffic. Subplot (b) demonstrates how the instance-level normalization impacts performance in long-term forecasting. Subplots (c) examine how trends and seasonality impact performance across all long-term forecasting datasets and horizons.

Subplot (d) further investigates the combined effects of trend and seasonality, using lighter and smaller circles to indicate situations where AR-based models are favored over NAR-based ones.

foundation for probabilistic forecasting. However, the downside is more significant: no matter using which decoding schemes, existing probabilistic models face considerable challenges when applied to long-term distributional forecasting. We will dive deeper into the specific challenges associated with each decoding scheme next. Here the performance gap underscores the need for future research to systematically investigate long-term distributional forecasting.

**Different Decoding Schemes & Challenges in Long-term Distributional Forecasting** Existing probabilistic forecasting methods exhibit a balanced preference for both AR and NAR decoding schemes. For instance, TimeGrad employs an AR decoding scheme, whereas CSDI utilizes an NAR approach. This contrasts starkly with the aforementioned customized architectures, which solely opt for NAR decoding. These two types of decoding schemes, however, confront distinctive challenges when applied to long-term probabilistic forecasting. With original normalization strategy, AR probabilistic models like TimeGrad struggle with error accumulation, particularly as the forecast horizon extends or trends strengthen, the performance gap widens, as shown in Figures 3a and 3c. On the other hand, NAR models such as CSDI encounter memory constraints in long-term forecasts (detailed in Appendix D.7). Moreover, Table 10 reveals that even on smaller datasets, such as ETTm2 and ETTh1, CSDI's performance in long-term scenarios is less than optimal, indicating reduced learning efficiency by the extension of the forecasting horizon.

**The Unexpected Superiority of AR Decoding in Addressing Strong Seasonality** Despite its drawbacks, AR-based models, as used in TimeGrad, excels in capturing strong seasonality, outperforming models like PatchTST in scenarios such as the Traffic dataset (Table 10). This advantage is further analyzed in Figures 3c and 3d, showing AR's increasing benefit with stronger seasonal patterns. This suggests AR's potential in long-term forecasting could be revitalized with solutions to its error accumulation challenge in long horizons.

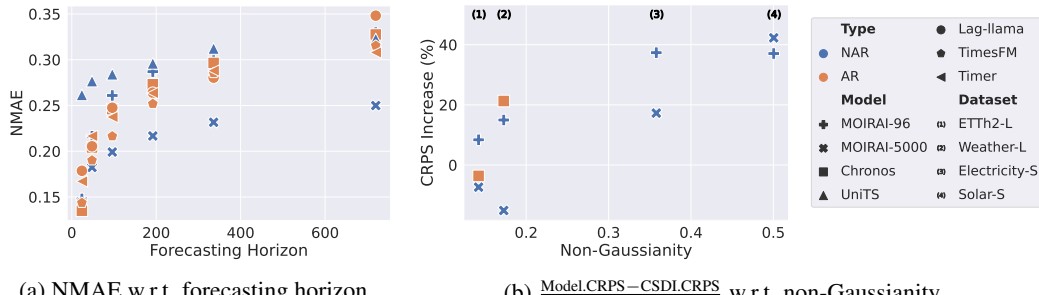

(a) NMAE w.r.t. forecasting horizon.  (b) $\frac{\text{Model.CRPS} - \text{CSDI.CRPS}}{\text{CSDI.CRPS}}$ w.r.t. non-Gaussianity.

Figure 4: We evaluate the efficacy of time-series foundation models for various forecasting horizons and distributional estimation. Subplot (a), derived from Table 11 and excluding results from the Electricity dataset, demonstrates the short-term forecasting capabilities and long-term error accumulation of AR-based models. Subplot (b), draw from Table 12, investigates short-term distributional estimation, highlighting the performance challenges of foundation models compared to CSDI in handling complex data distributions. Note that we include MOIRAI with two different context lengths, 96 and 5000, as context length significantly affects its transfer performance.

**ReVIN's Effectiveness in Long-term Forecasting with Exceptions.** RevIN significantly enhances AR-based models in long-term forecasting, as shown in Figures 3b and 7. Notably, on the ETTh1 dataset, GRU NVP (w/ RevIN) even outperforms PatchTST (w/ RevIN). While RevIN offers substantial improvements for most models across most datasets, it brings negative impact on the Traffic dataset. The Traffic dataset features strong seasonality but minimal trends, thus we speculate that the major distribution shift addressed by RevIN is related to normalizing the effect of trending. These findings indicate that normalizing the trending effect could be a direction to alleviate error accumulation of AR-based models in long-term forecasting. However, we also observe that RevIN does not seem to be an ideal match for the NAR probabilistic model. For instance, CSDI (w/ RevIN) performs worse than CSDI (w/ Scaling) on the Weather dataset. Further research in developing effective normalization strategies for NAR probabilistic models is necessary.

**No Dominating Normalization Strategies in Short-term Forecasting.** While RevIN is effective for long-term scenarios, it does not adequately address the challenges faced by short-term probabilistic models. As shown in Table 14, RevIN fails to consistently deliver significant improvements for models such as CSDI, TimeGrad, and GRU NVP in short-term forecasting. The mean scaling strategy has proven to be the most reliable option for these models, explaining its widespread use. Although omitting instance-level normalization is occasionally acceptable, it can lead to significant issues, as seen with TimeGrad (without normalization) on the Wikipedia and Solar datasets, and GRU NVP (without normalization) on Electricity. We provide detailed analysis in Appendix D.1.

## 4.2 Analyzing Foundation Models for Universal Time-series Forecasting

We next explore the capabilities of recent foundation models in universal time-series forecasting, focusing on their performance across different prediction horizons and their ability to estimate distributions, especially regarding their zero-shot transfer capabilities on unseen datasets. Table 11 showcases their significant progress, sometimes outperforming traditional models without re-training. Using the analytic framework of ProbTS, we delve into their methodological pros and cons.

**Navigating the AR Decoding Challenge over Extended Forecasting Horizons** Figure 4a illustrates how the performance of various time-series foundation models evolves in relation to expanding forecasting horizons. It is evident that for shorter horizons, AR-based foundation models such as TimesFM and Timer exhibit highly competitive performance, on par with NAR-based models like MOIRAI. However, the advantage of NAR-based decoding becomes increasingly apparent as the forecasting horizon lengthens, as demonstrated by the widening performance gap between TimesFM and MOIRAI. This trend is consistent with our earlier observation that without proper normalization strategies, AR-based methods could suffer from significant error accumulation when applied to long-term time-series forecasting. Given the inherent strengths of AR decoding, such as its superiority at capturing strong seasonality and its robust performance in certain short-term forecasting

scenarios, it is clear that further research is warranted to explore ways to overcome its limitations in long-term forecasting contexts. This could potentially unlock new avenues for enhancing the versatility and effectiveness of AR-based time-series foundation models across a broader range of forecasting horizons.

**The Critical Role of Addressing Complex Data Distributions**    Figure 4a illustrates the incremental changes in CRPS among leading probabilistic time-series foundation models, such as MOIRAI and Chronos, compared to the best-performing short-term probabilistic model, CSDI. It becomes apparent that in scenarios characterized by complex data distributions, indicated by higher non-Gaussianity scores, the performance decline of MOIRAI in relation to CSDI becomes notably more pronounced. This phenomenon may be attributed to MOIRAI's approach to supporting distributional forecasting, which involves utilizing a mixture of predefined distribution heads. While this method is efficient and effective for certain applications, it may lack the expressiveness required to accurately model more complex data distributions. Furthermore, these observations underscore that, in specific contexts, foundation models might not be able to fully replace traditional models that have been specifically tailored and trained for particular domains. Additionally, the prospect of fine-tuning these foundation models as a remedy is less economically viable, primarily due to their significantly larger size. This highlights the importance of not only continuing to refine foundation models to enhance their adaptability and performance across a wide spectrum of data distributions but also recognizing the continued relevance of domain-specific models, especially for handling intricate data distributions where a more nuanced approach may be necessary.

## 5    Conclusion

In this study, we introduced `ProbTS`, a benchmark tool tailored for evaluating essential forecasting needs, which facilitates a detailed comparison of various state-of-the-art models in the context of time-series forecasting. Through our comprehensive analysis, we identified significant challenges and opportunities in the realm of time-series forecasting, particularly highlighting the need for models that can effectively address both point and probabilistic forecasting across diverse horizons.

**Limitations**    While our study represents a significant step forward in understanding and evaluating time-series forecasting models, it does come with many limitations. A predominant focus of our work is on empirical analysis, relying heavily on intuitions and experimental observations, which may lack the depth that theoretical foundations could provide. Additionally, our exploration, though extensive, might not encompass all the nuanced factors that influence model performance. By concentrating on major methodological decisions such as AR versus NAR decoding schemes, we may inadvertently overlook other critical aspects that could play a decisive role in forecasting accuracy. Moreover, the datasets employed for evaluation, despite their diversity and relevance to current research threads, may not fully capture the vast spectrum of real-world forecasting challenges. This limitation is particularly pronounced when comparing different foundation models, as their pre-training often involves an even broader array of data, potentially skewing the comparative analysis.

**Future Directions**    The insights derived from our study open the door to numerous promising research directions. Addressing the shortcomings of AR and NAR decoding schemes, especially in their application across varying forecasting horizons, emerges as a critical area for future exploration. Innovating effective architecture designs that can navigate the intricacies of short-term forecasting challenges and devising efficient methods for long-term probabilistic forecasting stand out as urgent needs. For AR-based models, reducing error accumulation remains essential, with ReVIN-style normalization showing potential for improving long-term forecasting. Additionally, exploring effective normalization strategies for NAR-based probabilistic models is an underdeveloped yet promising area. Equally important is the enhancement of models' abilities to characterize complex data distributions, which could significantly improve the adaptability and effectiveness of foundation models. Beyond these technical endeavors, expanding the scope of datasets used for evaluation to encompass a wider range of real-world scenarios will be crucial for validating the robustness and versatility of future forecasting models. Lastly, integrating theoretical insights with empirical findings could provide a more holistic understanding of model behaviors, contributing to the development of more sophisticated and nuanced forecasting solutions.

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

# A  Additional Related Work

## A.1  A Comparison with Traditional Models on Covering Essential Forecasting Needs and methodological decisions

Table 2 presents a comparative summary of our approach, which adopts an integrated perspective, and representative studies from the existing literature.

Table 2: We provide a concise comparison between the methodologies presented in this paper and those from two distinct research branches. The comparison is based on data scenarios (short-term versus long-term forecasting), primary evaluation metrics (point versus distributional forecasts), and key methodological choices (general or customized neural architecture designs, and autoregressive or non-autoregressive decoding schemes).

| Method | Pred. Horizon | | Paradigm | | Arch. Design | | Dec. Scheme | |
|---|---|---|---|---|---|---|---|---|
| | Short | Long | Point | Distr. | General | Customized | AR | Non-AR |
| N-BEATS [53] | ✗ | ✓ | ✓ | ✗ | ✗ | ✓ | ✗ | ✓ |
| Autoformer [72] | ✗ | ✓ | ✓ | ✗ | ✗ | ✓ | ✗ | ✓ |
| Informer [78] | ✗ | ✓ | ✓ | ✗ | ✗ | ✓ | ✗ | ✓ |
| Pyraformer [38] | ✗ | ✓ | ✓ | ✗ | ✗ | ✓ | ✗ | ✓ |
| N-HiTS [11] | ✗ | ✓ | ✓ | ✗ | ✗ | ✓ | ✗ | ✓ |
| LTSF-Linear [75] | ✗ | ✓ | ✓ | ✗ | ✗ | ✓ | ✗ | ✓ |
| PatchTST [49] | ✗ | ✓ | ✓ | ✗ | ✗ | ✓ | ✗ | ✓ |
| TimesNet [71] | ✓ | ✓ | ✓ | ✗ | ✗ | ✓ | ✗ | ✓ |
| iTransformer [40] | ✗ | ✓ | ✓ | ✗ | ✗ | ✓ | ✗ | ✓ |
| DeepAR [60] | ✓ | ✗ | ✗ | ✓ | ✓ | ✗ | ✓ | ✗ |
| GP-copula [59] | ✓ | ✗ | ✗ | ✓ | ✓ | ✗ | ✓ | ✗ |
| LSTM NVP [58] | ✓ | ✗ | ✗ | ✓ | ✓ | ✗ | ✓ | ✗ |
| LSTM MAF [58] | ✓ | ✗ | ✗ | ✓ | ✓ | ✗ | ✓ | ✗ |
| Trans MAF [58] | ✓ | ✗ | ✗ | ✓ | ✓ | ✗ | ✓ | ✗ |
| TimeGrad [57] | ✓ | ✗ | ✗ | ✓ | ✓ | ✗ | ✓ | ✗ |
| CSDI [62] | ✓ | ✗ | ✗ | ✓ | ✗ | ✓ | ✗ | ✓ |
| SPD [7] | ✓ | ✗ | ✗ | ✓ | ✓ | ✗ | ✗ | ✓ |
| TSDiff [33] | ✓ | ✗ | ✗ | ✓ | ✗ | ✓ | ✗ | ✓ |
| This Study | ✓ | ✓ | ✓ | ✓ | ✓ | ✓ | ✓ | ✓ |

## A.2  A Comparison of Pre-trained Time-series Foundation Models

We have incorporated eight recently emerged time series foundation models, namely Lag-Llama [56], Chronos [2], TimesFM [15], Timer [41], MOIRAI [70], UniTS [23], ForecastPFN [18], and TTM [20], into our framework. These foundation models are categorized based on their capabilities, such as zero-shot forecasting, adaptability to varying prediction lengths, and support for probabilistic predictions, as well as their architectural designs, including whether they are auto-regressive and the nature of their backbone networks. Additionally, we have detailed their training processes, including the lengths of prediction horizons used during pre-training and the sizes of look-back windows. These details are summarized in Table 3.

Furthermore, we have compiled a summary of these foundation models' pre-training and evaluation on several classical time series forecasting datasets. This compilation is presented in Table 4.

## A.3  A Comparison with Existing Libraries on the Coverage of Data, Model, and Metrics

ProbTS is a research toolkit designed to advance forecasting research across varied horizons, focusing on both point and distributional forecasting. To achieve these objectives, ProbTS includes state-of-the-art models, comprehensive evaluation protocols (point vs. distributional), and explores different methodological aspects of forecasting models, particularly in terms of distributional estimation

Table 3: Foundation Models for Time Series. **Zero-shot** indicates whether the original work tests zero-shot capabilities. **Any-horizon** indicates if the same pre-trained model can adapt to prediction tasks of varying lengths. **AR** denotes if the model performs auto-regressive forecasting. **Prob.** indicates if the model natively supports probabilistic forecasting. **Arch.** denotes the model's backbone architecture: D-O for decoder-only transformer, E-O for encoder-only transformer, E-D for encoder-decoder transformer, and unique for specially designed backbones. **Multi-variate** indicates if the model explicitly handles multivariate relationships. **Pre-train Horizons** specifies the forecasting task horizons during pre-training. **Look-back Window** specifies the context history length settings used in the original experiments.

| Model | Zero-shot | Any-horizon | AR | Prob. | Arch. | Multi-variate | Pre-train Horizons | Look-back Window |
|---|---|---|---|---|---|---|---|---|
| Lag-Llama | ✓ | ✓ | ✓ | ✓ | D-O | ✗ | 24~60 | 32~1024 |
| Chronos | ✓ | ✓ | ✓ | ✓ | E-D | ✗ | 64 | 512 |
| TimesFM | ✓ | ✓ | ✓ | ✗ | D-O | ✗ | − | 512 |
| Timer | ✓ | ✓ | ✓ | ✗ | D-O | ✗ | up to 1440 | 672 |
| MOIRAI | ✓ | ✓ | ✗ | ✓ | E-O | ○ | varying | 100~5000 |
| UniTS | ✓ | ✓ | ✗ | ✗ | E-O | ✗ | − | 60~720 |
| ForecastPFN | ✓ | ✓ | ✗ | ✗ | E-O | ✗ | 0 50 | 50 500 |
| TTM | ✓ | ✗ | ✗ | ✗ | Unique | ✓ | 96 | 512 |

Table 4: Evaluation Datasets for Time-series Foundation Models. We selected several popular datasets to evaluate time-series foundation models. ✓ indicates pre-training on the dataset, ○ indicates zero-shot evaluation on the dataset, few indicates few-shot evaluation on the dataset, and ✗ indicates the dataset is not mentioned in the paper or documentation. '*' indicates that the data comes from the same source but may be processed differently.

| Model | Solar | Wikipedia | ETTm1 | ETTm2 | ETTh1 | ETTh2 | Electricity | Traffic | Weather | Exchange | ILI |
|---|---|---|---|---|---|---|---|---|---|---|---|
| MOIRAI | ○ | ✓ | ○ | ○ | ○ | ○ | ○ | ✓ | ○ | ✗ | ✗ |
| Lag-Llama | ✓* | ✗ | ✓ | ○ | ✓ | ✓ | ✓ | ✓ | ○ | ○ | ✗ |
| TimesFM | ○* | ✓* | ○ | ○ | ○ | ○ | ✓ | ✓ | ✓ | ✗ | ✗ |
| Chronos | ✓ | ✓* | ○ | ○ | ○ | ○ | ✓ | ○ | ○ | ○ | ✗ |
| TTM | ✗ | ✗ | ○ | ○ | ○ | ○ | ○ | ○ | ○ | ✗ | ✗ |
| UniTS | ○ | ✗ | few | ✗ | ✓ | few | ✓ | ✓ | ✓ | ✓ | ✓ |
| Timer | ✗ | ✗ | few | few | few | few | few | few | few | ✗ | ✗ |

methods and decoding schemes (AR vs. NAR). In Table 5, we provide a comprehensive comparison of ProbTS with existing libraries in terms of toolkit functionalities and the benchmarking aspects we aim to investigate.

# B More Details on `ProbTS`

## B.1 Data

The data module unifies varied data scenarios to facilitate thorough evaluation and implements standardized pre-processing techniques to ensure fair comparison.

Moreover, we utilize a quantitative approach to visually delineate datasets' intrinsic characteristics, which employs decomposition to assess trends and seasonality in a time series and evaluate the similarity between data distribution and a Gaussian to depict the complexity of data distribution.

### B.1.1 Time-series Forecasting Datasets

Table 6 provides a summary of the public datasets employed in our study. These datasets have been sourced from recent research studies in the field of deep time-series forecasting.

Table 5: Comparison of Various Time Series Tools.

| Tools | Features | | Benchmarking | | | |
|---|---|---|---|---|---|---|
| | SOTA Model | Dist. Evaluation | Short-term | Long-term | AR vs. NAR | Point vs Prob. |
| Merlion [6] | ✗ | ✗ | ✓ | ✗ | ✗ | ✗ |
| Kats [31] | ✗ | ✗ | ✗ | ✗ | ✗ | ✗ |
| pytorch-transformer-ts [65] | ✗ | ✗ | ✗ | ✗ | ✗ | ✗ |
| Prophet [63] | ✗ | ✗ | ✓ | ✗ | ✗ | ✗ |
| Darts [27] | ✗ | ✓ | ✗ | ✗ | ✗ | ✗ |
| sktime [42] | ✗ | ✓ | ✗ | ✗ | ✗ | ✗ |
| pytorch-forecasting [64] | ✗ | ✓ | ✗ | ✗ | ✗ | ✗ |
| NeuralForecast [51] | ✓ | ✓ | ✗ | ✗ | ✗ | ✗ |
| tsai [52] | ✓ | ✗ | ✗ | ✗ | ✗ | ✗ |
| TFB [54] | ✓ | ✗ | ✗ | ✓ | ✗ | ✗ |
| TSlib [71] | ✓ | ✗ | ✓ | ✓ | ✓ | ✗ |
| GluonTS [1] | ✓ | ✓ | ✓ | ✗ | ✗ | ✓ |
| ProbTS | ✓ | ✓ | ✓ | ✓ | ✓ | ✓ |

Table 6: Dataset Summary.

| Horizon | Dataset | #var. | range | freq. | timesteps | Description |
|---|---|---|---|---|---|---|
| **Long-term** | ETTh1/h2 | 7 | $\mathbb{R}^+$ | H | 17,420 | Electricity transformer temperature per hour |
| | ETTm1/m2 | 7 | $\mathbb{R}^+$ | 15min | 69,680 | Electricity transformer temperature every 15 min |
| | Electricity | 321 | $\mathbb{R}^+$ | H | 26,304 | Electricity consumption (Kwh) |
| | Traffic | 862 | (0,1) | H | 17,544 | Road occupancy rates |
| | Exchange | 8 | $\mathbb{R}^+$ | Busi. Day | 7,588 | Daily exchange rates of 8 countries |
| | ILI | 7 | (0,1) | W | 966 | Ratio of patients seen with influenza-like illness |
| | Weather | 21 | $\mathbb{R}^+$ | 10min | 52,696 | Local climatological data |
| **Short-term** | Exchange | 8 | $\mathbb{R}^+$ | Busi. Day | 6,071 | Daily exchange rates of 8 countries |
| | Solar | 137 | $\mathbb{R}^+$ | H | 7,009 | Solar power production records |
| | Electricity | 370 | $\mathbb{R}^+$ | H | 5,833 | Electricity consumption |
| | Traffic | 963 | (0,1) | H | 4,001 | Road occupancy rates |
| | Wikipedia | 2,000 | $\mathbb{N}$ | D | 792 | Page views of 2000 Wikipedia pages |

### B.1.2 Data Visualization

To provide a more tangible understanding of the different forecasting scenarios, we visualize time-series segments from both short-term and long-term forecasting datasets. The segments' window size is determined by the specific forecasting setup.

In Figure 5, we present samples extracted from short-term forecasting scenarios. At this scale, the series primarily exhibit local variations, and the compact window size often obscures pronounced seasonal or trending patterns. However, these short-term scenarios may reveal irregularly varied patterns, suggesting a more complex underlying data distribution.

On the contrary, Figure 6 illustrates long-term forecasting scenarios. With extended forecasting horizons, as showcased in datasets like Traffic, Electricity, and ETT, the series display more pronounced seasonality and trends. These characteristics render the series more regular patterns in the long-term scenarios.

It's important to note that these visualizations are not selectively chosen or "cherry-picked". We have depicted multiple time-series segments from various time steps, and the observed patterns remain consistent across these different instances.

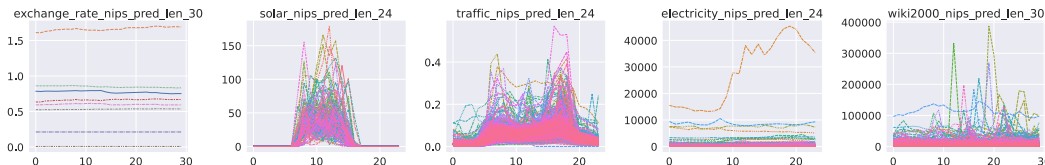

Figure 5: We have sampled and visualized multiple time-series segments from the short-term forecasting datasets. The size of the segment window is set equal to the prediction horizon.

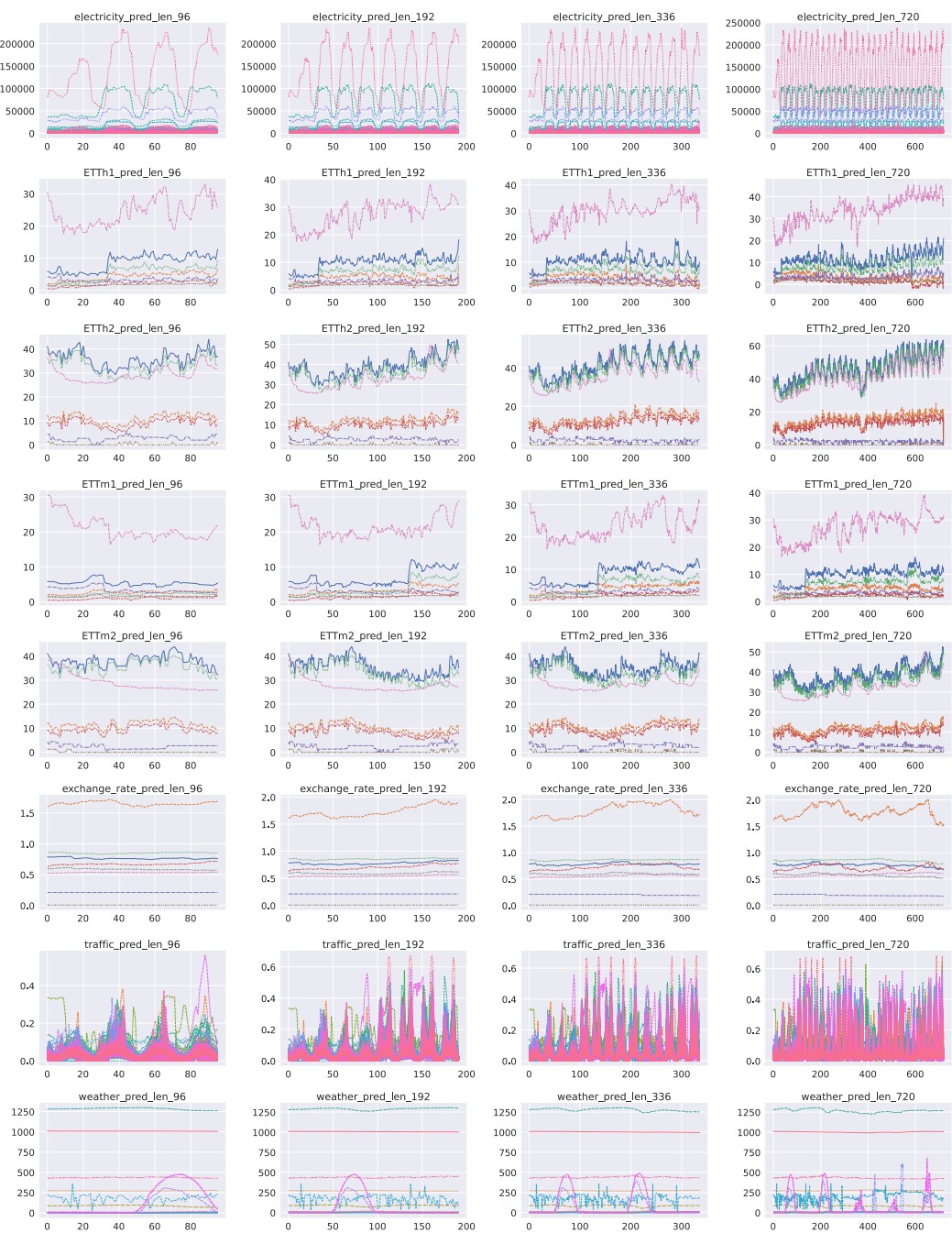

Figure 6: We have also sampled and visualized multiple time-series segments from the long-term forecasting datasets, where the size of the segment window matches the prediction horizon.

### B.1.3 Quantifying Trend and Seasonality Strengths

Based on the intuition obtained from data visualization, we would like to quantify the strengths of trend and seasonality for a time-series segment with a predefined window size (corresponding to the prediction horizon). Then we can quantify the trend and seasonality strengths at the dataset level by averaging over all time-series segments of a dataset.

To quantify the strengths of trend and seasonality for a fixed-length time-series segment, we draw upon methodologies outlined in the work of [68]. In particular, we employed a time series decomposition

model expressed as:

$$y_t = T_t + S_t + R_t,$$

where $T_t$ represents the smoothed trend component, $S_t$ signifies the seasonal component, and $R_t$ denotes the remainder component. In order to obtain each component, we followed the STL decomposition approach [4].

In the case of strongly trended data, the variation within the seasonally adjusted data should considerably exceed that of the remainder component. Consequently, the ratio $\text{Var}(R_t)/\text{Var}(T_t + R_t)$ is expected to be relatively small. As such, the measure of trend strength can be formulated as:

$$F_T = \max\left(0, 1 - \frac{\text{Var}(R_t)}{\text{Var}(T_t + R_t)}\right).$$

The quantified trend strength, ranging from 0 to 1, characterizes the degree of trend presence. Similarly, the evaluation of seasonal intensity employs the detrended data:

$$F_S = \max\left(0, 1 - \frac{\text{Var}(R_t)}{\text{Var}(S_t + R_t)}\right).$$

A series with $F_S$ near 0 indicates minimal seasonality, while strong seasonality is indicated by $F_S$ approaching 1 due to the considerably smaller variance of $\text{Var}(R_t)$ in comparison to $\text{Var}(S_t + R_t)$.

Tables 1 depict the results for each dataset. Notably, the ETT datasets and the Exchange dataset manifest conspicuous trends, whereas the Electricity, Solar, and Traffic datasets showcase marked seasonality. Additionally, the Exchange dataset stands out with distinctive features. Figure 6 also illustrates that with shorter prediction windows, the Exchange dataset sustains comparatively minor fluctuations, almost forming a linear trajectory. This enables effective forecasting through a straightforward batch mean approach. As the forecasting horizon extends, the dataset appears a more pronounced trend while retaining minimal seasonality.

### B.1.4 Quantifying Data Distribution Complexity

To differentiate between methods optimized for point or distributional forecasts, we aim to quantify the complexity of data distribution within a time-series segment whose window size equals the prediction horizon length. Such complexities may arise from the unpredictability of the data itself or from noises accidentally introduced during the data collection process [77].

We propose that assessing non-Gaussianity, i.e., how closely the distribution of time-series values within that window resembles a Gaussian distribution, could serve as a meaningful measure. This is because point forecasting methods, optimized with mean squared loss, are essentially equivalent to probabilistic counterparts that include a Gaussian output head and employ maximum a posteriori estimation. This suggests that point forecasting methods inherently assume that time-series values adhere to a Gaussian distribution. In contrast, advanced probabilistic methods, which do not make prior assumptions about data distribution, can adapt to complex data distributions in a data-driven manner.

Hence, we use the Jensen–Shannon divergence [50] to measure the similarity between the actual value distribution of a time-series segment and a Gaussian distribution fitted to the observed values. Short-term datasets used a window size of 30, while long-term datasets used a size of 336. By averaging the calculated divergence values across all time-series segments of a dataset, we obtain a dataset-level measure of non-Gaussianity. A larger divergence value indicates a larger deviation from a Gaussian distribution in the data.

## B.2 Metrics

In `ProbTS`, we integrate an extensive variety of metrics that take into account both point and distributional forecasts, thereby providing a comprehensive and multifaceted assessment of forecasting models.

---

[4]https://otexts.com/fpp2/stl.html

### B.2.1 Metrics for Point Forecasts

**Mean Absolute Error (MAE)**   The Mean Absolute Error (MAE) quantifies the average absolute deviation between the forecasts and the true values. Since it averages the absolute errors, MAE is robust to outliers. Its mathematical formula is given by:

$$\text{MAE} = \frac{1}{K \times T} \sum_{k=1}^{K} \sum_{t=1}^{T} |x_t^k - \hat{x}_t^k|,$$

where $K$ is the number of variates, $T$ is the length of series, $x_t^k$ and $\hat{x}_t^k$ denotes the ground-truth value and the predicted value, respectively. For multivariate time series, we also provide the aggregated version:

$$\text{MAE}_{\text{sum}} = \frac{1}{T} \sum_{t=1}^{T} |x_t^{\text{sum}} - \hat{x}_t^{\text{sum}}|,$$

where $x_t^{\text{sum}}$ and $\hat{x}_t^{\text{sum}}$ are the summation across the dimension $K$ of $x_t^k$ and $\hat{x}_t^k$, respectively.

**Normalized Mean Absolute Error (NMAE)**   The Normalized Mean Absolute Error (NMAE) is a normalized version of the MAE, which is dimensionless and facilitates the comparability of the error magnitude across different datasets or scales. The mathematical representation of NMAE is given by:

$$\text{NMAE} = \frac{\sum_{k=1}^{K} \sum_{t=1}^{T} |x_t^k - \hat{x}_t^k|}{\sum_{k=1}^{K} \sum_{t=1}^{T} |x_t^k|}.$$

Its aggregated version is:

$$\text{NMAE}_{\text{sum}} = \frac{\sum_{t=1}^{T} |x_t^{\text{sum}} - \hat{x}_t^{\text{sum}}|}{\sum_{t=1}^{T} |x_t^{\text{sum}}|}.$$

**Mean Squared Error (MSE)**   The Mean Squared Error (MSE) is a quantitative metric used to measure the average squared difference between the observed actual value and forecasts. It is defined mathematically as follows:

$$\text{MSE} = \frac{1}{K \times T} \sum_{k=1}^{K} \sum_{t=1}^{T} (x_t^k - \hat{x}_t^k)^2.$$

For multivariate time series, we also provide the aggregated version:

$$\text{MSE}_{\text{sum}} = \frac{1}{T} \sum_{t=1}^{T} (x_t^{\text{sum}} - \hat{x}_t^{\text{sum}})^2.$$

**Normalized Root Mean Squared Error (NRMSE)**   The Normalized Root Mean Squared Error (NRMSE) is a normalized version of the Root Mean Squared Error (RMSE), which quantifies the average squared magnitude of the error between forecasts and observations, normalized by the expectation of the observed values. It can be formally written as:

$$\text{NRMSE} = \frac{\sqrt{\frac{1}{K \times T} \sum_{k=1}^{K} \sum_{t=1}^{T} (x_t^k - \hat{x}_t^k)^2}}{\frac{1}{K \times T} \sum_{k=1}^{K} \sum_{t=1}^{T} |x_t^k|}.$$

For multivariate time series, we also provide the aggregated version:

$$\text{NRMSE}_{\text{sum}} = \frac{\sqrt{\frac{1}{T} \sum_{t=1}^{T} (x_t^{\text{sum}} - \hat{x}_t^{\text{sum}})^2}}{\frac{1}{T} \sum_{t=1}^{T} |x_t^{\text{sum}}|}.$$

**Mean Absolute Scaled Error (MASE)**   The Mean Absolute Scaled Error (MASE) divideds the MAE of forecasted values by MAE of the in-sample one-step naive forecast, which is a scale-invariant metrics:

$$\text{MASE} = \frac{\frac{1}{K \times T} \sum_{k=1}^{K} \sum_{t=1}^{T} |x_t^k - \hat{x}_t^k|}{\frac{1}{K \times T} \sum_{k=1}^{K} \sum_{t=1}^{T} |x_t^k - x_{t-1}^k|}.$$

### B.2.2 Metrics for Distributional Forecasts

**Continuous Ranked Probability Score (CRPS)**   The Continuous Ranked Probability Score (CRPS) [47] quantifies the agreement between a cumulative distribution function (CDF) $F$ and an observation $x$, represented as:

$$\mathrm{CRPS} = \int_{\mathbb{R}} (F(z) - \mathbb{I}\{x \leq z\})^2 dz,$$

where $\mathbb{I}\{x \leq z\}$ denotes the indicator function, equating to one if $x \leq z$ and zero otherwise.

Being a proper scoring function, CRPS reaches its minimum when the predictive distribution $F$ coincides with the data distribution. When using the empirical CDF of $F$, denoted as $\hat{F}(z) = \frac{1}{n} \sum_{i=1}^{n} \mathbb{I}\{X_i \leq z\}$, where $n$ represents the number of samples $X_i \sim F$, CRPS can be precisely calculated from the simulated samples of the conditional distribution $p_\theta(\boldsymbol{x}_t | \boldsymbol{h}_t)$. In our practice, 100 samples are employed to estimate the empirical CDF.

For multivariate time series, the aggregate CRPS, denoted as $\mathrm{CRPS}_{\mathrm{sum}}$, is derived by summing across the $K$ time series, both for the ground-truth data and sampled data, and subsequently averaging over the forecasting horizon. Formally, it is represented as:

$$\mathrm{CRPS}_{\mathrm{sum}} = \mathbb{E}_t \left[ \mathrm{CRPS} \left( \hat{F}_{\mathrm{sum}}(t), \sum_{i=1}^{K} x_{i,l}^0 \right) \right].$$

### B.3 Baselines

To ensure the integrity of the results, `ProbTS` adheres to a standard implementation process, employing unified data splitting, standardization techniques, and adopting fair settings for hyperparameter tuning across all methods.

**Implementation Details**   `ProbTS` was developed using PyTorch Lightning [22]. During training, we sampled 100 batches per epoch and limited training to 50 epochs, using the CRPS metric for checkpointing. All experiments employed the Adam optimizer and were run on single NVIDIA Tesla V100 GPUs with CUDA 11.3. To enable evaluation of distribution-level metrics, we conducted 100 samplings to calculate metrics on the test set.

Following the most commonly adopted settings [75, 49, 71], in the long-term forecasting context, all of the models are following the same experimental setup with prediction length $T \in \{24, 36, 48, 60\}$ for ILI-L dataset and $T \in \{96, 192, 336, 720\}$ for other datasets. Note that the lookback window here is 96 for all the models, to ensure a fair comparison. In the short-term forecasting context, the length of the lookback window is the same as the forecasting horizons, which are 30 for Exchange-S dataset and Wikipedia-S dataset, and 24 for the rest, the same as [59].

**Hyper-parameter Tuning**   For a fair comparison, we conducted a comprehensive grid search for critical hyperparameters across all models in this study. Table 7 details the shared hyperparameters tuned within the `ProbTS` pipeline, along with those kept constant. Due to the vast array of model-specific hyperparameters, we present an example configuration in Table 8. Complete hyperparameter configurations for each model, identified through this process, will be made available in a public GitHub repository for transparency and reproducibility.

**Implementation Details on Foundation Models**   We used reference implementations of eight time series foundation models into `ProbTS`.

For Lag-Llama [56], we use its official code[5] and integrate the `LagLlamaEstimator` with its pre-trained checkpoint. The look-back window is uniformly set to 512, irrespective of the forecast horizon. The other hyper-parameters are aligned with the recommended settings.

For Chronos [2], we use its official code[6] and integrate the `ChronosPipeline` into `ProbTS` using `amazon/chronos-t5` checkpoints (three models were tested: small, base and large). Two look-back

---

[5]`https://github.com/time-series-foundation-models/lag-llama`
[6]`https://github.com/amazon-science/chronos-forecasting`

Table 7: Hyper-parameters values fixed or range searched in hyper-parameter tuning.

| Hyper-parameter | Value or Range Searched |
|---|---|
| learning rate | [1e-4, 1e-3, 1e-2] |
| dropout | [0, 0.1, 0.2] |
| batch_size | [8, 16, 32, 64] |
| use_lags | [True, False] |
| use_feat_idx_emb | [True, False] |
| use_time_feat | [True, False] |
| autoregressive | [True, False] |
| scaler | [Standard, Scaling, None] |
| limit_train_batches | 100 |
| num_samples | 100 |
| quantiles_num | 20 |

Table 8: Hyperparameter settings for Electricity-S dataset.

| Model | Hyperparameter |
|---|---|
| DLinear | learning_rate=0.01, kernel_size=3, f_hidden_size=40 |
| PatchTST | learning_rate=0.0001, stride=3, patch_len=6, n_layers=3, n_heads=8, dropout=0.1, kernel_size=3, f_hidden_size=32 |
| TimesNet | learning_rate=0.001, n_layers=2, num_kernels=6, top_k=5, f_hidden_size=64, d_ff=64 |
| GRU NVP | learning_rate=0.001, f_hidden_size=40, num_layers=2, n_blocks=3, hidden_size=100, conditional_length=200 |
| GRU MAF | learning_rate=0.001, f_hidden_size=40, num_layers=2, n_blocks=4, hidden_size=100, conditional_length=200 |
| Trans MAF | learning_rate=0.001, f_hidden_size=32, num_heads=8, n_blocks=4, hidden_size=100, conditional_length=200 |
| TimeGrad | learning_rate=0.001, f_hidden_size=128, num_layers=4, conditional_length=100, beta_end=0.1, diff_steps=100 |
| CSDI | learning_rate=0.001, channels=64, emb_time_dim=128, emb_feature_dim=16, num_steps=50, num_heads=8, n_layers=4 |

windows are used during evaluation: 96 and 512. We set `limit_prediction_length=False` to enable it to predict horizons longer than 64. However, this may potentially lead to a decrease in predictive performance since the model was only trained to consider prediction lengths of 64 or less during pre-training.

For TimesFM [15], we modify its official code[7] into `ProbTS` and load checkpoints from `google/timesfm-1.0-200m`. Look-back window is set to 96 for a fair comparison.

For Timer [41], we modify its official code[8] into `ProbTS` and `Timer_67M_UTSD_4G` checkpoint downloaded from its repo. Look-back window is also set to 96 for a fair comparison.

For MOIRAI [70], we employ its official code[9] and load checkpoints from `Salesforce/moirai-1.0-R-base`. Two look-back windows are utilized during evaluation: 96 and 5000. The original experiments suggest that MOIRAI's forecasting capability can be consistently enhanced by increasing the look-back window. Consequently, we have included a 5000 look-back window to test the model's performance.

For UniTS [23], we have adapted its official code[10] into our `ProbTS` framework and loaded the `saved_weights` from its repo. The look-back window is set to 96 to ensure a fair comparison. It is important to note that this checkpoint was originally used for the Zero-Shot New-length Forecasting experiment in the original work (where models are challenged to predict new lengths by adjusting from the trained length, with offsets ranging from 0 to 384), which differs from the objectives of our experiments.

For ForecastPFN [18], we have integrated its official code[11] into our `ProbTS` framework and have utilized the `units_x128_pretrain_checkpoint`. However, we have encountered some challenges

---

[7] https://github.com/google-research/timesfm
[8] https://github.com/thuml/Large-Time-Series-Model
[9] https://github.com/SalesforceAIResearch/uni2ts
[10] https://github.com/abacusai/ForecastPFN
[11] https://github.com/SalesforceAIResearch/uni2ts

Table 9: Results (mean$_{\text{std}}$) on short-term forecasting scenarios, each containing five independent runs with different seeds.

| Model | Exchange-S | | Solar-S | | Electricity-S | | Traffic-S | | Wikipedia-S | |
| --- | --- | --- | --- | --- | --- | --- | --- | --- | --- | --- |
| | CRPS | NMAE | CRPS | NMAE | CRPS | NMAE | CRPS | NMAE | CRPS | NMAE |
| Glob. mean | 0.188 | 0.188 | 1.403 | 1.403 | 0.412 | 0.412 | 0.540 | 0.540 | 0.577 | 0.577 |
| Batch mean | 0.012 | 0.012 | 1.244 | 1.244 | 0.365 | 0.365 | 0.503 | 0.503 | 0.336 | 0.336 |
| Linear | $0.012_{001}$ | $0.012_{001}$ | $0.704_{036}$ | $0.704_{036}$ | $0.138_{009}$ | $0.138_{009}$ | $0.327_{032}$ | $0.327_{032}$ | $0.874_{151}$ | $0.874_{151}$ |
| GRU | $0.013_{002}$ | $0.013_{002}$ | $0.594_{144}$ | $0.594_{144}$ | $0.134_{009}$ | $0.134_{009}$ | $0.193_{002}$ | $0.193_{002}$ | $0.394_{013}$ | $0.394_{013}$ |
| Transformer | $0.016_{001}$ | $0.016_{001}$ | $0.538_{066}$ | $0.538_{066}$ | $0.115_{005}$ | $0.115_{005}$ | $0.204_{006}$ | $0.204_{006}$ | $0.408_{011}$ | $0.408_{011}$ |
| N-HiTS | $0.012_{000}$ | $0.012_{000}$ | $0.572_{020}$ | $0.572_{020}$ | $0.074_{003}$ | $0.074_{003}$ | $0.193_{002}$ | $0.193_{002}$ | $0.332_{011}$ | $0.332_{011}$ |
| NLinear | $0.010_{000}$ | $\mathbf{0.010_{000}}$ | $0.560_{002}$ | $0.560_{002}$ | $0.083_{002}$ | $0.083_{002}$ | $0.233_{001}$ | $0.233_{001}$ | $0.321_{001}$ | $0.321_{001}$ |
| DLinear | $0.012_{001}$ | $0.012_{001}$ | $0.547_{009}$ | $0.547_{009}$ | $0.076_{003}$ | $0.076_{003}$ | $0.250_{002}$ | $0.250_{002}$ | $0.412_{001}$ | $0.412_{001}$ |
| PatchTST | $0.010_{000}$ | $\mathbf{0.010_{000}}$ | $0.496_{002}$ | $0.496_{002}$ | $0.067_{001}$ | $0.067_{001}$ | $0.202_{001}$ | $0.202_{001}$ | $0.257_{001}$ | $\mathbf{0.257_{001}}$ |
| TimesNet | $0.011_{001}$ | $0.011_{001}$ | $0.507_{019}$ | $0.507_{019}$ | $0.071_{002}$ | $0.071_{002}$ | $0.205_{002}$ | $0.205_{002}$ | $0.304_{002}$ | $0.304_{002}$ |
| iTransformer | $0.010_{000}$ | $0.010_{000}$ | $0.496_{000}$ | $0.496_{000}$ | $0.074_{000}$ | $0.074_{000}$ | $0.158_{000}$ | $\mathbf{0.158_{000}}$ | $0.262_{000}$ | $0.262_{000}$ |
| GRU NVP | $0.016_{003}$ | $0.020_{003}$ | $0.396_{021}$ | $0.507_{022}$ | $0.055_{002}$ | $0.073_{003}$ | $0.161_{006}$ | $0.203_{009}$ | $0.282_{003}$ | $0.330_{003}$ |
| GRU MAF | $0.015_{001}$ | $0.020_{001}$ | $0.386_{026}$ | $0.492_{027}$ | $0.051_{001}$ | $0.067_{001}$ | $0.131_{006}$ | $0.165_{009}$ | $0.281_{004}$ | $0.337_{005}$ |
| Trans MAF | $0.011_{001}$ | $0.014_{001}$ | $0.400_{022}$ | $0.503_{022}$ | $0.054_{004}$ | $0.071_{005}$ | $\mathbf{0.129_{004}}$ | $0.165_{006}$ | $0.289_{008}$ | $0.344_{008}$ |
| TimeGrad | $0.011_{001}$ | $0.014_{002}$ | $\mathbf{0.359_{011}}$ | $\mathbf{0.445_{023}}$ | $0.052_{001}$ | $0.067_{001}$ | $0.164_{091}$ | $0.201_{115}$ | $0.272_{008}$ | $0.327_{011}$ |
| CSDI | $\mathbf{0.008_{000}}$ | $0.011_{000}$ | $0.366_{005}$ | $0.484_{008}$ | $\mathbf{0.050_{001}}$ | $\mathbf{0.065_{001}}$ | $0.146_{012}$ | $0.176_{013}$ | $\mathbf{0.219_{006}}$ | $0.259_{009}$ |

in replicating the performance levels reported in the original paper across various datasets. It appears that our experience aligns with the observations made in the Chronos paper [2], specifically as depicted in Figure 5, where ForecastPFN's performance was not as robust as initially anticipated.

For TTM [20], we have adapted its official code[12] into our `ProbTS` framework and have loaded the `ibm-granite/granite-timeseries-ttm-v1` checkpoint. However, this method does not support arbitrary lengths for forecasting. The publicly available model currently supports a forecast length of 96 only, and thus, we have not conducted evaluations at other forecast lengths.

It is worth noting that, due to time constraints, we did not adjust the context window for all models. Instead, we chose 96 as a balanced and fair window size, which might result in suboptimal performance for some models.

### B.4 Data and Code Availability

We release the ProbTS toolkit, documentation, and running scripts at `https://github.com/microsoft/ProbTS` under the MIT license. The repository includes parameter configurations for benchmarking experiments, ensuring reproducibility of all results presented in the paper. Most datasets used in this paper licensed under Creative Commons Attribution 4.0 International (CC BY 4.0), accessible via instructions in the repository.

## C Overall Comparison Results

### C.1 An Overall Comparison of Traditional Time-series Models on Short-term Forecasting

Table 9 presents a comprehensive comparison of various time-series models in `ProbTS` on short-term forecasting scenarios. The results, reported as mean ± standard deviation, are derived from five independent runs with different seeds for each scenario.

### C.2 An Overall Comparison of Traditional Time-series Models on Long-term Forecasting

Table 10 presents a comprehensive comparison of various time-series models in `ProbTS` on long-term forecasting scenarios. The results, reported as mean ± standard deviation, are derived from five independent runs with different seeds for each scenario. The input sequence length is set to 36 for the ILI dataset and 96 for the others. Due to the excessive time and memory consumption of CSDI in producing long-term forecasts, its results are unavailable in some datasets.

---

[12]`https://github.com/ibm-granite/granite-tsfm/blob/main/notebooks/hfdemo/ttm_getting_started.ipynb`

Table 10: Results (mean$_{std}$) on long-term forecasting scenarios, each containing five independent runs with different seeds. The input sequence length is set to 36 for the ILI-L dataset and 96 for the others. Due to the excessive time and memory consumption of CSDI in producing long-term forecasts, its results are unavailable in some datasets.

| Dataset | Pred len | iTransformer CRPS | iTransformer NMAE | PatchTST CRPS | PatchTST NMAE | DLinear CRPS | DLinear NMAE | Autoformer CRPS | Autoformer NMAE | CSDI CRPS | CSDI NMAE | TimeGrad CRPS | TimeGrad NMAE | GRU NVP CRPS | GRU NVP NMAE |
|---|---|---|---|---|---|---|---|---|---|---|---|---|---|---|---|
| ETTm1-L | 96 | 0.271$_{.000}$ | **0.271**$_{.000}$ | 0.272$_{.001}$ | 0.272$_{.001}$ | 0.282$_{.002}$ | 0.282$_{.002}$ | 0.388$_{.001}$ | 0.388$_{.001}$ | **0.236**$_{.006}$ | 0.308$_{.005}$ | 0.522$_{.105}$ | 0.645$_{.129}$ | 0.383$_{.053}$ | 0.488$_{.058}$ |
|  | 192 | 0.301$_{.000}$ | 0.301$_{.000}$ | 0.295$_{.001}$ | 0.295$_{.001}$ | 0.309$_{.004}$ | 0.309$_{.004}$ | 0.442$_{.001}$ | 0.442$_{.001}$ | **0.291**$_{.025}$ | 0.377$_{.026}$ | 0.603$_{.092}$ | 0.748$_{.084}$ | 0.396$_{.030}$ | 0.514$_{.042}$ |
|  | 336 | 0.333$_{.000}$ | 0.333$_{.000}$ | 0.323$_{.001}$ | 0.323$_{.001}$ | 0.338$_{.008}$ | 0.338$_{.008}$ | 0.429$_{.000}$ | 0.429$_{.000}$ | **0.322**$_{.033}$ | 0.419$_{.042}$ | 0.601$_{.028}$ | 0.759$_{.015}$ | 0.486$_{.032}$ | 0.630$_{.029}$ |
|  | 720 | 0.376$_{.000}$ | 0.376$_{.000}$ | 0.353$_{.001}$ | 0.353$_{.001}$ | 0.387$_{.006}$ | 0.387$_{.006}$ | 0.440$_{.000}$ | 0.440$_{.000}$ | 0.448$_{.038}$ | 0.578$_{.051}$ | 0.621$_{.037}$ | 0.793$_{.034}$ | 0.546$_{.036}$ | 0.707$_{.050}$ |
| ETTm2-L | 96 | 0.137$_{.000}$ | 0.137$_{.000}$ | 0.132$_{.001}$ | 0.132$_{.001}$ | 0.138$_{.000}$ | 0.138$_{.000}$ | 0.158$_{.000}$ | 0.158$_{.000}$ | **0.115**$_{.009}$ | 0.146$_{.012}$ | 0.427$_{.042}$ | 0.525$_{.047}$ | 0.319$_{.044}$ | 0.413$_{.059}$ |
|  | 192 | 0.161$_{.000}$ | 0.161$_{.000}$ | 0.157$_{.001}$ | 0.157$_{.001}$ | 0.163$_{.003}$ | 0.163$_{.003}$ | 0.175$_{.000}$ | 0.175$_{.000}$ | **0.147**$_{.008}$ | 0.189$_{.012}$ | 0.424$_{.061}$ | 0.530$_{.060}$ | 0.326$_{.025}$ | 0.427$_{.033}$ |
|  | 336 | 0.180$_{.000}$ | 0.180$_{.000}$ | 0.176$_{.000}$ | 0.176$_{.000}$ | 0.188$_{.001}$ | 0.188$_{.001}$ | 0.191$_{.000}$ | 0.191$_{.000}$ | 0.190$_{.018}$ | 0.248$_{.024}$ | 0.469$_{.049}$ | 0.566$_{.047}$ | 0.449$_{.145}$ | 0.580$_{.169}$ |
|  | 720 | 0.211$_{.000}$ | 0.211$_{.000}$ | 0.205$_{.001}$ | 0.205$_{.001}$ | 0.219$_{.003}$ | 0.219$_{.003}$ | 0.217$_{.000}$ | 0.217$_{.000}$ | 0.239$_{.035}$ | 0.306$_{.040}$ | 0.470$_{.054}$ | 0.561$_{.044}$ | 0.561$_{.273}$ | 0.749$_{.385}$ |
| ETTh1-L | 96 | **0.321**$_{.000}$ | **0.321**$_{.000}$ | 0.328$_{.003}$ | 0.328$_{.003}$ | 0.352$_{.011}$ | 0.352$_{.011}$ | 0.367$_{.000}$ | 0.367$_{.000}$ | 0.437$_{.018}$ | 0.557$_{.022}$ | 0.455$_{.046}$ | 0.585$_{.058}$ | 0.379$_{.030}$ | 0.481$_{.037}$ |
|  | 192 | **0.359**$_{.000}$ | **0.359**$_{.000}$ | 0.359$_{.002}$ | 0.359$_{.002}$ | 0.393$_{.001}$ | 0.393$_{.001}$ | 0.392$_{.000}$ | 0.392$_{.000}$ | 0.496$_{.051}$ | 0.625$_{.065}$ | 0.516$_{.038}$ | 0.680$_{.058}$ | 0.425$_{.019}$ | 0.531$_{.018}$ |
|  | 336 | 0.388$_{.000}$ | 0.388$_{.000}$ | 0.384$_{.002}$ | 0.384$_{.002}$ | 0.419$_{.007}$ | 0.419$_{.007}$ | 0.398$_{.000}$ | 0.398$_{.000}$ | 0.454$_{.025}$ | 0.574$_{.026}$ | 0.512$_{.026}$ | 0.666$_{.047}$ | 0.458$_{.054}$ | 0.580$_{.064}$ |
|  | 720 | 0.408$_{.000}$ | 0.408$_{.000}$ | 0.397$_{.002}$ | 0.397$_{.002}$ | 0.502$_{.029}$ | 0.502$_{.029}$ | 0.433$_{.000}$ | 0.433$_{.000}$ | 0.528$_{.012}$ | 0.657$_{.014}$ | 0.523$_{.027}$ | 0.672$_{.015}$ | 0.502$_{.039}$ | 0.643$_{.046}$ |
| ETTh2-L | 96 | **0.177**$_{.000}$ | **0.177**$_{.000}$ | 0.177$_{.000}$ | 0.177$_{.000}$ | 0.211$_{.027}$ | 0.211$_{.027}$ | 0.203$_{.000}$ | 0.203$_{.000}$ | **0.164**$_{.013}$ | 0.214$_{.018}$ | 0.358$_{.026}$ | 0.448$_{.031}$ | 0.432$_{.141}$ | 0.548$_{.158}$ |
|  | 192 | 0.203$_{.000}$ | 0.203$_{.000}$ | 0.201$_{.001}$ | 0.201$_{.001}$ | 0.238$_{.028}$ | 0.238$_{.028}$ | 0.226$_{.000}$ | 0.226$_{.000}$ | 0.226$_{.018}$ | 0.294$_{.027}$ | 0.457$_{.081}$ | 0.575$_{.089}$ | 0.625$_{.170}$ | 0.766$_{.223}$ |
|  | 336 | 0.243$_{.000}$ | 0.243$_{.000}$ | 0.240$_{.001}$ | 0.240$_{.001}$ | 0.284$_{.008}$ | 0.284$_{.008}$ | 0.264$_{.000}$ | 0.264$_{.000}$ | 0.274$_{.022}$ | 0.353$_{.028}$ | 0.481$_{.078}$ | 0.606$_{.095}$ | 0.793$_{.319}$ | 0.942$_{.408}$ |
|  | 720 | 0.264$_{.000}$ | 0.264$_{.000}$ | 0.252$_{.000}$ | 0.252$_{.000}$ | 0.307$_{.000}$ | 0.307$_{.000}$ | 0.287$_{.000}$ | 0.287$_{.000}$ | 0.302$_{.040}$ | 0.382$_{.030}$ | 0.445$_{.016}$ | 0.550$_{.018}$ | 0.539$_{.090}$ | 0.688$_{.161}$ |
| Electricity-L | 96 | 0.098$_{.000}$ | 0.098$_{.000}$ | 0.086$_{.001}$ | 0.086$_{.001}$ | 0.090$_{.001}$ | 0.090$_{.001}$ | 0.140$_{.000}$ | 0.140$_{.000}$ | 0.153$_{.137}$ | 0.203$_{.189}$ | 0.096$_{.002}$ | 0.119$_{.003}$ | 0.094$_{.003}$ | 0.118$_{.003}$ |
|  | 192 | 0.106$_{.000}$ | 0.106$_{.000}$ | 0.092$_{.001}$ | 0.092$_{.001}$ | 0.095$_{.001}$ | 0.095$_{.001}$ | 0.136$_{.000}$ | 0.136$_{.000}$ | 0.200$_{.094}$ | 0.264$_{.129}$ | 0.100$_{.004}$ | 0.124$_{.005}$ | 0.097$_{.002}$ | 0.121$_{.003}$ |
|  | 336 | 0.115$_{.000}$ | 0.115$_{.000}$ | 0.100$_{.000}$ | 0.100$_{.000}$ | 0.104$_{.000}$ | 0.104$_{.000}$ | 0.147$_{.000}$ | 0.147$_{.000}$ | — | — | 0.102$_{.007}$ | 0.126$_{.008}$ | **0.099**$_{.001}$ | 0.123$_{.001}$ |
|  | 720 | 0.133$_{.000}$ | 0.133$_{.000}$ | 0.116$_{.000}$ | 0.116$_{.000}$ | 0.122$_{.001}$ | 0.122$_{.001}$ | 0.159$_{.000}$ | 0.159$_{.000}$ | — | — | 0.108$_{.003}$ | 0.134$_{.004}$ | 0.114$_{.013}$ | 0.144$_{.017}$ |
| Traffic-L | 96 | 0.246$_{.000}$ | 0.246$_{.000}$ | 0.248$_{.001}$ | 0.248$_{.001}$ | 0.356$_{.009}$ | 0.356$_{.009}$ | 0.293$_{.000}$ | 0.293$_{.000}$ | — | — | 0.202$_{.004}$ | 0.234$_{.006}$ | **0.187**$_{.002}$ | **0.231**$_{.003}$ |
|  | 192 | 0.259$_{.000}$ | 0.259$_{.000}$ | 0.245$_{.001}$ | 0.245$_{.001}$ | 0.346$_{.009}$ | 0.346$_{.009}$ | 0.318$_{.000}$ | 0.318$_{.000}$ | — | — | 0.208$_{.003}$ | 0.239$_{.004}$ | **0.192**$_{.001}$ | **0.236**$_{.002}$ |
|  | 336 | 0.283$_{.000}$ | 0.283$_{.000}$ | 0.257$_{.002}$ | 0.257$_{.002}$ | 0.350$_{.008}$ | 0.350$_{.008}$ | 0.332$_{.000}$ | 0.332$_{.000}$ | — | — | 0.213$_{.003}$ | 0.246$_{.003}$ | **0.201**$_{.004}$ | 0.248$_{.006}$ |
|  | 720 | 0.275$_{.000}$ | 0.275$_{.000}$ | 0.266$_{.001}$ | 0.266$_{.001}$ | 0.365$_{.009}$ | 0.365$_{.009}$ | 0.341$_{.003}$ | 0.341$_{.003}$ | — | — | 0.220$_{.002}$ | 0.263$_{.001}$ | **0.211**$_{.004}$ | 0.264$_{.006}$ |
| Weather-L | 96 | 0.089$_{.000}$ | 0.089$_{.000}$ | 0.087$_{.002}$ | 0.087$_{.002}$ | 0.112$_{.001}$ | 0.112$_{.001}$ | 0.239$_{.004}$ | 0.239$_{.004}$ | **0.068**$_{.008}$ | **0.087**$_{.012}$ | 0.130$_{.017}$ | 0.164$_{.023}$ | 0.116$_{.013}$ | 0.145$_{.017}$ |
|  | 192 | 0.093$_{.000}$ | 0.093$_{.000}$ | 0.090$_{.001}$ | 0.090$_{.001}$ | 0.122$_{.001}$ | 0.122$_{.001}$ | 0.213$_{.000}$ | 0.213$_{.000}$ | **0.068**$_{.006}$ | **0.086**$_{.007}$ | 0.127$_{.019}$ | 0.158$_{.024}$ | 0.122$_{.021}$ | 0.147$_{.025}$ |
|  | 336 | 0.096$_{.000}$ | 0.096$_{.000}$ | 0.092$_{.002}$ | 0.092$_{.002}$ | 0.130$_{.002}$ | 0.130$_{.002}$ | 0.176$_{.000}$ | 0.176$_{.000}$ | **0.083**$_{.002}$ | 0.098$_{.002}$ | 0.130$_{.006}$ | 0.162$_{.006}$ | 0.128$_{.011}$ | 0.160$_{.012}$ |
|  | 720 | 0.099$_{.000}$ | 0.099$_{.000}$ | 0.094$_{.001}$ | 0.094$_{.001}$ | 0.144$_{.001}$ | 0.144$_{.001}$ | 0.170$_{.001}$ | 0.170$_{.001}$ | **0.087**$_{.003}$ | 0.102$_{.005}$ | 0.113$_{.011}$ | 0.136$_{.020}$ | 0.110$_{.004}$ | 0.135$_{.008}$ |
| Exchange-L | 96 | 0.025$_{.000}$ | 0.025$_{.000}$ | 0.023$_{.000}$ | 0.023$_{.000}$ | 0.024$_{.000}$ | 0.024$_{.000}$ | 0.032$_{.000}$ | 0.032$_{.000}$ | 0.028$_{.003}$ | 0.036$_{.005}$ | 0.068$_{.003}$ | 0.079$_{.002}$ | 0.071$_{.006}$ | 0.091$_{.009}$ |
|  | 192 | 0.036$_{.000}$ | 0.036$_{.000}$ | 0.034$_{.000}$ | 0.034$_{.000}$ | 0.035$_{.000}$ | 0.035$_{.000}$ | 0.041$_{.000}$ | 0.041$_{.000}$ | 0.045$_{.003}$ | 0.058$_{.005}$ | 0.087$_{.013}$ | 0.100$_{.019}$ | 0.068$_{.004}$ | 0.087$_{.005}$ |
|  | 336 | **0.048**$_{.000}$ | **0.048**$_{.000}$ | 0.048$_{.000}$ | 0.048$_{.000}$ | 0.048$_{.001}$ | 0.048$_{.001}$ | 0.056$_{.000}$ | 0.056$_{.000}$ | 0.060$_{.004}$ | 0.076$_{.006}$ | 0.074$_{.009}$ | 0.086$_{.008}$ | 0.072$_{.002}$ | 0.091$_{.002}$ |
|  | 720 | 0.076$_{.000}$ | 0.076$_{.000}$ | 0.072$_{.000}$ | 0.072$_{.000}$ | 0.075$_{.002}$ | 0.075$_{.002}$ | 0.112$_{.002}$ | 0.112$_{.002}$ | 0.143$_{.020}$ | 0.173$_{.020}$ | 0.099$_{.015}$ | 0.113$_{.016}$ | 0.079$_{.009}$ | 0.103$_{.009}$ |
| ILI-L | 24 | **0.094**$_{.000}$ | **0.094**$_{.000}$ | 0.169$_{.005}$ | 0.169$_{.005}$ | 0.213$_{.038}$ | 0.213$_{.038}$ | 0.122$_{.000}$ | 0.122$_{.000}$ | 0.250$_{.013}$ | 0.263$_{.012}$ | 0.275$_{.047}$ | 0.296$_{.044}$ | 0.257$_{.003}$ | 0.283$_{.001}$ |
|  | 36 | **0.102**$_{.000}$ | **0.102**$_{.000}$ | 0.156$_{.005}$ | 0.156$_{.005}$ | 0.230$_{.015}$ | 0.230$_{.015}$ | 0.111$_{.000}$ | 0.111$_{.000}$ | 0.285$_{.010}$ | 0.298$_{.011}$ | 0.272$_{.057}$ | 0.298$_{.048}$ | 0.281$_{.004}$ | 0.307$_{.007}$ |
|  | 48 | **0.103**$_{.000}$ | **0.103**$_{.000}$ | 0.156$_{.008}$ | 0.156$_{.008}$ | 0.221$_{.009}$ | 0.221$_{.009}$ | 0.134$_{.000}$ | 0.134$_{.000}$ | 0.285$_{.036}$ | 0.301$_{.034}$ | 0.295$_{.033}$ | 0.320$_{.025}$ | 0.288$_{.008}$ | 0.314$_{.009}$ |
|  | 60 | **0.128**$_{.000}$ | **0.128**$_{.000}$ | 0.147$_{.003}$ | 0.147$_{.003}$ | 0.230$_{.013}$ | 0.230$_{.013}$ | 0.144$_{.000}$ | 0.144$_{.000}$ | 0.283$_{.012}$ | 0.299$_{.013}$ | 0.295$_{.083}$ | 0.325$_{.068}$ | 0.307$_{.005}$ | 0.333$_{.005}$ |

## C.3 An Overall Comparison of Time-series Foundation Models on Diverse Prediction Horizons

Based on the results in Table 4, we selected several datasets that most foundation models have not been pre-trained on for zero-shot evaluation. We compared these foundation models with fully-supervised traditional non-universal time-series models. The results, presented in Table 11, are reported as normalized MAE (NMAE). To comprehensively evaluate short-term and long-term forecasting performance, we selected prediction horizons of $\{24, 48, 96, 192, 336, 720\}$. Due to time constraints, the context window for most models was set to 96 unless otherwise specified. For Lag-Llama, we chose a context window of 512 based on explicit recommendations from the original paper and model specifications.

Additionally, on the MOIRAI model, we explored the impact of longer context windows. In some datasets (e.g., ETTh2-L, Weather-L), there were significant improvements, which we retained for reference. Other models also utilized longer context windows, but without consistent performance gains. It is worth noting that the longer context window of MOIRAI had an adverse effect on the Electricity dataset. To achieve optimal performance on unseen data, these models may require a hyperparameter search using validation data, which we leave for future work.

Table 11: Results of time-series foundation models on diverse prediction horizons. Mean NMAE value of five independent runs with different seeds is reported. The input sequence length is set to 96 if not specified. For every model, we exclude the evaluation results on its pre-trained datasets

| Dataset | Pred | MOIRAI-5000 | MOIRAI | Lag-Llama-512 | Chronos | TimesFM | Timer | UniTS | ForecastPFN | CSDI | DLinear | PatchTST | iTransformer |
|---|---|---|---|---|---|---|---|---|---|---|---|---|---|
| ETTm1-L | 24 | 0.144 | 0.128 | 0.231 | **0.113** | 0.133 | 0.182 | 0.353 | 0.878 | 0.115 | 0.121 | 0.199 | 0.116 |
|  | 48 | 0.269 | 0.343 | 0.244 | 0.307 | 0.269 | 0.339 | 0.432 | 0.997 | 0.266 | 0.238 | **0.199** | 0.243 |
|  | 96 | 0.296 | 0.449 | 0.399 | 0.393 | 0.324 | 0.384 | 0.457 | 0.961 | 0.303 | 0.283 | 0.271 | **0.269** |
|  | 192 | 0.311 | 0.479 | 0.416 | 0.422 | 0.378 | 0.423 | 0.466 | 1.031 | 0.389 | 0.309 | **0.295** | 0.304 |
|  | 336 | **0.321** | 0.471 | 0.429 | 0.439 | 0.435 | 0.446 | 0.475 | 1.012 | 0.449 | 0.337 | 0.324 | 0.337 |
|  | 720 | **0.351** | 0.515 | 0.472 | 0.467 | 0.511 | 0.480 | 0.499 | 1.053 | 0.530 | 0.385 | 0.353 | 0.380 |
| ETTm2-L | 24 | 0.100 | 0.113 | 0.128 | 0.110 | 0.117 | 0.130 | 0.173 | 1.506 | **0.096** | 0.103 | 0.162 | 0.106 |
|  | 48 | **0.116** | 0.132 | 0.162 | 0.132 | 0.132 | 0.141 | 0.165 | 1.471 | 0.121 | 0.121 | 0.162 | 0.118 |
|  | 96 | 0.141 | 0.168 | 0.188 | 0.158 | 0.156 | 0.153 | 0.169 | 1.386 | 0.133 | 0.138 | **0.133** | 0.141 |
|  | 192 | 0.159 | 0.192 | 0.205 | 0.183 | 0.186 | 0.175 | 0.184 | 1.397 | 0.201 | 0.167 | **0.158** | 0.158 |
|  | 336 | **0.173** | 0.209 | 0.226 | 0.206 | 0.209 | 0.193 | 0.199 | 1.422 | 0.226 | 0.188 | 0.176 | 0.184 |
|  | 720 | **0.200** | 0.242 | 0.249 | 0.240 | 0.246 | 0.220 | 0.223 | 1.485 | 0.264 | 0.222 | 0.206 | 0.213 |
| ETTh1-L | 24 | 0.304 | 0.297 | 0.313 | **0.265** | 0.277 | 0.315 | 0.453 | 1.141 | 0.292 | 0.277 | 0.356 | 0.275 |
|  | 48 | 0.312 | 0.326 | 0.341 | **0.294** | 0.307 | 0.339 | 0.461 | 1.161 | 0.392 | 0.311 | 0.356 | 0.304 |
|  | 96 | 0.324 | 0.354 | 0.353 | 0.321 | 0.332 | 0.361 | 0.469 | 1.157 | 0.534 | 0.340 | 0.326 | **0.319** |
|  | 192 | **0.350** | 0.393 | 0.376 | 0.375 | 0.383 | 0.399 | 0.482 | 1.226 | 0.698 | 0.394 | 0.357 | 0.364 |
|  | 336 | **0.366** | 0.417 | 0.393 | 0.410 | 0.411 | 0.433 | 0.500 | 1.183 | 0.603 | 0.423 | 0.383 | 0.389 |
|  | 720 | **0.380** | 0.448 | 0.424 | 0.432 | 0.418 | 0.460 | 0.499 | 1.037 | 0.664 | 0.501 | 0.399 | 0.402 |
| ETTh2-L | 24 | **0.126** | 0.142 | 0.160 | 0.127 | 0.134 | 0.146 | 0.197 | 1.650 | 0.133 | 0.146 | 0.213 | 0.135 |
|  | 48 | **0.147** | 0.170 | 0.195 | 0.161 | 0.166 | 0.166 | 0.199 | 1.667 | 0.186 | 0.176 | 0.213 | 0.161 |
|  | 96 | **0.162** | 0.200 | 0.203 | 0.194 | 0.190 | 0.179 | 0.204 | 1.685 | 0.204 | 0.209 | 0.176 | 0.177 |
|  | 192 | **0.189** | 0.244 | 0.220 | 0.225 | 0.220 | 0.205 | 0.223 | 1.768 | 0.272 | 0.206 | 0.200 | 0.205 |
|  | 336 | **0.224** | 0.269 | 0.245 | 0.252 | 0.266 | 0.247 | 0.262 | 1.719 | 0.321 | 0.293 | 0.240 | 0.244 |
|  | 720 | **0.243** | 0.293 | 0.248 | 0.290 | 0.277 | 0.254 | 0.264 | 1.542 | 0.417 | 0.307 | 0.251 | 0.263 |
| Weather-L | 24 | **0.043** | 0.060 | 0.062 | 0.060 | – | 0.063 | – | 1.916 | 0.050 | 0.095 | 0.083 | 0.074 |
|  | 48 | **0.066** | 0.110 | 0.086 | 0.128 | – | 0.098 | – | 1.924 | 0.093 | 0.125 | 0.156 | 0.090 |
|  | 96 | **0.074** | 0.134 | 0.096 | 0.163 | – | 0.109 | – | 1.924 | 0.092 | 0.113 | 0.086 | 0.089 |
|  | 192 | **0.074** | 0.126 | 0.103 | 0.161 | – | 0.116 | – | 1.926 | 0.079 | 0.121 | 0.089 | 0.099 |
|  | 336 | **0.075** | 0.129 | 0.109 | 0.177 | – | 0.121 | – | 1.930 | 0.100 | 0.131 | 0.092 | 0.100 |
|  | 720 | **0.076** | 0.151 | 0.120 | 0.208 | – | 0.124 | – | 1.970 | 0.108 | 0.145 | 0.094 | 0.111 |
| Electricity-L | 24 | 0.227 | 0.091 | – | – | – | 0.114 | – | – | – | 0.093 | – | **0.085** |
|  | 48 | 0.210 | 0.100 | – | – | – | 0.128 | – | – | – | 0.097 | – | **0.089** |
|  | 96 | 0.194 | 0.101 | – | – | – | 0.130 | – | – | 0.153 | 0.090 | **0.086** | 0.098 |
|  | 192 | 0.200 | 0.107 | – | – | – | 0.147 | – | – | 0.200 | 0.095 | **0.092** | 0.106 |
|  | 336 | 0.202 | 0.119 | – | – | – | 0.168 | – | – | – | 0.104 | **0.100** | 0.115 |
|  | 720 | 0.217 | 0.190 | – | – | – | 0.205 | – | – | – | 0.121 | **0.116** | 0.133 |

## C.4 An Overall Comparison of Time-series Foundation Models on Short-term Probabilistic Forecasting

Table 12 presents a comparison of two time-series probabilistic foundation models in short-term forecasting scenarios. We excluded the results of datasets that has been used in pre-training based on table 4. We also explored both short and long context windows for MOIRAI.

Table 12: Results of probabilistic foundation models on short-term distributional forecasting. For every model, we exclude the evaluation results on its pre-trained datasets.

| Model | Exchange-S | | Solar-S | | Electricity-S | | Traffic-S | |
|---|---|---|---|---|---|---|---|---|
| | CRPS | NMAE | CRPS | NMAE | CRPS | NMAE | CRPS | NMAE |
| CSDI | $0.008_{.000}$ | $0.011_{.000}$ | $\mathbf{0.366_{.005}}$ | $\mathbf{0.484_{.008}}$ | $\mathbf{0.050_{.001}}$ | $\mathbf{0.065_{.001}}$ | $\mathbf{0.146_{.012}}$ | $\mathbf{0.176_{.013}}$ |
| Chronos | 0.007 | 0.010 | – | – | – | – | 0.178 | 0.211 |
| MOIRAI-96 | **0.007** | **0.010** | 0.502 | 0.681 | 0.069 | 0.084 | – | – |
| MOIRAI-5000 | 0.007 | 0.010 | 0.521 | 0.702 | 0.059 | 0.079 | – | – |

# D  Additional Results and Experiments

## D.1  The Impact of Normalization

Normalization is a crucial aspect of time-series models, with different research branches adopting distinct strategies that can affect model performance across various data scenarios. Motivated by this, we provide a detailed analysis of different normalization methods in this section. Unless stated otherwise, our benchmarking follows the default normalization methods used by each model.

### D.1.1  Different Normalization Choice Between Distinct Research Branches

In time series forecasting, normalization typically occurs in two stages. First, during preprocessing, *dataset-level normalization* is applied, where global statistics (e.g., mean and standard deviation) from the training set are used to normalize all time-series values. Then, a local normalization module might be used to perform *instance-level normalization* when feeding a batch of time-series segments into the model.

Different research branches prefer distinct instance-level normalization strategies. Long-term point forecasting models [49, 40] typically adopt the RevIN [32]. Given a batch of time-series segments within a lookback window, RevIN applies a per-series z-score normalization, augmented with learnable affine parameters. Its main advantage is its effectiveness in addressing distribution shifts, particularly in long-term forecasting.

In contrast, most short-term probabilistic forecasting models [57, 58] employ an ad-hoc but still effective normalization strategy. For example, given a batch of time-series segments $X \in \mathbb{R}^{K \times L}$ (where $K$ is the number of variables and $L$ is the length of the lookback window), a per-series scaling is applied as $X_i^{\mathrm{norm}} = \frac{X_i}{\sum_{t=1}^{L} |X_{i,t}|/L}$, $i = 1, \ldots, K$ to stabilize value ranges. For simplicity, we refer to this type of normalization as *Mean Scaling*.

We summarize the instance-level normalization choices originally used by each model in the Table 13.

Table 13: Original instance-level normalization choices of each model.

| Normalization Choice | Model |
|---|---|
| ReVIN | iTransformer, PatchTST |
| Mean Scaling | TimeGrad, GRU NVP |
| w/o Norm | GRU, CSDI, DLinear |

Existing probabilistic models rarely use RevIN and are seldom combined with AR-based models that employ RevIN-style normalization. Similarly, the mean scaling strategy, commonly used in probabilistic forecasting models, is rarely applied to models designed for long-term forecasting. To better understand the effects of different normalization strategies, we selected representative models from both categories and combined them with three normalization methods: *RevIN*, *Scaling* (i.e., mean scaling), and *w/o Norm* (no instance-level normalization, using time-series values as provided by the dataset-level preprocessing). The results of these experiments are presented in Table 14, 15, 16, and 17.

Table 14: The impact of different normalization methods in short-term forecasting scenarios (CRPS).

| Model | PatchTST | | | CSDI | | | TimeGrad | | | GRU NVP | | |
|---|---|---|---|---|---|---|---|---|---|---|---|---|
| | ReVIN | Scaling | w/o Norm | ReVIN | Scaling | w/o Norm | ReVIN | Scaling | w/o Norm | ReVIN | Scaling | w/o Norm |
| Electricity-S | 0.0659 | **0.0645** | 0.0660 | 0.0524 | - | 0.0502 | 0.0673 | **0.0563** | 0.9681 | 0.0659 | **0.0706** | 0.1607 |
| Exchange Rate-S | **0.0102** | 0.0108 | 0.0111 | **0.0070** | 0.0083 | 0.0110 | 0.0100 | **0.0093** | 0.0170 | **0.0090** | 0.0147 | 0.0133 |
| Solar-S | **0.6275** | 0.7105 | 0.7169 | 0.4903 | **0.4347** | 0.4603 | **0.4945** | 0.5455 | 0.8356 | 0.9293 | 0.5926 | **0.4393** |
| Traffic-S | **0.2001** | 0.2036 | 0.2168 | 0.1505 | 0.1552 | **0.1389** | 0.1806 | 0.1280 | 0.1400 | 0.1827 | **0.1770** | 0.2277 |
| Wikipedia-S | **0.2529** | 0.3245 | 0.3695 | 0.2164 | 0.2060 | 0.2276 | **0.2757** | 0.2773 | 0.9969 | 0.3317 | **0.3187** | 0.4561 |

Table 15: The impact of different normalization methods in short-term forecasting scenarios (NMAE).

| Model | PatchTST | | | CSDI | | | TimeGrad | | | GRU NVP | | |
|---|---|---|---|---|---|---|---|---|---|---|---|---|
| | ReVIN | Scaling | w/o Norm | ReVIN | Scaling | w/o Norm | ReVIN | Scaling | w/o Norm | ReVIN | Scaling | w/o Norm |
| Electricity-S | 0.0659 | 0.0645 | 0.0660 | 0.0666 | - | **0.0648** | 0.0852 | **0.0710** | 0.9742 | 0.0861 | **0.0929** | 0.2246 |
| Exchange Rate-S | **0.0102** | 0.0108 | 0.0111 | 0.0096 | 0.0111 | 0.0151 | **0.0115** | 0.0118 | 0.0220 | **0.0114** | 0.0189 | 0.0170 |
| Solar-S | **0.6275** | 0.7105 | 0.7169 | 0.5988 | 0.5616 | 0.5680 | **0.6041** | 0.7011 | 0.9162 | 1.1931 | 0.7424 | **0.5893** |
| Traffic-S | **0.2001** | 0.2036 | 0.2168 | 0.1752 | 0.1877 | **0.1669** | 0.2167 | 0.1516 | 0.1693 | 0.2257 | **0.2216** | 0.2837 |
| Wikipedia-S | **0.2529** | 0.3245 | 0.3695 | 0.2585 | 0.2437 | 0.2698 | 0.3278 | **0.3257** | 0.9998 | 0.4041 | **0.3559** | 0.5131 |

### D.1.2 Analysis of Instance-level Normalization

**RevIN Significantly Improves Most Models in Long-term Forecasting Scenarios, with Some Exceptions.** RevIN's ability to mitigate the effects of data distribution shifts leads to significant performance improvements in most models for long-term forecasting, as shown in Table 16. This benefit extends beyond models like PatchTST and iTransformer, which originally employed RevIN, to others such as DLinear that do not inherently use this approach. Notably, RevIN has greatly enhanced AR-based models in long-term scenarios. For instance, on the ETT datasets, GRU NVP (w/ RevIN) outperforms even PatchTST (w/ RevIN), suggesting that normalizing trend effects can help reduce error accumulation in AR-based models.

However, RevIN can have a negative impact in certain cases. On the Traffic dataset, GRU (w/ ReVIN) and GRU NVP (w/ ReVIN) perform worse than without normalization, as shown in Table 16. Interestingly, this aligns with our analysis of data characteristics: the Traffic dataset displays strong seasonality but less trending. We speculate that RevIN's effectiveness in other datasets stems from its ability to normalize trend-related distribution shifts, which is less relevant for the Traffic dataset. Additionally, RevIN appears less suited for NAR probabilistic models. For instance, CSDI (w/ RevIN) performs worse than CSDI (w/ Scaling) on the Weather, Electricity, Exchange, and ILI dataset. Further research is needed to develop more effective normalization strategies for NAR probabilistic models.

**No Dominating Normalization Strategies in Short-term Forecasting.** As shown in Table 14, RevIN does not consistently provide robust or significant improvements for models such as CSDI, TimeGrad, and GRU NVP in short-term forecasting. The Mean Scaling strategy, though empirical, proves to be the most reliable choice for these probabilistic models, likely explaining its widespread use. In some cases, instance-level normalization can be omitted, but this approach can lead to serious issues, as seen with TimeGrad (w/o normalization) on the Wikipedia and Solar datasets, and GRU NVP (w/o normalization) on the Electricity dataset. Developing effective instance-level normalization methods for complex data distributions in short-term forecasting remains an important yet often overlooked research direction.

### D.2 The Impact of Data Scale

To further explore critical characteristics of time-series forecasting, we have examined the correlation between model performance gains, relative to the baseline model (GRU), and dataset dimensions, length, and volume (see Table 18). However, our analysis does not identify a significant correlation between these factors and model performance.

### D.3 Statistical and Gradient Boosting Decision Tree Baselines

To enhance the empirical robustness of our study, we integrate classical statistical models, including ARIMA [44] and ETS [30], along with the Gradient Boosting Decision Tree (GBDT) model, XGBoost, into the `ProbTS` framework. The results in Table 19 clearly demonstrate the superior performance of deep learning methods over simple statistical baselines, emphasizing the importance

Table 16: The impact of different normalization methods in long-term forecasting scenarios (CRPS).

| Dataset | Pred len | iTransformer w/o Norm | iTransformer ReVIN | iTransformer Scaling | DLinear w/o Norm | DLinear ReVIN | DLinear Scaling | PatchTST w/o Norm | PatchTST ReVIN | PatchTST Scaling | CSDI w/o Norm | CSDI ReVIN | CSDI Scaling | TimeGrad w/o Norm | TimeGrad ReVIN | TimeGrad Scaling | GRU NVP w/o Norm | GRU NVP ReVIN | GRU NVP Scaling | GRU w/o Norm | GRU ReVIN | GRU Scaling |
|---|---|---|---|---|---|---|---|---|---|---|---|---|---|---|---|---|---|---|---|---|---|---|
| ETTh1-L | 96 | 0.3514 | **0.3148** | 0.3481 | 0.3613 | **0.3210** | 0.3240 | 0.3447 | **0.3212** | 0.3437 | 0.3671 | **0.2764** | 0.3630 | 0.5434 | **0.2958** | 0.6963 | 0.4444 | **0.2771** | 0.5946 | 0.5090 | **0.4457** | 0.9734 |
|  | 192 | 0.4427 | **0.3479** | 0.3843 | 0.3673 | **0.3574** | 0.3724 | 0.3785 | **0.3562** | 0.3943 | 0.4673 | **0.3553** | 0.4352 | 0.5698 | **0.3119** | 0.6410 | 0.5056 | **0.3076** | 0.4729 | 0.5915 | **0.4763** | 1.0658 |
|  | 336 | 0.4192 | **0.3654** | 0.4001 | **0.3750** | 0.3878 | 0.3806 | 0.4099 | **0.3737** | 0.4213 | 0.4814 | **0.3614** | 0.4192 | 0.6192 | **0.3950** | 0.6586 | 0.4803 | **0.3081** | 0.6240 | 0.5956 | **0.4998** | 1.1416 |
|  | 720 | 0.5172 | **0.3902** | 0.5576 | 0.4446 | **0.4023** | 0.4738 | 0.4648 | **0.3909** | 0.4881 | 0.5609 | **0.3960** | 0.4283 | 0.5837 | **0.3435** | 0.8670 | 0.5862 | **0.3641** | 0.7295 | 0.7265 | **0.5901** | 1.2715 |
| ETTh2-L | 96 | 0.2790 | **0.1796** | 0.2099 | 0.2034 | 0.2179 | **0.2031** | 0.1892 | **0.1763** | 0.1900 | 0.1681 | **0.1446** | 0.1590 | 0.6859 | **0.1738** | 0.3658 | 0.4061 | **0.1866** | 0.4399 | 0.8732 | **0.2913** | 0.5277 |
|  | 192 | 0.3322 | **0.2044** | 0.2346 | 0.2700 | **0.2449** | 0.2572 | 0.2189 | **0.2012** | 0.2167 | 0.2078 | **0.1734** | 0.1983 | 0.6811 | **0.1905** | 0.4806 | 0.4510 | **0.1894** | 0.5601 | 0.9771 | **0.3334** | 0.5329 |
|  | 336 | 0.4475 | **0.2209** | 0.2726 | 0.2617 | **0.2339** | 0.2604 | 0.2446 | **0.2244** | 0.2446 | 0.2728 | **0.2094** | 0.2124 | 0.6393 | **0.2213** | 0.4178 | 0.6722 | **0.2379** | 0.4289 | 0.7353 | **0.3658** | 0.5879 |
|  | 720 | 0.4315 | **0.2329** | 0.3148 | 0.3009 | **0.2768** | 0.2942 | 0.2819 | **0.2301** | 0.2829 | 0.3062 | **0.2054** | 0.2512 | 0.4990 | **0.2198** | 0.3736 | 0.9457 | **0.2560** | 0.4629 | 1.0921 | **0.3594** | 0.6519 |
| ETTm1-L | 96 | 0.3222 | **0.2845** | 0.2890 | **0.2662** | 0.2674 | 0.2699 | 0.2989 | **0.2649** | 0.2858 | 0.2569 | **0.2255** | 0.2597 | 0.6828 | **0.2886** | 0.4473 | 0.3964 | **0.3132** | 0.5557 | 0.4433 | **0.4411** | 0.9213 |
|  | 192 | 0.3446 | **0.3040** | 0.3122 | 0.2914 | **0.2911** | 0.3007 | 0.3349 | **0.2925** | 0.3077 | **0.3280** | 0.3313 | 0.3495 | 0.7930 | **0.2975** | 0.5341 | 0.5106 | **0.3313** | 0.5342 | 0.6205 | **0.4824** | 0.9978 |
|  | 336 | 0.3738 | **0.3276** | 0.3303 | 0.3180 | **0.3118** | 0.3215 | 0.3618 | **0.3107** | 0.3338 | 0.3663 | **0.3243** | 0.4120 | 0.6675 | **0.3201** | 0.5657 | 0.4604 | **0.3296** | 0.5614 | 0.6186 | **0.4879** | 1.0089 |
|  | 720 | 0.4248 | **0.3660** | 0.3831 | 0.3508 | **0.3468** | 0.3683 | 0.3790 | **0.3468** | 0.3780 | 0.3933 | **0.3789** | 0.4367 | 0.8248 | **0.3357** | 0.5826 | 0.4383 | **0.3520** | 0.5568 | 0.7808 | **0.4941** | 1.0310 |
| ETTm2-L | 96 | 0.1505 | **0.1381** | 0.1528 | 0.1438 | **0.1390** | 0.1436 | 0.1522 | **0.1335** | 0.1587 | 0.1314 | **0.1134** | 0.1340 | 0.4493 | **0.1206** | 0.2376 | 0.4107 | **0.1150** | 0.4053 | 0.4133 | **0.1802** | 0.4339 |
|  | 192 | 0.1833 | **0.1674** | 0.1750 | 0.1645 | **0.1625** | 0.1642 | 0.1884 | **0.1601** | 0.1940 | 0.1391 | **0.1386** | 0.1583 | 0.3959 | **0.1399** | 0.3469 | 0.4690 | **0.1363** | 0.4170 | 0.5699 | **0.2074** | 0.4159 |
|  | 336 | 0.2628 | **0.1929** | 0.2122 | 0.1933 | **0.1828** | 0.1939 | 0.1994 | **0.1798** | 0.2249 | 0.2139 | **0.1592** | 0.1782 | 0.4661 | **0.1561** | 0.3187 | 0.4163 | **0.1571** | 0.4050 | 0.7246 | **0.2216** | 0.4764 |
|  | 720 | 0.3098 | **0.2142** | 0.2278 | 0.2184 | **0.2094** | 0.2183 | 0.2940 | **0.2063** | 0.2905 | 0.2218 | **0.1892** | 0.2002 | 0.4260 | **0.1822** | 0.2896 | 0.3968 | **0.1801** | 0.5590 | 0.6574 | **0.3022** | 0.5101 |
| Electricity-L | 96 | 0.0887 | **0.0827** | 0.0845 | 0.0872 | **0.0862** | 0.0871 | **0.0848** | 0.0857 | 0.0867 | **0.0735** | 0.0761 | 0.0735 | 0.0961 | **0.0771** | 0.0904 | 0.0923 | **0.0805** | 0.0882 | **0.1218** | 0.2261 | 0.1462 |
|  | 192 | 0.0911 | **0.0892** | 0.0906 | 0.0977 | **0.0931** | 0.0934 | **0.0910** | 0.0912 | 0.0918 | **0.2475** | 0.2749 | 0.2584 | 0.0980 | **0.0806** | 0.0936 | 0.0935 | **0.0833** | 0.0945 | **0.1299** | 0.2381 | 0.1851 |
|  | 336 | 0.1026 | **0.0984** | 0.1024 | 0.1020 | **0.1018** | 0.1019 | 0.1006 | **0.1001** | 0.1005 | 0.3136 | **0.2153** | 0.2835 | 0.1125 | **0.0905** | 0.1002 | 0.0982 | **0.0925** | 0.0932 | **0.1364** | 0.2807 | 0.2405 |
|  | 720 | 0.1148 | **0.1110** | 0.1123 | 0.1193 | **0.1170** | 0.1192 | 0.1178 | **0.1160** | 0.1187 | **0.2667** | 25.7527 | 0.2945 | 0.1096 | 0.1164 | **0.0957** | 0.1065 | 0.1107 | **0.1036** | **0.1455** | 0.3228 | 0.2340 |
| Exchange-L | 96 | 0.0461 | **0.0251** | 0.0318 | 0.0244 | **0.0240** | 0.0243 | 0.0299 | **0.0235** | 0.0254 | 0.0343 | 0.0216 | **0.0210** | 0.0837 | **0.0279** | 0.0478 | 0.0605 | **0.0246** | 0.0559 | 0.0737 | **0.0385** | 0.1539 |
|  | 192 | 0.0694 | **0.0345** | 0.0425 | 0.0343 | 0.0341 | **0.0338** | 0.0404 | **0.0336** | 0.0365 | 0.0418 | **0.0383** | 0.0388 | 0.0638 | **0.0364** | 0.0743 | 0.0860 | **0.0340** | 0.0814 | 0.0858 | **0.0486** | 0.1704 |
|  | 336 | 0.0843 | **0.0460** | 0.0572 | **0.0450** | 0.0472 | 0.0451 | 0.0597 | **0.0462** | 0.0532 | 0.0510 | 0.0508 | **0.0485** | 0.1030 | **0.0510** | 0.1142 | 0.0637 | **0.0475** | 0.0948 | 0.0778 | **0.0630** | 0.2240 |
|  | 720 | 0.1317 | **0.0769** | 0.1145 | **0.0704** | 0.0777 | 0.0830 | 0.0824 | **0.0777** | 0.0810 | 0.0964 | **0.0881** | 0.1042 | 0.1177 | **0.0800** | 0.1428 | 0.0724 | **0.0644** | 0.1257 | 0.0987 | **0.0955** | 0.3978 |
| ILL-L | 24 | 0.3792 | 0.1093 | **0.0971** | 0.1920 | **0.1183** | 0.2138 | 0.2665 | **0.0794** | 0.1478 | 0.2541 | 0.1344 | **0.1182** | 0.2776 | **0.0815** | 0.0920 | 0.2823 | **0.0720** | 0.1312 | 0.2845 | **0.1819** | 0.3277 |
|  | 36 | 0.3453 | **0.1422** | 0.1632 | 0.1864 | **0.1551** | 0.2024 | 0.2955 | **0.1239** | 0.1422 | 0.2807 | **0.1214** | 0.1526 | 0.3164 | **0.0962** | 0.1524 | 0.2806 | **0.0983** | 0.1403 | 0.3568 | **0.1771** | 0.4417 |
|  | 48 | 0.3349 | **0.1617** | 0.1653 | 0.2021 | **0.1732** | 0.2059 | 0.2957 | **0.1312** | 0.1704 | 0.3383 | **0.1026** | 0.1453 | 0.3229 | **0.1124** | 0.2275 | 0.2908 | **0.1193** | 0.1499 | 0.6083 | **0.1893** | 0.4355 |
|  | 60 | 0.3365 | **0.1631** | 0.2148 | 0.2389 | **0.1678** | 0.2321 | 0.3034 | **0.1493** | 0.1964 | 0.2651 | **0.1333** | 0.1496 | 0.3162 | **0.1250** | 0.1484 | 0.3254 | **0.1204** | 0.1817 | 0.4341 | **0.1654** | 0.4491 |
| Traffic-L | 96 | 0.3495 | 0.2379 | **0.2377** | 0.3528 | 0.3581 | 0.3546 | **0.2526** | 0.2553 | 0.3546 | - | - | **0.2546** | 0.2094 | 0.2309 | **0.2033** | **0.1952** | 0.2021 | 0.2134 | **0.2693** | 0.3070 | 0.3025 |
|  | 192 | 0.3821 | 0.2434 | **0.2394** | 0.3345 | **0.3307** | 0.3366 | 0.2483 | **0.2479** | 0.3366 | - | - | **0.2447** | **0.2079** | 0.2110 | 0.2091 | **0.2005** | 0.2110 | 0.2034 | **0.2647** | 0.3189 | 0.3065 |
|  | 336 | 0.3749 | 0.2503 | **0.2467** | 0.3319 | **0.3294** | 0.3316 | **0.2489** | 0.2492 | 0.3316 | - | - | **0.2470** | 0.2271 | 0.2306 | **0.2091** | **0.1983** | 0.2220 | 0.2143 | **0.2783** | 0.3313 | 0.3405 |
|  | 720 | 0.3940 | 0.2597 | **0.2580** | 0.3672 | **0.3337** | 0.3671 | 0.2577 | **0.2572** | 0.3671 | - | - | **0.2568** | 0.2302 | 0.2659 | **0.2184** | **0.2198** | 0.2524 | 0.2222 | **0.2994** | 0.4225 | 0.4079 |
| Weather-L | 96 | 0.1083 | **0.0930** | 0.1084 | 0.1107 | **0.0959** | 0.1092 | 0.1051 | **0.0837** | 0.1082 | 0.0591 | 0.1202 | **0.0575** | 0.2939 | **0.0737** | 0.2308 | 0.1005 | **0.0837** | 0.1546 | 0.4758 | **0.1166** | 0.2710 |
|  | 192 | 0.1160 | **0.0940** | 0.1093 | 0.1227 | **0.0984** | 0.1209 | 0.1136 | **0.0858** | 0.1171 | 0.0720 | 0.1599 | **0.0717** | 0.3138 | **0.0788** | 0.2495 | 0.1177 | **0.0866** | 0.1342 | 0.4587 | **0.1239** | 0.3160 |
|  | 336 | 0.1131 | **0.0932** | 0.1208 | 0.1317 | **0.1009** | 0.1296 | 0.1118 | **0.0903** | 0.1167 | 0.0977 | 0.4296 | **0.0856** | 0.2249 | **0.0860** | 0.2646 | 0.1145 | **0.0868** | 0.1715 | 0.4651 | **0.1247** | 0.3196 |
|  | 720 | 0.1277 | **0.1009** | 0.1212 | 0.1442 | **0.1050** | 0.1425 | 0.1117 | **0.0953** | 0.1234 | **0.0975** | 0.1785 | 0.1419 | 0.2069 | **0.0941** | 0.1973 | 0.1979 | **0.0925** | 0.1146 | 0.4580 | **0.1624** | 0.3297 |

Table 17: The impact of different normalization methods in long-term forecasting scenarios (NMAE).

| Dataset | Pred len | iTransformer w/o Norm | iTransformer ReVIN | iTransformer Scaling | DLinear w/o Norm | DLinear ReVIN | DLinear Scaling | PatchTST w/o Norm | PatchTST ReVIN | PatchTST Scaling | CSDI w/o Norm | CSDI ReVIN | CSDI Scaling | TimeGrad w/o Norm | TimeGrad ReVIN | TimeGrad Scaling | GRU NVP w/o Norm | GRU NVP ReVIN | GRU NVP Scaling | GRU w/o Norm | GRU ReVIN | GRU Scaling |
|---|---|---|---|---|---|---|---|---|---|---|---|---|---|---|---|---|---|---|---|---|---|---|
| ETTh1-L | 96 | 0.3514 | **0.3148** | 0.3481 | 0.3613 | **0.3210** | 0.3240 | 0.3447 | **0.3212** | 0.3437 | 0.4718 | **0.3643** | 0.4605 | 0.7105 | **0.3783** | 0.8983 | 0.5452 | **0.3569** | 0.7760 | 0.5090 | **0.4457** | 0.9734 |
| | 192 | 0.4427 | **0.3479** | 0.3843 | 0.3673 | 0.3574 | 0.3724 | 0.3785 | **0.3562** | 0.3943 | 0.5839 | **0.4445** | 0.5683 | 0.7218 | **0.3926** | 0.8034 | 0.6226 | **0.3946** | 0.5698 | 0.5915 | **0.4763** | 1.0658 |
| | 336 | 0.4192 | **0.3654** | 0.4001 | 0.3750 | 0.3690 | 0.3806 | 0.4099 | **0.3737** | 0.4213 | 0.6081 | **0.4524** | 0.5321 | 0.7928 | **0.4942** | 0.8381 | 0.5773 | **0.3933** | 0.7277 | 0.5956 | **0.4998** | 1.1416 |
| | 720 | 0.5172 | **0.3902** | 0.5576 | 0.4446 | 0.3878 | 0.4738 | 0.4648 | **0.3909** | 0.4881 | 0.7027 | **0.5118** | 0.5351 | 0.7552 | **0.4332** | 1.0233 | 0.7305 | **0.4409** | 0.9122 | 0.7265 | **0.5901** | 1.2715 |
| ETTh2-L | 96 | 0.2790 | **0.1796** | 0.2099 | 0.2034 | 0.2179 | **0.2031** | 0.1892 | **0.1763** | 0.1900 | 0.2139 | **0.1879** | 0.2030 | 0.8161 | **0.2080** | 0.4498 | 0.4846 | **0.2224** | 0.5214 | 0.8732 | **0.2913** | 0.5277 |
| | 192 | 0.3322 | **0.2044** | 0.2346 | 0.2700 | **0.2449** | 0.2572 | 0.2189 | 0.2012 | 0.2167 | 0.2683 | **0.2234** | 0.2550 | 0.8217 | **0.2323** | 0.5783 | 0.5304 | **0.2282** | 0.7003 | 0.9771 | **0.3334** | 0.5329 |
| | 336 | 0.4475 | **0.2209** | 0.2726 | 0.2617 | 0.2339 | 0.2604 | 0.2446 | 0.2244 | 0.2446 | 0.3443 | **0.2671** | 0.2725 | 0.7171 | **0.2657** | 0.5096 | 0.8343 | **0.2789** | 0.5543 | 0.7353 | **0.3658** | 0.5879 |
| | 720 | 0.4315 | **0.2329** | 0.3148 | 0.3009 | 0.2768 | 0.2942 | 0.2819 | 0.2301 | 0.2829 | 0.3792 | **0.2688** | 0.3303 | 0.6229 | **0.2589** | 0.4772 | 1.0349 | **0.2886** | 0.5579 | 1.0921 | **0.3594** | 0.6519 |
| ETTm1-L | 96 | 0.3222 | **0.2845** | 0.2890 | **0.2662** | 0.2674 | 0.2699 | 0.2989 | **0.2649** | 0.2858 | 0.3351 | **0.2879** | 0.3228 | 0.7941 | **0.3647** | 0.5763 | 0.4806 | **0.3848** | 0.7064 | 0.4433 | **0.4411** | 0.9213 |
| | 192 | 0.3446 | **0.3040** | 0.3122 | 0.2914 | **0.2911** | 0.3007 | 0.3349 | **0.2925** | 0.3077 | 0.4312 | **0.4235** | 0.4310 | 0.9226 | **0.3680** | 0.7200 | 0.6149 | **0.4158** | 0.6791 | 0.6205 | **0.4824** | 0.9978 |
| | 336 | 0.3738 | **0.3276** | 0.3303 | 0.3180 | **0.3118** | 0.3215 | 0.3618 | 0.3107 | 0.3338 | 0.4691 | **0.3978** | 0.5297 | 0.7887 | **0.3904** | 0.7476 | 0.5751 | **0.4171** | 0.7054 | 0.6186 | **0.4879** | 1.0089 |
| | 720 | 0.4248 | **0.3660** | 0.3831 | 0.3508 | **0.3468** | 0.3683 | 0.3790 | **0.3468** | 0.3780 | 0.4876 | **0.4748** | 0.5148 | 0.9607 | **0.4063** | 0.7444 | 0.5375 | **0.4452** | 0.7194 | 0.7808 | **0.4941** | 1.0310 |
| ETTm2-L | 96 | 0.1505 | **0.1381** | 0.1528 | 0.1438 | **0.1390** | 0.1436 | 0.1522 | **0.1335** | 0.1587 | 0.1681 | **0.1422** | 0.1713 | 0.5247 | **0.1513** | 0.2911 | 0.5324 | **0.1454** | 0.4928 | 0.4133 | **0.1802** | 0.4339 |
| | 192 | 0.1833 | **0.1674** | 0.1750 | 0.1645 | **0.1625** | 0.1642 | 0.1884 | **0.1601** | 0.1940 | 0.1787 | **0.1747** | 0.2011 | 0.4909 | **0.1742** | 0.4467 | 0.5476 | **0.1692** | 0.5217 | 0.5699 | **0.2074** | 0.4159 |
| | 336 | 0.2628 | **0.1929** | 0.2122 | 0.1933 | **0.1828** | 0.1939 | 0.1994 | **0.1798** | 0.2249 | 0.2803 | **0.2010** | 0.2262 | 0.5636 | **0.1879** | 0.3843 | 0.5435 | **0.1939** | 0.5221 | 0.7246 | **0.2216** | 0.4764 |
| | 720 | 0.3098 | **0.2142** | 0.2278 | 0.2184 | **0.2094** | 0.2183 | 0.2940 | **0.2063** | 0.2905 | 0.2879 | **0.2340** | 0.2546 | 0.5060 | **0.2150** | 0.3588 | 0.4969 | **0.2200** | 0.7256 | 0.6574 | **0.3022** | 0.5101 |
| Electricity-L | 96 | 0.0887 | **0.0827** | 0.0845 | 0.0872 | **0.0862** | 0.0871 | **0.0848** | 0.0857 | 0.0867 | 0.0924 | 0.0975 | **0.0910** | 0.1186 | **0.0975** | 0.1148 | 0.1141 | **0.1007** | 0.1133 | **0.1218** | 0.2261 | 0.1462 |
| | 192 | 0.0911 | **0.0892** | 0.0906 | 0.0977 | **0.0931** | 0.0934 | **0.0910** | 0.0912 | 0.0918 | **0.3218** | 0.3788 | 0.3380 | 0.1243 | **0.1013** | 0.1201 | 0.1170 | **0.1044** | 0.1231 | **0.1299** | 0.2381 | 0.1851 |
| | 336 | 0.1026 | **0.0984** | 0.1024 | 0.1020 | **0.1018** | 0.1019 | 0.1006 | **0.1001** | 0.1005 | 0.4213 | **0.2776** | 0.3689 | 0.1423 | **0.1134** | 0.1279 | 0.1225 | **0.1174** | 0.1208 | **0.1364** | 0.2807 | 0.2405 |
| | 720 | 0.1148 | **0.1110** | 0.1123 | 0.1193 | **0.1170** | 0.1192 | 0.1178 | **0.1160** | 0.1187 | **0.3726** | 16.3914 | 0.3873 | 0.1350 | 0.1424 | **0.1220** | 0.1339 | 0.1413 | **0.1337** | **0.1455** | 0.3228 | 0.2340 |
| Exchange-L | 96 | 0.0461 | **0.0251** | 0.0318 | 0.0244 | **0.0240** | 0.0243 | 0.0299 | **0.0235** | 0.0254 | 0.0444 | 0.0275 | **0.0272** | 0.0920 | **0.0320** | 0.0617 | 0.0780 | **0.0282** | 0.0662 | 0.0737 | **0.0385** | 0.1539 |
| | 192 | 0.0694 | **0.0345** | 0.0425 | 0.0343 | 0.0341 | **0.0338** | 0.0404 | **0.0336** | 0.0365 | 0.0545 | 0.0475 | **0.0465** | 0.0714 | **0.0403** | 0.0909 | 0.1097 | **0.0382** | 0.1117 | 0.0858 | **0.0486** | 0.1704 |
| | 336 | 0.0843 | **0.0460** | 0.0572 | 0.0450 | 0.0472 | 0.0451 | 0.0597 | **0.0462** | 0.0532 | 0.0671 | 0.0654 | **0.0543** | 0.1117 | **0.0554** | 0.1392 | 0.0834 | **0.0526** | 0.1252 | 0.0778 | **0.0630** | 0.2240 |
| | 720 | 0.1317 | **0.0769** | 0.1145 | 0.0704 | 0.0777 | 0.0830 | 0.0824 | **0.0777** | 0.0810 | 0.1223 | **0.1112** | 0.1198 | 0.1328 | **0.0833** | 0.1809 | 0.0891 | **0.0722** | 0.1648 | 0.0987 | **0.0955** | 0.3978 |
| ILL-L | 24 | 0.3792 | **0.1093** | 0.0971 | 0.1920 | **0.1183** | 0.2138 | 0.2665 | **0.0794** | 0.1478 | 0.2718 | 0.1550 | **0.1436** | 0.2964 | **0.0955** | 0.1165 | 0.3129 | **0.0866** | 0.1787 | 0.2845 | **0.1819** | 0.3277 |
| | 36 | 0.3453 | **0.1422** | 0.1632 | 0.1864 | **0.1551** | 0.2024 | 0.2955 | **0.1239** | 0.1422 | 0.2966 | **0.1461** | 0.1830 | 0.3386 | **0.1082** | 0.1905 | 0.3115 | **0.1162** | 0.1937 | 0.3568 | **0.1771** | 0.4417 |
| | 48 | 0.3349 | **0.1617** | 0.1653 | 0.2021 | **0.1732** | 0.2059 | 0.2957 | **0.1312** | 0.1704 | 0.3601 | **0.1290** | 0.1687 | 0.3463 | 0.1252 | 0.2650 | 0.3165 | **0.1364** | 0.1989 | 0.6083 | **0.1893** | 0.4355 |
| | 60 | 0.3365 | **0.1631** | 0.2148 | 0.2389 | **0.1678** | 0.2321 | 0.3034 | **0.1493** | 0.1964 | 0.2840 | **0.1607** | 0.1738 | 0.3300 | **0.1409** | 0.1734 | 0.3599 | **0.1427** | 0.2397 | 0.4341 | **0.1654** | 0.4491 |
| Traffic-L | 96 | 0.3495 | **0.2379** | 0.2377 | **0.3528** | 0.3581 | 0.3546 | **0.2526** | 0.2553 | 0.2546 | - | - | - | 0.2436 | 0.2791 | **0.2414** | 0.2422 | 0.2517 | 0.2647 | **0.2693** | 0.3070 | 0.3025 |
| | 192 | 0.3821 | **0.2434** | 0.2394 | 0.3345 | **0.3307** | 0.3366 | 0.2483 | 0.2479 | **0.2447** | - | - | - | **0.2432** | 0.2521 | 0.2461 | **0.2484** | 0.2623 | 0.2534 | **0.2647** | 0.3189 | 0.3065 |
| | 336 | 0.3749 | **0.2503** | 0.2467 | 0.3319 | **0.3294** | 0.3316 | 0.2489 | 0.2492 | **0.2470** | - | - | - | 0.2672 | 0.2739 | **0.2471** | **0.2472** | 0.2772 | 0.2691 | **0.2783** | 0.3313 | 0.3405 |
| | 720 | 0.3940 | **0.2597** | 0.2580 | 0.3672 | **0.3337** | 0.3671 | 0.2577 | 0.2572 | **0.2568** | - | - | - | 0.2711 | 0.3125 | **0.2620** | **0.2742** | 0.3142 | 0.2815 | **0.2994** | 0.4225 | 0.4079 |
| Weather-L | 96 | 0.1083 | **0.0930** | 0.1084 | 0.1107 | **0.0959** | 0.1092 | 0.1051 | **0.0837** | 0.1082 | 0.0746 | 0.1280 | **0.0710** | 0.3731 | **0.0891** | 0.2877 | 0.1225 | **0.1045** | 0.2055 | 0.4758 | **0.1166** | 0.2710 |
| | 192 | 0.1160 | **0.0940** | 0.1093 | 0.1227 | **0.0984** | 0.1209 | 0.1136 | **0.0858** | 0.1171 | 0.0891 | 0.2102 | **0.0857** | 0.3787 | **0.0938** | 0.3222 | 0.1493 | **0.1069** | 0.1706 | 0.4587 | **0.1239** | 0.3160 |
| | 336 | 0.1131 | **0.0932** | 0.1208 | 0.1317 | **0.1009** | 0.1296 | 0.1118 | **0.0903** | 0.1167 | 0.1143 | 0.5821 | **0.1023** | 0.2737 | **0.1041** | 0.3432 | 0.1367 | **0.1030** | 0.2285 | 0.4651 | **0.1247** | 0.3196 |
| | 720 | 0.1277 | **0.1009** | 0.1212 | 0.1442 | **0.1050** | 0.1425 | 0.1117 | **0.0953** | 0.1234 | **0.1167** | 0.2316 | 0.1739 | 0.2540 | **0.1162** | 0.2637 | 0.2480 | **0.1075** | 0.1458 | 0.4580 | **0.1624** | 0.3297 |

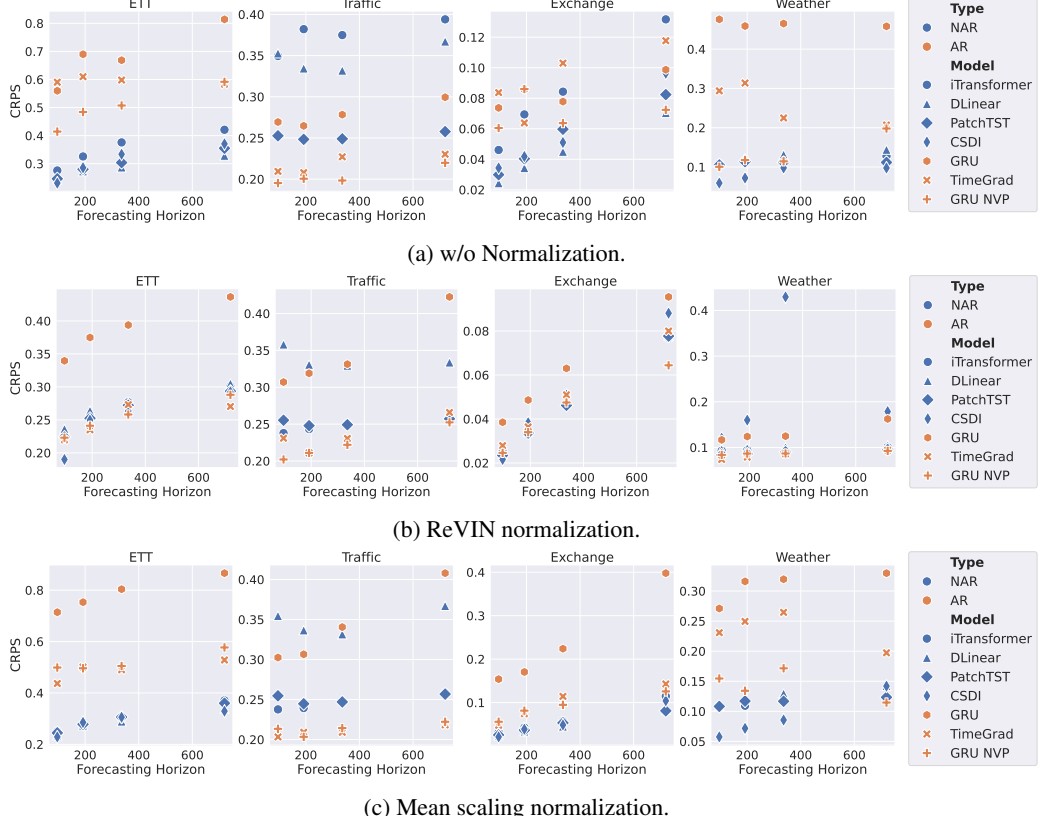

(a) w/o Normalization.

(b) ReVIN normalization.

(c) Mean scaling normalization.

Figure 7: Impact of different instance-level normalization methods on model performance.

Table 18: The correlation coefficient between the data volume and the relative performance improvement compared to the baseline model (GRU).

| Model | DLinear | | PatchTST | | GRU NVP | | TimeGrad | | CSDI | |
| --- | --- | --- | --- | --- | --- | --- | --- | --- | --- | --- |
| | CRPS | NMAE | CRPS | NMAE | CRPS | NMAE | CRPS | NMAE | CRPS | NMAE |
| # Var. | 0.2422 | 0.2422 | -0.2676 | -0.2676 | -0.1856 | -0.2136 | -0.1665 | -0.1793 | -0.2315 | -0.2592 |
| # Total timestep | -0.1422 | -0.1422 | 0.3821 | 0.3821 | 0.3072 | 0.3329 | 0.2860 | 0.2971 | 0.3542 | 0.3826 |
| # Var. × Timestep | 0.0162 | 0.0162 | 0.0166 | 0.0166 | -0.0068 | -0.0011 | 0.0082 | 0.0117 | -0.0053 | -0.0133 |

of capturing non-linear dependencies for accurate forecasts. Notably, ARIMA and ETS exhibit varied performance across different data characteristics. ARIMA struggles with datasets like Solar, characterized by weak trending and strong seasonality, while ETS shows better adaptability. Conversely, in cases of strong trending and weak seasonality, as observed in the 'Wikipedia' dataset, ARIMA significantly outperforms ETS.

Utilizing the implementation from [21], we find that XGBoost competes well, even surpassing neural network models in certain scenarios. However, for datasets with more complex distributions like 'Solar' and 'Electricity,' advanced probabilistic estimation methods demonstrate a substantial advantage over traditional learning methods and point estimation techniques. This highlights the adaptability and strength of advanced probabilistic methods in handling intricate forecasting scenarios.

### D.4 Experiments on Univariate Datasets

In pursuit of a comprehensive analysis spanning univariate and multivariate scenarios, we examined a subset of M4 [45], M5 [46], and TOURISM datasets [3]—crucial datasets for univariate time-series forecasting. Table 20 provides a quantitative assessment of the intrinsic characteristics of these new datasets, focusing on trending strength, seasonality, and data distribution complexity, as detailed in

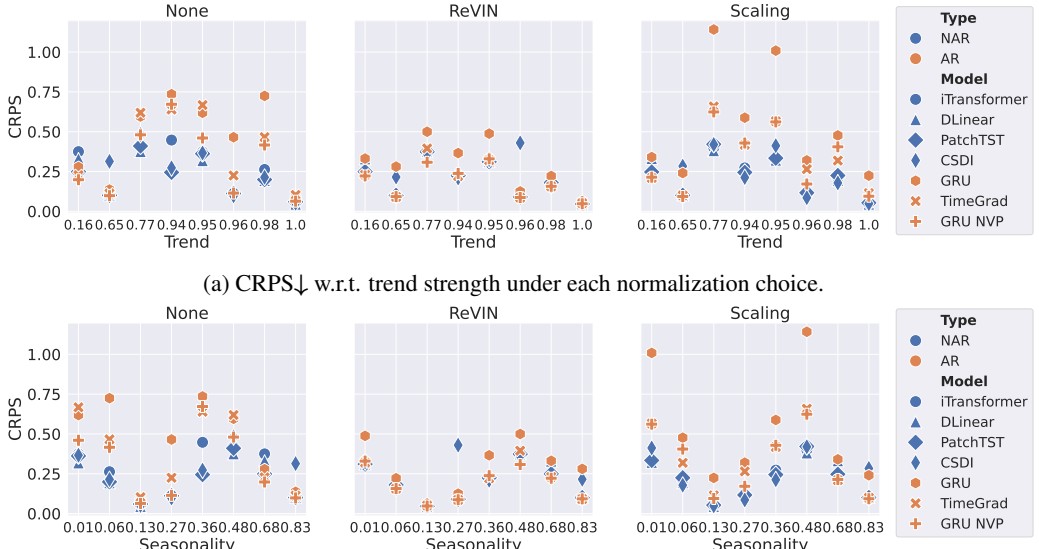

(a) CRPS↓ w.r.t. trend strength under each normalization choice.

(b) CRPS↓ w.r.t. seasonality strength under each normalization choice.

Figure 8: Impact of data characteristics on the effectiveness of different instance-level normalization strategies.

Table 19: Results of statistical models and GBDT baseline on short-term forecasting datasets.

| Model | Exchange-S | | Solar-S | | Electricity-S | | Traffic-S | | Wikipedia-S | |
|---|---|---|---|---|---|---|---|---|---|---|
| | CRPS | NMAE | CRPS | NMAE | CRPS | NMAE | CRPS | NMAE | CRPS | NMAE |
| ARIMA | 0.009 | 0.009 | 1.000 | 1.000 | 0.164 | 0.164 | 0.461 | 0.461 | 0.348 | 0.348 |
| ETS | 0.011 | 0.011 | 0.580 | 0.580 | 0.121 | 0.121 | 0.413 | 0.413 | 0.685 | 0.685 |
| ETS-prob | 0.008 | 0.011 | 0.795 | 0.695 | 0.123 | 0.129 | 0.380 | 0.433 | 0.625 | 0.697 |
| XGBoost | 0.011 | 0.011 | 0.599 | 0.599 | 0.074 | 0.074 | 0.196 | 0.196 | - | - |
| DLinear | $0.012_{.001}$ | $0.012_{.001}$ | $0.547_{.009}$ | $0.547_{.009}$ | $0.095_{.006}$ | $0.095_{.006}$ | $0.273_{.012}$ | $0.273_{.012}$ | $1.046_{.037}$ | $1.046_{.037}$ |
| PatchTST | $\underline{0.010}_{.000}$ | $\mathbf{0.010}_{.000}$ | $0.496_{.002}$ | $0.496_{.002}$ | $0.076_{.001}$ | $0.076_{.001}$ | $0.202_{.001}$ | $0.202_{.001}$ | $\underline{0.257}_{.001}$ | $\mathbf{0.257}_{.001}$ |
| TimesNet | $0.011_{.001}$ | $0.011_{.001}$ | $0.507_{.019}$ | $0.507_{.019}$ | $0.071_{.002}$ | $0.071_{.002}$ | $0.205_{.002}$ | $0.205_{.002}$ | $0.304_{.002}$ | $0.304_{.002}$ |
| GRU NVP | $0.016_{.003}$ | $0.020_{.003}$ | $0.396_{.021}$ | $0.507_{.022}$ | $0.055_{.002}$ | $0.073_{.003}$ | $0.161_{.006}$ | $0.203_{.009}$ | $0.282_{.003}$ | $0.330_{.003}$ |
| GRU MAF | $0.015_{.001}$ | $0.020_{.001}$ | $0.386_{.026}$ | $0.492_{.027}$ | $\underline{0.051}_{.001}$ | $0.067_{.001}$ | $0.131_{.006}$ | $0.165_{.009}$ | $0.281_{.004}$ | $0.337_{.005}$ |
| Trans MAF | $0.011_{.001}$ | $0.014_{.001}$ | $0.400_{.022}$ | $0.503_{.022}$ | $0.054_{.004}$ | $0.071_{.005}$ | $\mathbf{0.129}_{.004}$ | $\mathbf{0.165}_{.006}$ | $0.289_{.008}$ | $0.344_{.008}$ |
| TimeGrad | $0.011_{.001}$ | $0.014_{.002}$ | $\mathbf{0.359}_{.011}$ | $\mathbf{0.445}_{.023}$ | $0.052_{.001}$ | $\underline{0.067}_{.001}$ | $0.164_{.091}$ | $0.201_{.115}$ | $0.272_{.008}$ | $0.327_{.011}$ |
| CSDI | $\mathbf{0.008}_{.000}$ | $\underline{0.011}_{.000}$ | $0.366_{.005}$ | $\underline{0.484}_{.008}$ | $\mathbf{0.050}_{.001}$ | $\mathbf{0.065}_{.001}$ | $0.146_{.012}$ | $0.176_{.013}$ | $\mathbf{0.219}_{.006}$ | $\underline{0.259}_{.009}$ |

our paper. Notably, these datasets, except for M4-Daily may exhibit fewer seasonal patterns, do not introduce particularly unique characteristics.

Table 20: Quantitative assessment of the intrinsic characteristics of the univariate datasets. The JS Div. denotes Jensen–Shannon divergence, where a lower score indicates closer approximations to a Gaussian distribution.

| Dataset | M4-Weekly | M4-Daily | M5 | TOURISM-Monthly |
|---|---|---|---|---|
| Trend $F_T$ | 0.7677 | 0.9808 | 0.3443 | 0.7979 |
| Seasonality $F_S$ | 0.3401 | 0.0467 | 0.2480 | 0.6826 |
| JS Div. | 0.5106 | 0.4916 | 0.6011 | 0.3291 |

Table 21 presents experimental results for representative methods, consistent with our initial observations. Probabilistic estimation methods like GRU NVP and TimeGrad excel on datasets with complex distributions (e.g., M4-Weekly and M5), while simpler point forecasting methods such as DLinear and PatchTST perform well on datasets with relatively simple data distribution, like TOURISM-Monthly. Both autoregressive and non-autoregressive decoding schemes show comparable performance in short-term forecasting, as discussed in the main paper.

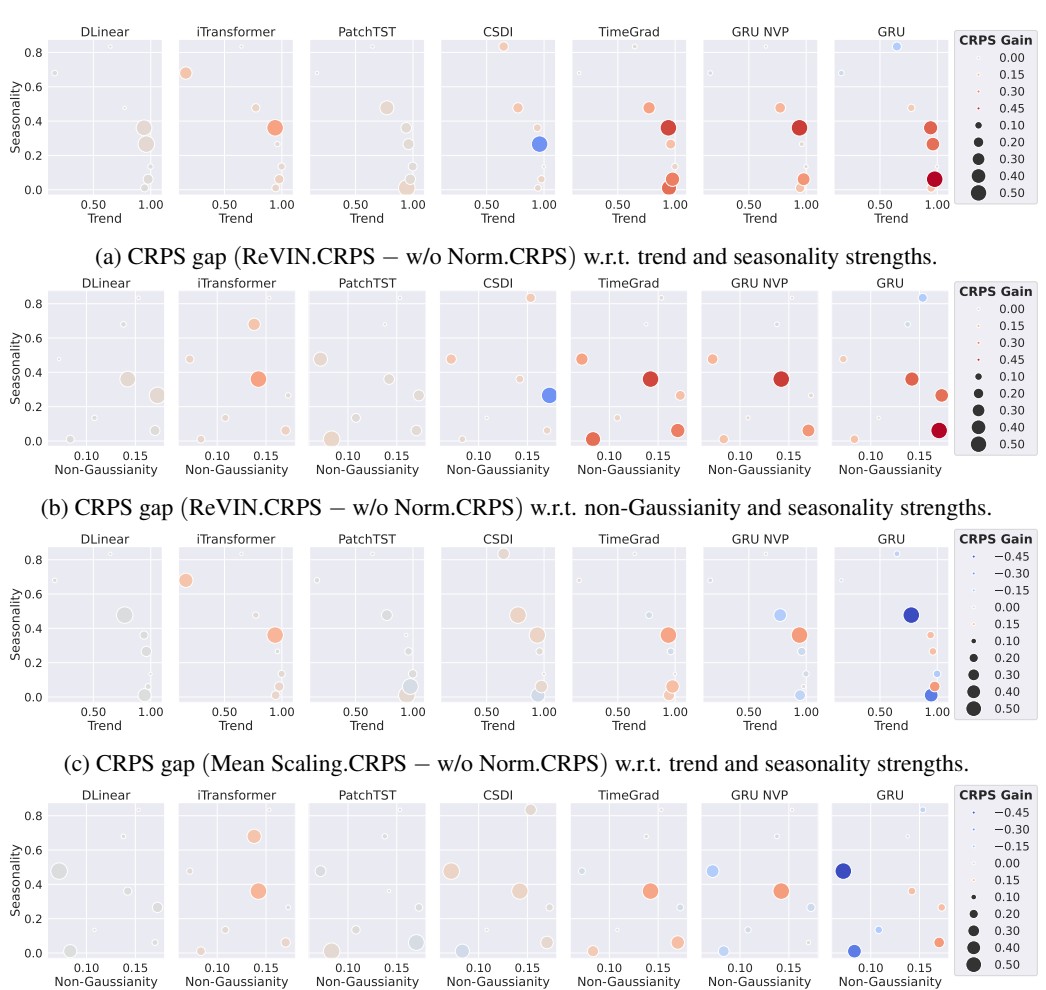

(a) CRPS gap (ReVIN.CRPS − w/o Norm.CRPS) w.r.t. trend and seasonality strengths.

(b) CRPS gap (ReVIN.CRPS − w/o Norm.CRPS) w.r.t. non-Gaussianity and seasonality strengths.

(c) CRPS gap (Mean Scaling.CRPS − w/o Norm.CRPS) w.r.t. trend and seasonality strengths.

(d) CRPS gap (Mean Scaling.CRPS − w/o Norm.CRPS) w.r.t. non-Gaussianity and seasonality strengths.

Figure 9: Impact of different instance-level normalization methods on model performance.

Table 21: Results on M4, M5, and TOURISM datasets. We utilize a lookback window of 3H, with 'H' denoting the forecasting horizon.

| Dataset | DLinear | | PatchTST | | GRU NVP | | TimeGrad | |
|---|---|---|---|---|---|---|---|---|
| | CRPS | NMAE | CRPS | NMAE | CRPS | NMAE | CRPS | NMAE |
| M4-Weekly | 0.081 | 0.081 | 0.089 | 0.089 | 0.066 | 0.077 | 0.055 | 0.065 |
| M4-Daily | 0.034 | 0.034 | 0.035 | 0.035 | 0.030 | 0.038 | 0.026 | 0.032 |
| M5 | 0.891 | 0.891 | 0.898 | 0.898 | 0.679 | 0.864 | - | - |
| TOURISM-Monthly | 0.168 | 0.168 | 0.136 | 0.136 | 0.171 | 0.223 | 0.152 | 0.191 |

## D.5 Experiments on Synthetic Datasets

To enhance the rigor of the insights presented, we employ synthetic datasets created with the GluonTS library[13], encompassing a baseline dataset and variants with pronounced trends, strong seasonality, and complex data distribution (see Table 22). Specifically, we generate these datasets by superimposing four components - trend, seasonality, noise, and anomaly - each with adjustable intensity parameters. The seasonality component is defined by period hyper-parameters and intensity coefficients; the trend by slope intensity; the noise by Gaussian distribution sampling with adjustable intensity; and the anomaly by occurrence probability and maximum intensity.

---

[13]`https://ts.gluon.ai/stable/tutorials/data_manipulation/index.html`

Subsequent experiments on these synthetic datasets (refer to Table 23), using representative models, validate the empirical findings established on other datasets with `ProbTS`. Key observations include the declining performance of autoregressive decoding models, such as TimeGrad, in the presence of increasing trends, improved performance for models using autoregressive decoding with intensifying seasonality, and the competitive performance of probabilistic methods like CSDI in handling more complex data distributions.

Table 22: Quantitative assessment of intrinsic characteristics for synthetic datasets. The JS Div denotes Jensen–Shannon divergence, where a lower score indicates closer approximations to a Gaussian distribution.

| Dataset | Normal | Strong Trend | Strong Seasonality | Complex Distribution |
|---|---|---|---|---|
| Trend $F_T$ | 0.105 | 0.554 | 0.105 | 0.064 |
| Seasonality $F_S$ | 0.302 | 0.302 | 0.791 | 0.190 |
| JS Div. | 0.261 | 0.248 | 0.272 | 0.469 |

Table 23: Results on synthetic datasets. The look-back window and forecasting horizon are 30.

| Model | Normal | | Strong Trend | | Strong Seasonality | | Complex Distribution | |
|---|---|---|---|---|---|---|---|---|
| | CRPS | NMAE | CRPS | NMAE | CRPS | NMAE | CRPS | NMAE |
| DLinear | 0.013 | 0.013 | **0.001** | **0.001** | 0.014 | 0.014 | 0.301 | 0.301 |
| PatchTST | **0.012** | **0.012** | **0.001** | **0.001** | **0.012** | **0.012** | 0.275 | **0.275** |
| TimeGrad | 0.024 | 0.032 | 0.042 | 0.048 | 0.022 | 0.028 | 0.283 | 0.338 |
| CSDI | 0.013 | 0.014 | 0.010 | 0.007 | 0.020 | 0.027 | **0.269** | 0.301 |

## D.6 Case Study

To intuitively demonstrate the distinct characteristics of point and probabilistic estimations, a case study was conducted on short-term datasets. Figure 10 illustrates that point estimation yields single-valued, deterministic estimates, in contrast to probabilistic methods, which model continuous data distributions as depicted in Figure 11. This modeling of data distributions captures the uncertainty in forecasts, aiding decision-makers in fields such as weather and finance to make more informed choices. It is also observed that while both methods align well with ground truth values in short-term forecasting datasets, they struggle to accurately capture outliers, particularly noted in the Wikipedia dataset.

## D.7 Model Efficiency

For reference, detailed results regarding memory usage and time efficiency for five representative models on long-term forecasting datasets are provided here. Table 24 displays the computation memory of various models with a forecasting horizon set to 96. Additionally, Table 25 compares the inference time of these models on long-term forecasting datasets, illustrating the impact of changes in the forecasting horizon.

Table 24: Computation memory. The batch size is 1 and the prediction horizon is set to 96.

| Metric | Dataset | DLinear | PatchTST | LSTM NVP | TimeGrad | CSDI |
|---|---|---|---|---|---|---|
| | ETTm1 | 0.075 | 2.145 | 1.079 | 1.233 | 1.720 |
| | Electricity-L | 0.076 | 2.146 | 3.680 | 3.472 | 1.370 |
| NPARAMS (MB) | Traffic-L | 0.078 | 2.149 | 15.926 | 8.298 | 1.390 |
| | Weather-L | 0.075 | 2.145 | 3.085 | 0.574 | 1.721 |
| | Exchange-L | 0.075 | 0.135 | 1.979 | 0.488 | 1.720 |
| | ETTm1 | 0.002 | 0.009 | 0.010 | 0.012 | 0.027 |
| | Electricity-L | 0.060 | 0.068 | 0.129 | 0.128 | 1.411 |
| Max GPU Mem. (GB) | Traffic-L | 0.161 | 0.168 | 0.361 | 0.333 | 9.102 |
| | Weather-L | 0.004 | 0.012 | 0.021 | 0.012 | 0.070 |
| | Exchange-L | 0.002 | 0.002 | 0.013 | 0.008 | 0.030 |

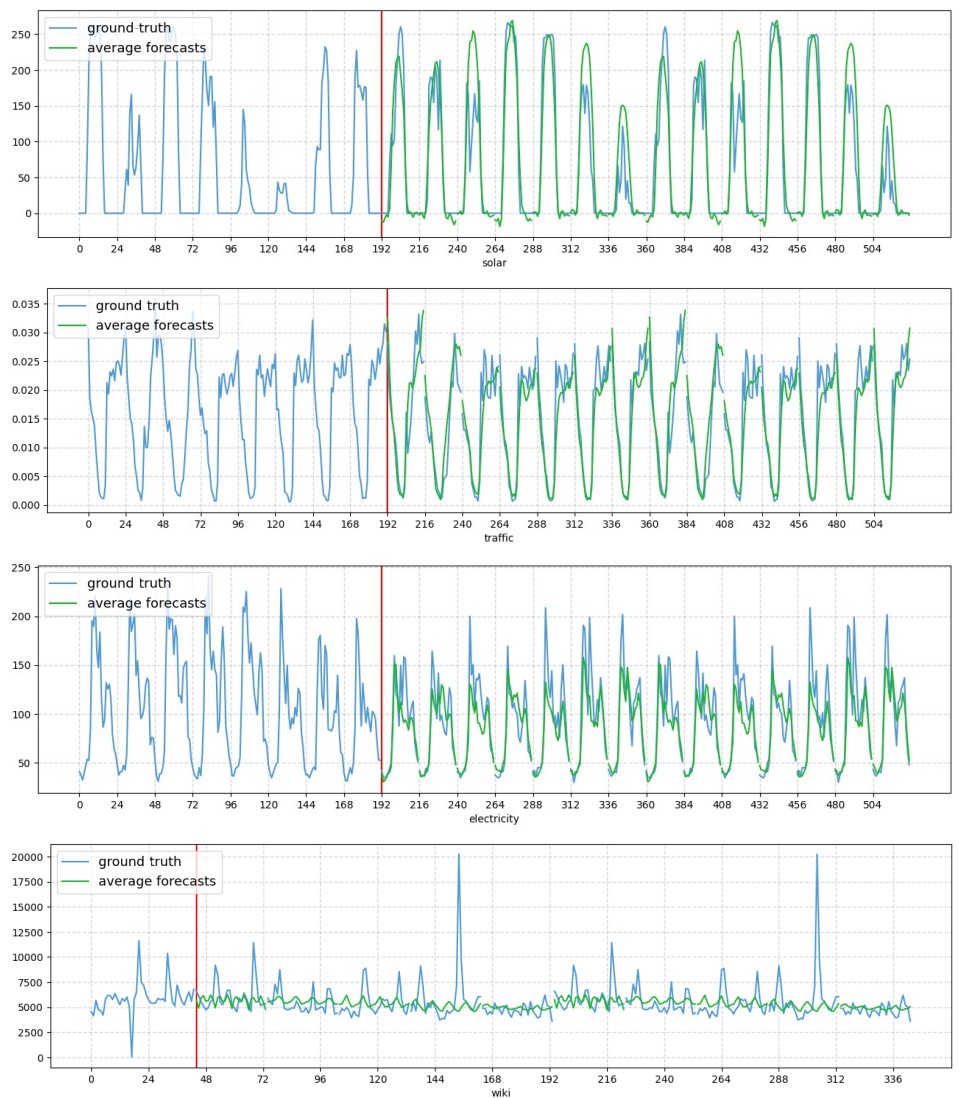

Figure 10: Point forecasts from the PatchTST model and the ground-truth value on short-term forecasting datasets.

## D.8 Further discussion on uni- / multi-variate modeling

Our experiments indicate that the choice between univariate and multivariate modeling is not a primary factor in fulfilling the essential forecasting needs considered in this paper.

### D.8.1 Discussion

We discuss the differences between univariate and multivariate modeling from two perspectives:

- Dataset Perspective: Whether the dataset is prepared for univariate or multivariate benchmarking.
- Model Perspective: How the model handles multivariate data, treating each variable channel independently or not.

**Dataset Perspective** All datasets listed in Table 5 are typically referred to as multivariate datasets, indicating that there may be strong connections across different variables. Despite this implication, when developing forecasting models, we can treat each variable channel independently, essentially turning a multivariate dataset into a univariate setup. In contrast, some datasets, like M4, M5, and

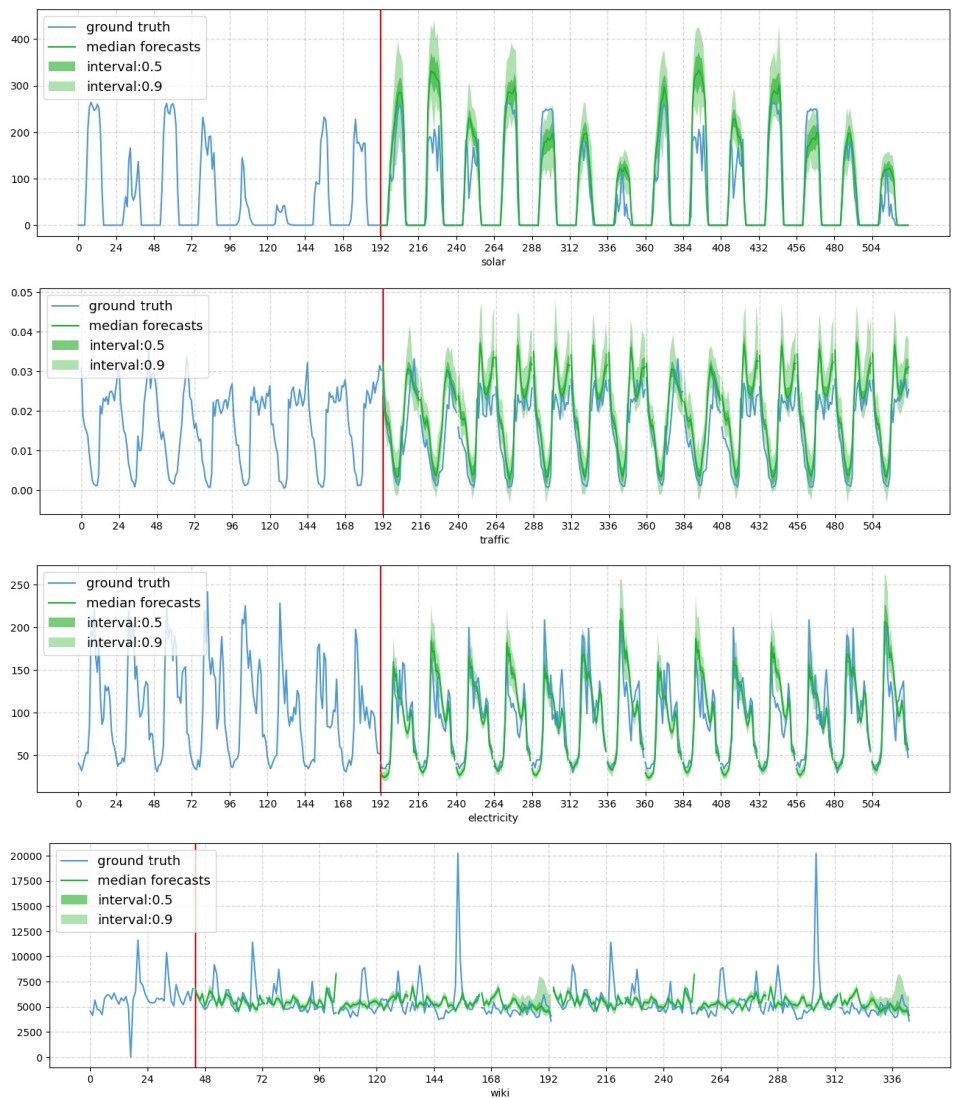

Figure 11: Forecasting intervals from the TimeGrad model and the ground-truth value on short-term forecasting datasets.

TOURISM listed in Table 20, explicitly serve univariate modeling. We rarely see multivariate models being developed for these univariate cases.

**Model Perspective**   We have observed different preferences for univariate and multivariate modeling. Existing models can be categorized into three groups:

- **Native Univariate Models**
    - Classical models like N-BEATS and N-HiTS.
    - Most time-series foundation models, such as TimesFM and Chronos.
- **Native Multivariate Models**
    - Most probabilistic models, such as CSDI and TimeGrad.
    - Some point forecasting models, such as Informer and Autoformer.
- **Hybrid Models of Univariate and Multivariate Modeling**
    - Some classical models, such as PatchTST.
    - Some time-series foundation models, such as MOIRAI.

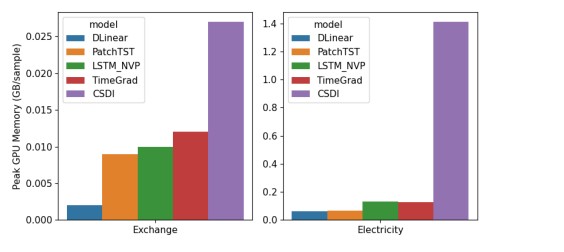

(a) Computational memory.

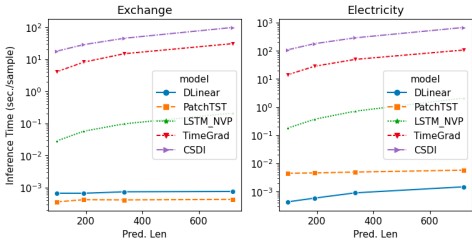

(b) Inference time.

Figure 12: Comparison of computational efficiency. The forecasting horizon is set to 96 for calculating memory usage.

Table 25: Comparison of inference time (sec./sample).

| Dataset | Pred len | DLinear | PatchTST | LSTM NVP | TimeGrad | CSDI |
|---------|----------|---------|----------|----------|----------|------|
| ETTm1-L | 96 | 0.0003 ± 0.0000 | 0.0003 ± 0.0000 | 0.0352 ± 0.0007 | 4.1067 ± 0.0504 | 16.3280 ± 0.0747 |
|         | 192 | 0.0003 ± 0.0000 | 0.0003 ± 0.0000 | 0.0697 ± 0.0020 | 7.8979 ± 0.0403 | 25.8378 ± 0.3124 |
|         | 336 | 0.0003 ± 0.0000 | 0.0003 ± 0.0000 | 0.1221 ± 0.0044 | 13.6197 ± 0.1023 | 39.8832 ± 0.2157 |
|         | 720 | 0.0004 ± 0.0000 | 0.0003 ± 0.0000 | 0.2603 ± 0.0020 | 28.6074 ± 1.1346 | 86.1862 ± 0.1863 |
| Electricity-L | 96 | 0.0004 ± 0.0000 | 0.0045 ± 0.0001 | 0.1783 ± 0.0006 | 13.8439 ± 0.0054 | 388.3150 ± 0.2155 |
|         | 192 | 0.0006 ± 0.0000 | 0.0046 ± 0.0000 | 0.3700 ± 0.0010 | 27.6683 ± 0.0368 | 659.4284 ± 0.2003 |
|         | 336 | 0.0008 ± 0.0000 | 0.0049 ± 0.0000 | 0.7157 ± 0.0028 | 48.4456 ± 0.0279 | - |
|         | 720 | 0.0015 ± 0.0000 | 0.0057 ± 0.0000 | 2.0785 ± 0.0186 | 104.1473 ± 0.1465 | - |
| Traffic-L | 96 | 0.0010 ± 0.0001 | 0.0102 ± 0.0000 | 0.3695 ± 0.0022 | 31.7644 ± 0.0101 | - |
|         | 192 | 0.0013 ± 0.0000 | 0.0106 ± 0.0000 | 0.8287 ± 0.0094 | 63.5832 ± 0.0060 | - |
|         | 336 | 0.0020 ± 0.0000 | 0.0114 ± 0.0001 | 1.6945 ± 0.0026 | 111.4147 ± 0.0169 | - |
|         | 720 | 0.0039 ± 0.0000 | 0.0137 ± 0.0000 | 5.0963 ± 0.0018 | 258.1274 ± 0.6088 | - |
| Weather-L | 96 | 0.0002 ± 0.0000 | 0.0004 ± 0.0000 | 0.0800 ± 0.0016 | 4.1261 ± 0.0812 | 37.8984 ± 0.0782 |
|         | 192 | 0.0003 ± 0.0000 | 0.0004 ± 0.0000 | 0.1568 ± 0.0008 | 8.2913 ± 0.5544 | 62.0223 ± 0.2329 |
|         | 336 | 0.0003 ± 0.0000 | 0.0004 ± 0.0000 | 0.2482 ± 0.0297 | 14.2391 ± 0.4891 | 96.8704 ± 0.2258 |
|         | 720 | 0.0003 ± 0.0000 | 0.0005 ± 0.0000 | 0.5447 ± 0.0249 | 29.4407 ± 0.3519 | 216.6044 ± 0.4253 |
| Exchange-L | 96 | 0.0006 ± 0.0000 | 0.0004 ± 0.0000 | 0.0284 ± 0.0001 | 4.1069 ± 0.0981 | 17.8655 ± 0.1282 |
|         | 192 | 0.0007 ± 0.0000 | 0.0004 ± 0.0000 | 0.0563 ± 0.0008 | 8.1576 ± 0.0911 | 28.5456 ± 0.0873 |
|         | 336 | 0.0007 ± 0.0000 | 0.0004 ± 0.0000 | 0.0966 ± 0.0007 | 14.4593 ± 0.4466 | 44.9733 ± 0.3820 |
|         | 720 | 0.0007 ± 0.0000 | 0.0004 ± 0.0000 | 0.2085 ± 0.0046 | 30.1443 ± 0.5378 | 97.7417 ± 0.2606 |
| ILI-L | 24 | 0.0002 ± 0.0000 | 0.0008 ± 0.0001 | 0.0080 ± 0.0001 | 1.0427 ± 0.0190 | 12.4038 ± 0.1681 |
|         | 192 | 0.0002 ± 0.0000 | 0.0008 ± 0.0000 | 0.0121 ± 0.0003 | 1.5762 ± 0.0282 | 12.7187 ± 0.1344 |
|         | 336 | 0.0002 ± 0.0000 | 0.0008 ± 0.0000 | 0.0155 ± 0.0002 | 2.1344 ± 0.0660 | 12.7386 ± 0.1868 |
|         | 720 | 0.0002 ± 0.0000 | 0.0008 ± 0.0000 | 0.0196 ± 0.0004 | 2.5787 ± 0.0594 | 12.5407 ± 0.0481 |

Native univariate models can also be applied to multivariate datasets by treating them as univariate cases. Similarly, native multivariate models can be applied to univariate datasets by setting the variable dimension to 1. Hybrid models typically include specific modes to activate univariate and multivariate functionalities. For example, in PatchTST, we can use a shared forecasting head for univariate modeling or assign a specific forecasting head for each variable channel to differentiate different variables.

We compile a summary table (Table 26) delineating how models from each branch address the multivariate aspect. Despite a thorough investigation, we have not identified a clear pattern linking the modeling of cross-channel interactions to overall model performance. A notable trend is the prevalent use of a channel-mixing approach in most studies. However, findings are diverse; models like DLinear and PatchTST suggest that processing channels independently can yield superior results, while others like CSDI indicate that explicit modeling of cross-channel interactions offers significant advantages. This diversity underscores the ongoing exploration of the impact of cross-channel interactions on forecasting performance.

Table 26: Summary of how existing models handle multivariate time series.

| Model | Research branch | Process channels independently |
|---|---|---|
| Customized neural architectures | N-BEATS [53] | ✓ |
| | N-HiTS [11] | ✓ |
| | Autoformer [72] | ✗ |
| | Informer [78] | ✗ |
| | LTSF-Linear [75] | ✗/✓ |
| | PatchTST [49] | ✗/✓ |
| | TimesNet [71] | ✗ |
| Probabilistic estimation | DeepAR [60] | ✓ |
| | GP-copula [59] | ✗ |
| | LSTM NVP [58] | ✗ |
| | LSTM MAF [58] | ✗ |
| | Trans MAF [58] | ✗ |
| | TimeGrad [57] | ✗ |
| | CSDI [62] | ✗ |
| | SPD [7] | ✗ |

### D.8.2 Additional Experiments on uni- / multi-variate modeling

In Table 27, we include additional experiments comparing univariate and multivariate modeling of PatchTST across different datasets. Our observation is that there is no definitive answer as to which approach is superior; it depends on the nature of the dataset. Some datasets benefit from modeling variable correlations, while others perform better with independent modeling. The performance gaps are not significant.

Similar observations have been reported in MOIRAI, which allows either univariate or multivariate modes by controlling its cross-variate attention masks. When applied to a downstream forecasting scenario, it can search over the validation set to determine which configurations to activate. We believe such a design could serve as a good example of unifying univariate and multvariate.

Table 27: The comparison of PatchTST on univariate and multivariate modeling.

| Dataset | Pred. Horizon | PatchTST (Multivariate) | PatchTST (Univariate) |
|---|---|---|---|
| ETTh1 | 96 | 0.3239 | 0.3212 |
| | 192 | 0.3609 | **0.3562** |
| | 336 | 0.3763 | **0.3737** |
| | 720 | **0.3882** | 0.3909 |
| ETTm1 | 96 | **0.2652** | 0.2739 |
| | 192 | **0.2926** | 0.2961 |
| | 336 | 0.3101 | 0.3188 |
| | 720 | 0.345 | 0.3463 |
| Electricity | 96 | **0.0832** | 0.0857 |
| | 192 | **0.0899** | 0.0912 |
| | 336 | **0.0995** | 0.1001 |
| | 720 | 0.1183 | 0.116 |
| Exchange | 96 | 0.0243 | **0.0235** |
| | 192 | 0.0348 | **0.0336** |
| | 336 | 0.0471 | **0.0462** |
| | 720 | 0.0787 | **0.0777** |
| Weather | 96 | 0.0872 | **0.0837** |
| | 192 | 0.0924 | **0.0858** |
| | 336 | 0.0934 | **0.0903** |
| | 720 | 0.0993 | **0.0953** |

