# OpenReview forum: "ProbTS: Benchmarking Point and Distributional Forecasting across Diverse Prediction Horizons"
_NeurIPS.cc/2024/Datasets_and_Benchmarks_Track — NeurIPS 2024 Track Datasets and Benchmarks Poster_

### Official Review · Reviewer_HvVZ · 2024-07-15
**An extensive benchmarking framework for evaluating both point-based and probabilistic time-series forecasting models.**

**Rating:** 8
**Confidence:** 4
**Clarity:** The paper is well written and well st…

**Review:**

The paper is well-written, well-structured and easy to follow. The research is of high quality with an extensive benchmark done with interesting analyses utilizing novel dimensions. The benchmarking both point-based and probabilistic forecasting methods, and  the dimensions along which this research is evaluated make this research original, and I think it will have high impact and set a new standard for how to  evaluate forecasting methods. If this is the case, it will be a win for the scientific community focusing on forecating.

The analysis is good and shows that deep learning methods perform better than traditional statistical methods while XGBoost keeps up. An interesting finding is that methods relying on auto-regression fail to predict longer horizons and non-auto regressive methods fail at shorter horizons but outperform AR-methods on longer horizons. While this is not very surprising, it opens up for a discussion on future methods combining these methods.

**Strengths:**

Strengths:
 * Reproducibility: Not only is the code of the framework share publicly on Github, but the authors have filled out the reproducibility checklist.
* Well-written, well-structured and clear.
* The appendix contains all a lot of information. This is really nice.
* Good analysis along interesting dimension.
* Non-deep learning baselines are also included.

**Additional Feedback:**

Thanks! I think this is a very nice contribution – whether it will be accepted at NeurIPS or not.

**Correctness:**

The experiments seem designed in a sound way to me. The evaluation and analysis are performed in a good way.

**Documentation:**

The framework relies on open and publicly available datasets that are documented elsewhere. The documentation on the Github pages looks adequate.

I did not do a very deep look at this though.

**Ethics:**

N/A.

**Limitations:**

In general, I think this paper controls for many of the sources of irreproducibility mentioned in (Gundersen et al., 2022), especially study design, data (observation), evaluation and documentation factors. I am curious to know what the authors think about how results are affected by algorithm and implementation factors.

Gundersen, O. E., Coakley, K., Kirkpatrick, C., & Gil, Y. (2022). Sources of irreproducibility in machine learning: A review. arXiv preprint arXiv:2204.07610.

**Opportunities For Improvement:**

Weaknesses:
* The abstract and title indicate that this is a general framework, which it is, but in the introduction, it is stated that this method focuses on deep learning methods. It was not clear to me that non-deep learning baselines were included until I read the appendix. This should be emphasized better, as DL-papers often do not add statistical and other machine learning methods as baselines although these perform better for some datasets.
* It is not clear from the paper how easily extendable this framework is. For this, framework to have high impact, it should be easy to extend this. A framework that could easily integrate with GluonTS or DarTS or other frameworks would be very valuable. By reading the paper, I do not get the impression that this is the case.  However, I see from the Girhub documentation that this can be done. Please emphasize this in the text.

**Relation To Prior Work:**

It is clearly discussed how the work differs from previous work. The prior work is covered well.
The Monash time series forecasting archive paper (Godahewa et al., 2021) published at this track at NeurIPS in 2021 is not mentioned. It could be.

Also, (Makridakis et al, 2018) mentions the need to compare to naïve and simple statistical baselines than deep learning baselines.

## References

Godahewa, R. W., Bergmeir, C., Webb, G. I., Hyndman, R., & Montero-Manso, P. Monash Time Series Forecasting Archive. In Thirty-fifth Conference on Neural Information Processing Systems, Datasets and Benchmarks Track (Round 2).

Makridakis, S.; Spiliotis, E.; and Assimakopoulos, V. 2018. Statistical and Machine Learning forecasting methods: Con- cerns and ways forward. PloS one, 13(3): e0194889.

**Summary And Contributions:**

The paper presents ProbTS that is a benchmark tool designed as a uniformed platform to evaluate both point-based and probabilistic forecasting methods on different datasets and over different horizons. An extensive amount of methods a benchmarked and their performance on long and short term horizons. The datasets are described along three different dimensions, trend, seasonality and non-Gaussianity, which are used in the analysis.

---

> ### Author Rebuttal · Authors · 2024-08-17
>
> Thank you for your thorough review. We sincerely appreciate your recognition of our work, and we hope that our responses will help further enhance the completeness and rigor of this paper.
>
> ## Scope of this framework
>
> > The abstract and title indicate that this is a general framework, which it is, but in the introduction, it is stated that this method focuses on deep learning methods. It was not clear to me that non-deep learning baselines were included until I read the appendix. This should be emphasized better, as DL-papers often do not add statistical and other machine learning methods as baselines although these perform better for some datasets.
> >
>
> The primary focus of this benchmarking study is to explore the divergences among recent deep learning approaches in time series forecasting, aiming to provide a unified perspective. To give a more comprehensive overview and to investigate the performance gains of these methods, we also included non-deep learning baselines for comparison.
>
> We appreciate your suggestion and will revise the introduction to explicitly mention that non-DL methods are also included in our study.
>
> ## Extendability
>
> > It is not clear from the paper how easily extendable this framework is. For this, framework to have high impact, it should be easy to extend this. A framework that could easily integrate with GluonTS or DarTS or other frameworks would be very valuable. By reading the paper, I do not get the impression that this is the case. However, I see from the Girhub documentation that this can be done. Please emphasize this in the text.
> >
>
> Thank you for your kind suggestion. We will emphasize the framework’s extensibility in the paper. Also, we will continuously improve the framework’s extensibility in the future update to further enhance its usability.
>
> ## How results are affected by algorithm and implementation factors
>
> > In general, I think this paper controls for many of the sources of irreproducibility mentioned in (Gundersen et al., 2022), especially study design, data (observation), evaluation and documentation factors. I am curious to know what the authors think about how results are affected by algorithm and implementation factors.
> >
>
> Thank you for your insightful question.
>
> As you pointed out, many algorithm and implementation factors can influence performance comparisons, and it is challenging to standardize all these elements across different studies.
>
> To address these variations, a unified tool and third-party implementations are crucial for validating model performance. This has been a key focus of our study. We have made substantial efforts to standardize the implementation and optimization of all models, including setting consistent initialization seeds, maintaining a uniform computational environment for training and inference, aligning hyper-parameter tuning processes, and ensuring consistent data handling practices such as shuffling and batch ordering.
>
> Moreover, we recognize that advancing reproducibility is a shared responsibility within the research community. We hope the proposed benchmarking platform, ProbTS, will be a valuable resource for researchers, paving the way for greater reproducibility of forecasting models and establishing a fair and comprehensive standard for evaluation.
>
> Finally, we will incorporate all of your other valuable suggestions in our revised manuscript, such as citing important references and considering integration with other frameworks. We are very grateful for your encouraging words and constructive feedback.

---

### Official Review · Reviewer_Yg3R · 2024-07-22

**Rating:** 8
**Confidence:** 3
**Clarity:** Overall, the paper is well written.

**Review:**

Pros:
+ The first work benchmarks wide range of models from point forecasting to distributional forecasting and LLM-based forecasting models, giving a wide overview on the current performance of deep-learning-based time-series forecasting models.
+ Analysis the models's performance regarding several aspects.
+ Provide valuable insights.

Cons:
+ Lack of definitions of long-term or short-term forecasting scenarios. They should be clearly define in the problem formulation. Moreover, readers outside of this domain might be difficult to understand the content.
+ There is unclear about the fairness in settings of competitors. For example, some models might use a normailization method (e.g., iTransformer, PatchTST) which could be highly boost the performance by countering the distribution shift [1]. Please explain this problem in details.

[1] Kim, Taesung, et al. "Reversible instance normalization for accurate time-series forecasting against distribution shift." International Conference on Learning Representations. 2021.

**Strengths:**

Along with these pros above, this paper provides a big picture of current status of time-series forecasting models, hinting potential research direction on the future work.

**Additional Feedback:**

None

**Correctness:**

There is an unclear point as above-mentioned which could affect on the correctness of claims and results.

**Documentation:**

There is sufficient detail to support reproducibility.

**Ethics:**

There is no ethical concerns with the submission that warrant further discussion or review.

**Limitations:**

The authors adequately addressed the limitations and potential negative societal impact of their work.

**Opportunities For Improvement:**

This study still utilized common benchmark datasets which are not large and not complex enough to see the true performance of competitors.

**Relation To Prior Work:**

It clearly discussed how this work differs from previous contributions.

**Summary And Contributions:**

This paper benchmarks a wide range of time-series forecasting models, both point and distributional forecasting models, including several recent Large Language Model (LLM) based models. The main contributions:
+ Provide a benchmark tool designed for evaluation of essential forecasting needs.
+ Provide comprehensive analysis of these models, regarding several aspects such as decoding schemes, strength of trend and seasonality and complexity of data distribution.

---

> ### Author Rebuttal · Authors · 2024-08-17
>
> ## More clearer definitions
>
> > Lack of definitions of long-term or short-term forecasting scenarios. They should be clearly define in the problem formulation. Moreover, readers outside of this domain might be difficult to understand the content.
> >
>
> Thank you for your feedback regarding the clarity of the paper. We would like to clarify that there is no strict formal definition for long-term and short-term forecasting; the distinction often depends on the specific application scenario. Historically, in 1990s, “long-term” dependencies were described qualitatively in the neural network literature. For example, Y. Bengio and S. Hochreiter discussed long-term dependencies as situations where predictions at time $t$ rely on inputs from much earlier time $\tau$ , with  $\tau\ll t$ [1,2].  In the context of time-series forecasting, around 2005, a more concrete distinction emerged, where one-step ahead predictions are typically considered short-term, and multi-step ahead predictions are viewed as long-term [3,4].
>
> In the era of DL-based models, these definitions remain flexible and are often application-specific. For example, in air quality forecasting, [5] considered predictions over one week as long-term, while predictions with lead times of one to three days were considered short-term. Similarly, in recent studies involving Transformers, predictions of 48 steps or fewer were categorized as short-term, while predictions of 96 steps or more were categorized as long-term [6,7].
>
> Based on extensive research and established conventions, we define short-term and long-term forecasting by the primary periodicity of the data. Specifically, if the main periodicity is T (for example, in most datasets with hourly frequency, the main periodicity T is 24), then predictions with a length  L ≤ T  can be considered short-term [8,9], while those with  L >> T are considered long-term [7,10,11].
>
> We will introduce a discussion in the Appendix about the explanations of long-term and short-term forecasting to make the content more accessible to readers outside this domain.
>
> [1] Bengio, Y., et al. (1994). Learning long-term dependencies with gradient descent is difficult. *IEEE transactions on neural networks*.
>
> [2] Hochreiter, S., & Schmidhuber, J. (1997). Long short-term memory. *Neural computation*.
>
> [3] Sorjamaa, A., et al. (2007). Methodology for long-term prediction of time series. *Neurocomputing*.
>
> [4] Menezes Jr, J. M. P., & Barreto, G. A. (2008). Long-term time series prediction with the NARX network: An empirical evaluation. *Neurocomputing*.
>
> [5] Jiang, S., et al. (2021). Long-and short-term time series forecasting of air quality by a multi-scale framework. *Environmental Pollution*.
>
> [6] Zhou, H., et al. (2021). Informer: Beyond efficient transformer for long sequence time-series forecasting. *AAAI*.
>
> [7] Zeng, A., et al. (2023). Are transformers effective for time series forecasting?. *AAAI*.
>
> [8] Rasul, K., et al. (2021). Autoregressive denoising diffusion models for multivariate probabilistic time series forecasting. *ICML*.
>
> [9] Tashiro, Y., et al. (2021). CSDI: Conditional score-based diffusion models for probabilistic time series imputation. *NeurIPS*.
>
> [10] Nie, Y., et al. (2023). A time series is worth 64 words: Long-term forecasting with transformers. *ICLR.*
>
> [11] Liu, Y., et al. (2024). iTransformer: Inverted transformers are effective for time series forecasting. *ICLR.*

---

> ### Author Rebuttal · Authors · 2024-08-17
>
> ## Benchmark datasets
>
> > This study still utilized common benchmark datasets which are not large and not complex enough to see the true performance of competitors.
> >
>
> We understand your concern regarding the datasets used for benchmarking.
>
> Here, we provide additional explanations to address these concerns.
>
> First and foremost, our dataset selection is motivated by the need to investigate the capabilities of existing studies in fulfilling essential forecasting needs (different prediction horizons, point, and distributional forecasts). Compared to existing studies that usually focus on a subset of these aspects, our study offers comprehensive coverage of datasets used in different research threads, including:
>
> - Almost all datasets used to evaluate short-term probabilistic forecasting
>     - Compiled from studies like CSDI, TimeGrad, GRU NVP, etc.
> - Almost all datasets used to evaluate long-term point forecasting
>     - Compiled from studies such as iTransformer, PatchTST, etc.
> - Datasets used in the evaluation of universal time-series foundation models (Overlapped with the above two)
>     - Such as TimesFM, MOIRAI, Chronos, etc.
>
> We have included these datasets to ensure a consistent comparison with existing models and because they have been endorsed by many previous studies and prestigious conferences. Moreover, we have included experiments on additional datasets in the appendix, including:
>
> - Some univariate datasets
>     - Compiled from M4, M5, Tourism (See Appendix D.3, Table 15 and 16)
> - Some synthetic datasets
>     - To re-confirm our findings (See Appendix D.4, Table 17 and 18)
>
> Second, we would like to highlight that the included datasets cover a wide range of data characteristics that help analyze the pros and cons of different methodological choices.
>
> - As shown in Table 1, these datasets display distinct characteristics of trend, seasonality, and distribution complexity.
> - As summarized in Table 6 (Appendix B.1.1), these datasets also cover a wide range of variable numbers, sampling frequencies, timesteps, and domains.
>
> Therefore, we believe these datasets constitute a solid foundation for the comparison of existing models.
>
> Third, please note that the primary focus of this study is to investigate how well existing methodological designs address essential forecasting needs. Therefore, our major efforts have been devoted to the methodological aspects, including:
>
> - Re-implementing, unifying, and comparing cutting-edge models from different research threads.
> - Identifying typical methodological choices and analyzing their reasons.
> - Investigating the pros and cons of different choices in fulfilling essential forecasting needs.
> - Revealing underexplored or unnoticed areas for future research.
>
> Finally, we appreciate your suggestion to include larger and more complex datasets, as this will undoubtedly enhance a benchmarking study. Our tool supports simple extensions for incorporating new datasets, and we are open to adding any new datasets that you believe are necessary, especially those with specific data characteristics beyond the existing coverage of our study.

---

> ### Author Rebuttal · Authors · 2024-08-17
>
> Thank you for your thorough review and valuable feedback. We greatly appreciate your suggestion to investigate the effects of different normalization methods, which led to some interesting and profound findings. Please see our detailed responses below, and let us know if you have any remaining concerns.
>
> ## Fairness in settings
>
> > There is unclear about the fairness in settings of competitors. For example, some models might use a normailization method (e.g., iTransformer, PatchTST) which could be highly boost the performance by countering the distribution shift [1]. Please explain this problem in details.
> >
>
> Thank you so much for posing such a sharp and insightful question! Following your suggestion, we have further investigated the effects of different normalization methods on short-term and long-term forecasting scenarios.
>
> In the submitted manuscript, our benchmarking mainly respects and follows the default normalization methods used by different models. Initially, a dataset typically undergoes a preprocessing stage, where global mean and standard deviation statistics from the training set are used to normalize all time-series values. Subsequently, when we feed a batch of time-series segments into a time-series model, this model usually includes a local normalization module. We do observe distinct preferences in the choice of local normalization strategies among different models.
>
> - Long-term point forecasting models (e.g., PatchTST, iTransformer) typically adopt the RevIN strategy. Given a batch of time-series segments within a lookback window, RevIN is essentially a per-series z-score normalization augmented with some learnable affine parameters. The claimed advantage is its effectiveness in addressing distribution shifts, particularly in long-term forecasting windows.
> - In contrast, the normalization strategy in short-term probabilistic forecasting models are rather ad-hoc yet very effective. Given a batch of time-series segments, such as $X \in R^{K \times L}$ (K: the number of variables, L: the length of the lookback window), it just perform a per-series scaling, $X_i^{norm} = \frac{X_i}{\sum_{t=1}^L |X_{i,t}| / L}, i=1,…,K$, to stabilize value ranges. For simplicity, we refer to this type of normalization as "Scaling.”
>
> To understand the effects of different normalization strategies, we selected representative models from these two categories and combined them with three types of normalization methods: RevIN, "Scaling", and "w/o norm" (no local normalization step, using the time-series values as given by the dataset-level preprocessing). Our results on short-term and long-term scenarios are included in Tables 1 and 2, respectively.
>
> Table 1. The impact of different normalization methods in short-term forecasting scenarios. The optimal normalization method for each model is indicated in bold.
> |  | PatchTST | PatchTST | PatchTST | CSDI | CSDI | CSDI | TimeGrad | TimeGrad | TimeGrad | GRU NVP | GRU NVP | GRU NVP |
> | --- | --- | --- | --- | --- | --- | --- | --- | --- | --- | --- | --- | --- |
> | Dataset (Metric) | ReVIN | Scaling | w/o norm | ReVIN | Scaling | w/o norm | ReVIN | Scaling | w/o norm | ReVIN | Scaling | w/o norm |
> | Exchange (NMAE) | **0.0102** |0.0108|0.0111|0.0109|0.0109| **0.0099** | **0.0115** |0.0118|0.022| **0.0117** |0.0193|0.0176|
> | Exchange (CRPS) | **0.0102** |0.0108|0.0111|0.0081|0.0083| **0.0075** |0.01| **0.0093** |0.017| **0.0098** |0.0147|0.014|
> | Solar (NMAE) | **0.6275** |0.7105|0.7169|0.5847| **0.5832** |1.0| **0.6041** |0.7011|0.9162|1.1338| **0.6004** |0.6084|
> | Solar (CRPS) | **0.6275** |0.7105|0.7169|0.4767| **0.4594** |0.9499| **0.4945** |0.5455|0.8356|0.8766| **0.4714** |0.48|
> | Electricity (NMAE) |0.0659| **0.0645** |0.066| **nan** |0.0657|nan|0.0852| **0.071** |0.9742| **0.0825** |0.0825|0.0908|
> | Electricity (CRPS) |0.0659| **0.0645** |0.066| **nan** |0.0519|nan|0.0673| **0.0563** |0.9681| **0.0628** |0.0628|0.0681|
> | Traffic (NMAE) | **0.2001** |0.2036|0.2168|0.1731|0.1806| **nan** |0.2167| **0.1516** |0.1693|0.2279| **0.2087** |0.4719|
> | Traffic (CRPS) | **0.2001** |0.2036|0.2168|0.1502|0.1572| **nan** |0.1806| **0.128** |0.14|0.1837| **0.1676** |0.3722|
> | Wikipedia (NMAE) | **0.2529** |0.3245|0.3695| **nan** |0.2437|0.2698|0.3278| **0.3257** |0.9998|0.4242| **0.3394** |0.5632|
> | Wikipedia (CRPS) | **0.2529** |0.3245|0.3695| **nan** |0.206|0.2276| **0.2757** |0.2773|0.9969|0.3445| **0.2898** |0.4952|

---

> > ### Author Rebuttal · Authors · 2024-08-17
> >
> > Table 2. The impact of different normalization methods in long-term forecasting scenarios (CRPS). The optimal normalization method for each model is indicated in bold.
> >
> > | Dataset | Pred hor. | PatchTST | PatchTST | PatchTST | CSDI | CSDI | CSDI | GRU NVP | GRU NVP | GRU NVP |
> > | --- | --- | --- | --- | --- | --- | --- | --- | --- | --- | --- |
> > |  |  | ReVIN | Scaling | w/o norm | ReVIN | Scaling | w/o norm | ReVIN | Scaling | w/o norm |
> > | ETTh1 | 96 | **0.3248** |0.3437|0.3474| **0.2764** |0.363|0.3671| **0.2771** |0.5946|0.4444|
> > | ETTh1 | 192 | **0.3597** |0.3943|0.3993| **0.3553** |0.4352|0.4673| **0.3076** |0.4729|0.5056|
> > | ETTh1 | 336 | **0.3768** |0.4213|0.404| **0.3614** |0.4192|0.4814| **0.3081** |0.624|0.4803|
> > | ETTh1 | 720 | **0.3871** |0.4881|0.4653| **0.396** |0.4283|0.5609| **0.3641** |0.7295|0.5862|
> > | Exchange | 96 | **0.0241** |0.0254|0.0292|0.0216| **0.021** |0.0343| **0.0246** |0.0559|0.0605|
> > | Exchange | 192 | **0.0342** |0.0365|0.0408| **0.0383** |0.0388|0.0418| **0.034** |0.0814|0.086|
> > | Exchange | 336 | **0.0466** |0.0532|0.0544|0.0508| **0.0485** |0.051| **0.0475** |0.0948|0.0637|
> > | Exchange | 720 | **0.0787** |0.081|0.0845| **0.0881** |0.1042|0.0964| **0.0644** |0.1257|0.0724|
> > | Traffic | 96 |0.2553|0.2546| **0.2526** | OOM | OOM |OOM|0.2021|0.2134| **0.1952** |
> > | Traffic | 192 |0.2479| **0.2447** |0.2483| OOM  |OOM|OOM|0.211|0.2034| **0.2005** |
> > | Traffic | 336 |0.2492| **0.247** |0.2489| OOM  |OOM|OOM|0.2191|0.2143| **0.1983** |
> > | Traffic | 720 |0.2572| **0.2568** |0.2577| OOM  |OOM|OOM|0.2351|0.2222| **0.2198** |
> > | Weather | 96 | **0.0813** |0.1082|0.1071|0.1202| **0.0575** |0.0591| **0.0837** |0.1546|0.1005|
> > | Weather | 192 | **0.0884** |0.1171|0.1096|0.1599| **0.0717** |0.072| **0.0866** |0.1342|0.1177|
> > | Weather | 336 | **0.0898** |0.1167|0.1138|0.4296| **0.0856** |0.0977| **0.0868** |0.1715|0.1145|
> > | Weather | 720 | **0.0941** |0.1234|0.1151|0.1785|0.1419| **0.0975** | **0.0925** |0.1146|0.1979|
> >
> > We have uncovered some interesting findings.
> >
> > - **No dominating normalization strategies in short-term forecasting scenarios**.
> >     - In Table 1, we observe that RevIN does not bring robust and significant improvements, not only for PatchTST but also for probabilistic models like CSDI, TimeGrad, and GRU NVP.
> >     - The "Scaling" strategy, despite being empirical and lacking a principled motivation, is the most robust choice for these probabilistic models. This may explain why they commonly choose this strategy.
> >     - Sometimes it is acceptable not to use local normalization, but this approach can lead to severe issues in some cases, such as CSDI ("w/o norm") on Solar, TimeGrad ("w/o norm") on Exchange, and GRU NVP ("w/o norm") on Traffic.
> >     - Additionally, we found that combining CSDI with either "RevIN" or "w/o norm" may lead to NaN values during training. We have not yet identified the specific reasons for this but will provide updates as we investigate further.
> > - **RevIN's Effectiveness in Long-Term Forecasting with Exceptions.**
> >     - For PatchTST in long-term forecasting, RevIN is an indispensable component that brings significant improvements in most cases. The exception is the Traffic dataset, where the performance difference between RevIN and other normalization methods is marginal. **Interestingly, this aligns with our analysis of data characteristics: the Traffic dataset displays strong seasonality but less trending.** We speculate that the major distribution shift addressed by RevIN in other datasets is related to normalizing the effect of trending.
> >     - **The most surprising result is that RevIN has greatly enhanced GRU NVP in long-term forecasting.** In some cases, such as on ETTh1, GRU NVP with RevIN even beat PatchTST with RevIN. This is a remarkable finding because **it indicates that normalizing the trending effect could be a direction to alleviate error accumulation of autoregressive-based models in long-term forecasting.**
> >     - **For non-autoregressive probabilistic models like CSDI, RevIN does not seem to be an ideal match, as it only brings significant improvements on ETTh1.** On the Weather dataset, CSDI with "Scaling" performs significantly better than PatchTST or GRU NVP with RevIN.  **These observations also indicate research opportunities in developing effective normalizaiton strategies for non-autoregressive probabilistic models.**
> >
> > We are thrilled to have identified these findings based on your suggestion. We believe these insights point to further exploration directions in future research and have significantly enhanced this study. We will definitely include these experiments and analyses in our revised manuscript.
> >
> > Finally, regarding fairness in general, we acknowledge that there may be other methodological components that are critical but have been neglected. However, we hope that this study can serve as a starting point, covering some prominent design considerations, and attract the community’s attention and joint efforts to fully address essential forecasting needs in future research.

---

> ### Comment · Reviewer_Yg3R · 2024-08-20
> **Feedback**
>
> Thank you for your clarification.
>
> Adding this to the main text is probably better than in the Appendix.
> >Based on extensive research and established conventions, we define short-term and long-term forecasting by the primary >periodicity of the data. Specifically, if the main periodicity is T (for example, in most datasets with hourly frequency, the main >periodicity T is 24), then predictions with a length L ≤ T can be considered short-term [8,9], while those with L >> T are considered >long-term [7,10,11].
>
> Regarding fairness in settings, thank you for your results.
> It seems that utilizing the pre-processing steps such as ReVIN or scaling can negatively or positively affect on the performance of models, and it seems not be consistent in both short-term and long-term tasks.
> Based on that, I still have concerns whether claims regarding to other components (decoding scheme or point and prob forecasting algorithm) in the paper are correct or not, because authors didn't remove components that could made unfair comparision.

---

> > ### Author Rebuttal · Authors · 2024-08-21
> >
> > Thank you for your feedback.
> >
> > Regarding the definitions of short-term and long-term forecasting, we will follow your advice and incorporate them into the main text. Specifically, we plan to insert these definitions around lines 31-32, where we first introduce these concepts. Additionally, we will move the full discussion of how these concepts have evolved in the literature to the appendix.
> >
> > Regarding your additional concerns about "whether claims regarding other components in the paper are correct or not, because the authors didn't remove components that could lead to unfair comparisons," we would like to address your concerns from three aspects.
> > - First and foremost, we clarify that most of our conclusions, as highlighted in Section 4, are not influenced by the choice of normalization.
> > - Meanwhile, we acknowledge that considering the impact of normalization indeed enhances our understanding of existing autoregressive models and uncovers more opportunities for future research. Therefore, we greatly appreciate your suggestion on the normalization aspect, facilitating the discovery of new findings.
> > - Finally, we outline our concrete plans to seamlessly integrate the results and analyses of the additional experiments regarding the choice of normalization into the revised manuscript to enhance this work.
> >
> > Below, we provide detailed explanations on these three aspects.
> >
> > **1. Most of Our Analyses in Section 4 Are Not Influenced by the Choice of Normalization**
> >
> > In the first place, we would like to emphasize that the focus of this paper is to benchmark how well existing models, developed across different research threads, fulfill essential forecasting needs—namely, precise point and distributional forecasts across various horizons. To achieve this, it is crucial that we **respect and follow the standard implementation of existing models, as the normalization component is an integral part of these models**. Therefore, we would like to underscore that **benchmarking existing models using their native normalizaiton is indispensable**.
> >
> > We appreciate your suggestion to decouple the normalization component. Following your advice, we conducted additional experiments, the results of which are compiled in **Tables 1 and 2 of our previous response addressing the "fairness concern"**. Our detailed analysis of these new results indicates that most of the conclusions presented in Section 4 remain unaffected. Below, we provide a point-by-point explanation to justify this assertion.
> >
> > The analyses in lines 212-223 of Section 4.1, "Diminishing Advantages of Customized Architectures in Short-term Forecasting Scenarios", **remain unaffected** by the choice of normalization. As shown in Table 1, adjusting the normalization for customized architectures like PatchTST does not lead to performance improvements in short-term forecasting.
> >
> > The analyses in lines 224-237 of Section 4.2, "Significant Performance Degradation for Existing Probabilistic Methods in Long-term Distributional Forecasting", **remain valid** because existing probabilistic models, when following their official implentation, indeed face severe challenges in long-term scenarios.
> >
> > The analyses in lines 238-250 of Section 4.3, "Different Decoding Schemes & Challenges in Long-term Distributional Forecasting", **are still valid** because we are discussing existing probabilistic models with their native normalization designs in this paragraph, covering both non-autoregressive (NAR) and autoregressive (AR) decoding. However, we acknowledge that these conclusions can be extended and further enhanced by considering the normalization factor, which we will explain later.
> >
> > The analyses in lines 251-256 of Section 4.1, "The Unexpected Superiority of AR Decoding in Addressing Strong Seasonality", **are not affected** because we observed the same phenomenon for different normalization choices of GRU NVP on the Traffic dataset, as shown in Table 2.
> >
> > The analyses in lines 263-275 of Section 4.2 (about time-series foundation models), "Navigating the AR Decoding Challenge over Extended Forecasting Horizons", **remain valid**. As explained earlier, the normalization component is an integral part of these models, tightly bound to their architectures and parameters. It is worthy noting that both TimesFM and MOIRAI use non-learnable RevIN while Chronos uses the same "Scaling" normalization (dividing by a mean absolute value per variable) as CSDI. **Despite AR-based time-series foundation models, such as TimesFM and Chronos, using different normalization methods, their performance gaps with NAR-based MOIRAI greatly enlarge as the forecasting horizon extends**.
> >
> > The analyses in lines 276-291 of Section 4.2, "The Critical Role of Addressing Complex Data Distributions", **are also unaffected**.  The ability to capture complex data distributions without predefined distribution heads is a unique and orthogonal aspect to the normalization method used. Although MOIRAI employs RevIN-style normalization, it still faces this challenge. However, this section could also be further enhanced by incorporating additional findings on the benefits of RevIN when applied to GRU NVP because these insights may inspire future designs of time-series foundation models.

---

> > > ### Author Rebuttal · Authors · 2024-08-21
> > >
> > > **2. Considering the Impact of Normalization Further Enhances This Study and Uncovers New Research Opportunities**
> > >
> > > We would like to extend our gratitude once again for your suggestion to consider normalization as a special confounder when comparing different models. Based on our additional results in Tables 1 and 2, **the most remarkable finding is that RevIN can significantly mitigate the error accumulation challenge in long-term forecasting scenarios for existing AR-based probabilistic models like GRU NVP**. We consider this a novel discovery rather than a trivial combination, as it emerged only after unifying two previously disjoint research threads. This is evident because:
> > > - Almost no AR-based models have been developed in the literature for long-term point forecasting, to the best of our knowledge.
> > > - Existing probabilistic models rarely employ RevIN and seldom extend to long-term scenarios.
> > >
> > > We believe this finding is particularly significant as it **opens up new opportunities for developing AR-based models for long-term point forecasting**. It is likely that no AR-based long-term point forecasting models have been developed in the past because researchers have not previously combined AR-based models with RevIN-style normalization.
> > >
> > > **However, we also observe that RevIN does not address other challenges faced by existing short-term probabilistic models**. For instance, it does not consistently benefit NAR models like CSDI. As shown in Table 2, on the Weather dataset, CSDI + RevIN performs significantly worse than CSDI + Scaling. **Therefore, for NAR probabilistic models, further research into developing new normalization methods is necessary.** Additionally, the efficiency challenges of CSDI in multivariate and long-term scenarios are independent of the normalization method used.
> > >
> > > **3. Concrete Plans to Improve This Work by Considering Normalization Choice as the Third Methodological Aspect**
> > >
> > > We believe that integrating the findings and insights derived from additional experiments on different normalization choices can further enhance this work. We are confident that the following concrete plans will seamlessly incorporate these insights into the existing narrative.
> > >
> > > First, in lines 57-68 of the Introduction, we will extend the content by considering the normalization choice as the third methodological aspect, following the other two aspects (the first being point versus probabilistic forecasting, and the second being AR versus NAR models).
> > >
> > > Then, in the analysis section, we will explicitly mention our new finding (in collaboration with Reviewer Yg3R) that the RevIN strategy, developed in the long-term point forecasting literature, can significantly alleviate the error accumulation problem in existing AR-based probabilistic models. We will include a figure similar to those in Figure 3 to highlight this finding. Besides, we will include comprehensive experimental results in the Appendix.
> > >
> > > Additionally, when discussing future directions, we will explore potential opportunities motivated by the findings from the normalization experiments. These opportunities include investigating AR-based architectures for long-term forecasting and designing effective normalization strategies for NAR probabilistic models.
> > >
> > > Finally, we would like to thank the anonymous reviewers for their valuable feedback, which has greatly helped us improve this work.
> > >
> > > ---
> > >
> > > At last, we hope the above explanations help address your concerns regarding fairness. If you have any additional questions, please feel free to let us know.

---

> ### Author Response · Authors · 2024-08-27
> **A Kind Reminder for Additional Feedback**
>
> Dear Reviewer Yg3R,
>
> Thank you for your insightful questions and constructive suggestions. As the end of the author-reviewer discussion phase approaches, we would like to kindly remind you to share any additional feedback or questions you might have. We are very willing to engage in further discussion with you.
>
> Regarding your core concern about the normalization part, we have clarified that this paper considers the normalization module as an integral component of existing models (e.g., RevIN even includes learnable parameters). Therefore, all analyses and conclusions remain valid. Following your advice, we have additionally explored variations of normalization methods, which led to more findings and revealed new research opportunities.
>
> We are deeply grateful for your suggestion to consider normalization as an additional factor. We would appreciate hearing your further feedback on our response.
>
> Looking forward to your reply!
>
> Best Regards,
> All Authors

---

> > ### Comment · Reviewer_Yg3R · 2024-08-27
> >
> > Thank you for your response in details.
> > It is interesting to discuss these points in the paper.
> > Based on your results and answers, I increased my overall score.

---

> > > ### Author Response · Authors · 2024-08-27
> > > **Sincere Thanks for Your Valuable Feedback**
> > >
> > > Dear Reviewer Yg3R,
> > >
> > > Thank you very much for your detailed review and for raising your overall score. We sincerely appreciate your thoughtful feedback, which has been incredibly valuable in refining our work. We will be sure to incorporate these suggestions and the new insights into the revised version of the paper.
> > >
> > > Best regards,
> > >
> > > The Authors

---

### Official Review · Reviewer_LXN2 · 2024-07-25
**review of ProbTS**

**Rating:** 5
**Confidence:** 3
**Correctness:** The claims made in the submission see…

**Review:**

## Pros
* First comparative study of short/long-term forecasting capabilities of AR and NAR-based foundational models for time series;
* Exhaustive selection of neural network-based models for the benchmark;
* Addresses the gap in the related work focusing solely on short-term probabilistic forecasting and long-term point forecast methods;
* Useful metrics describing the time series characteristics (like non-Gaussianity) are introduced in the work that can be used in the following studies;

## Cons
* Not a large selection of time series datasets; for example only five short-term datasets are included;
* Few benchmark datasets used in the study;
* Univariate / multi-variate time series performance differences are not clear from the study;
* IMHO, The figures presenting the main results are not the cleanest way of presenting the results and conveying the main message; there are some errors in the figures, like two series presented on a single figure etc..;

**Strengths:**

The introduced non-Gaussianity measure provides an interesting way of measuring the time series hardness for univariate series. Especially for long-term time series, there is a correlation of non-Gaussianity with minimal NMAE and CRPS metrics achieved by the tested forecasting techniques.
The first study of short/long-term forecasting capabilities of AR and NAR-based foundational models for time series demonstrates the stronger performance of AR models in the case of short-term series and NAR models excelling for long-term forecasting horizons.

**Additional Feedback:**

N/A

**Clarity:**

The paper is mostly well-written; additional study details are presented in the appendix. However, I have concerns regarding the clarity of the figures presenting the main results.

**Documentation:**

The paper includes a link to an open-source GitHub repository that seems to be adequately documented.

**Limitations:**

Yes, the limitations and potential negative societal impact was adequately discussed in the paper.

**Opportunities For Improvement:**

* For the considered metrics (e.g., NMAE, CRPS) mark clearly if smaller / larger values indicate better performance;
* Fig. 3(a) is hard to read why there are two sequences of markings; the first one is short, in the range 0-75, and the other one in the full range;
* Can a performance difference be observed comparing uni-variate and multi-variate time series?
* No information was given on the computational budget used to perform the benchmark;
* No clear information on the used benchmark datasets in terms of uni/multi-variateness and how the covariates  were used;
* Some plots comparing the computed point forecasts against the probabilistic forecasts would be useful;
* The AR / Non-AR comparison should be extended also  to other neural network approaches like LSTM;

**Relation To Prior Work:**

Yes, it is clearly explained and the paper's contributions are compared to the related work.

**Summary And Contributions:**

The ProbTS paper presents a comprehensive study of state-of-the-art point and distributional time series forecasting methods. It addresses the gap that earlier studies focused on either short-term distributional or long-term point forecasting methods. It dissects novel time series characteristics and studies the model's performance with respect to those found characteristics.

---

> ### Author Rebuttal · Authors · 2024-08-17
>
> Thank you for your comprehensive review. We greatly appreciate your constructive suggestions, which will help us improve. Below, we have prepared detailed, point-by-point responses to address your remaining concerns, and we look forward to further discussions with you.
>
> ## Datasets
>
> > -Not a large selection of time series datasets; for example only five short-term datasets are included
> -Few benchmark datasets used in the study;
> >
>
> We understand your concerns regarding the limited selection of benchmark datasets. Here, we provide additional explanations to address these concerns.
>
> First and foremost, our dataset selection is motivated by the need to investigate the capabilities of existing studies in fulfilling essential forecasting needs (different prediction horizons, point, and distributional forecasts). Compared to existing studies that usually focus on a subset of these aspects, our study offers comprehensive coverage of datasets used in different research threads, including:
>
> - Almost all datasets used to evaluate short-term probabilistic forecasting
>     - Compiled from studies like CSDI, TimeGrad, GRU NVP, etc.
> - Almost all datasets used to evaluate long-term point forecasting
>     - Compiled from studies such as iTransformer, PatchTST, etc.
> - Datasets used in the evaluation of universal time-series foundation models (Overlapped with the above two)
>     - Such as TimesFM, MOIRAI, Chronos, etc.
>
> We have included these datasets to ensure a consistent comparison with existing models and because they have been endorsed by many previous studies and prestigious conferences. Moreover, we have included experiments on additional datasets in the appendix, including:
>
> - Some univariate datasets
>     - Compiled from M4, M5, Tourism (See Appendix D.3, Table 15 and 16)
> - Some synthetic datasets
>     - To re-confirm our findings (See Appendix D.4, Table 17 and 18)
>
> Second, we would like to highlight that the included datasets cover a wide range of data characteristics that help analyze the pros and cons of different methodological choices.
>
> - As shown in Table 1, these datasets display distinct characteristics of trend, seasonality, and distribution complexity.
> - As summarized in Table 6 (Appendix B.1.1), these datasets also cover a wide range of variable numbers, sampling frequencies, timesteps, and domains.
>
> Therefore, we believe these datasets constitute a solid foundation for the comparison of existing models.
>
> Third, please note that the primary focus of this study is to investigate how well existing methodological designs address essential forecasting needs. Therefore, our major efforts have been devoted to the methodological aspects, including:
>
> - Re-implementing, unifying, and comparing cutting-edge models from different research threads.
> - Identifying typical methodological choices and analyzing their reasons.
> - Investigating the pros and cons of different choices in fulfilling essential forecasting needs.
> - Revealing underexplored or unnoticed areas for future research.
>
> Finally, we appreciate your suggestion to include larger and more datasets, as this will undoubtedly enhance a benchmarking study. Our tool supports simple extensions for incorporating new datasets, and we are open to adding any new datasets that you believe are necessary, especially those with specific data characteristics beyond the existing coverage of our study.

---

> ### Author Rebuttal · Authors · 2024-08-17
>
> ## Further discussion on uni- / multi-variate modeling
>
> > -Univariate / multi-variate time series performance differences are not clear from the study;
> -Can a performance difference be observed comparing uni-variate and multi-variate time series?
> >
>
> Thank you for the constructive suggestion to discuss univariate vs. multivariate modeling in greater detail.
>
> In the submitted manuscript, we included some discussions on this topic, but we did not highlight it in the main paper. Our experiments indicate that the choice between univariate and multivariate modeling is not a primary factor in fulfilling the essential forecasting needs considered in this paper (producing accurate point and distributional forecasts across diverse horizons).
>
> - In Appendix D.3, we include univariate datasets, such as M4, M5, and TOURISM, and confirm findings similar to those from multivariate datasets discussed in the main paper.
> - In Appendix D.7, we discuss how existing models handle multivariate datasets:
>     - Some models, like TimeGrad and CSDI, adopt multivariate modeling.
>     - Some models, like N-BEATS and N-HiTS, adopt univariate modeling.
>     - A few models, like PatchTST, can support either univariate or multivariate modeling by processing each variable channel independently or not.
>
> Your suggestion has inspired us to provide a more extensive discussion of univariate and multivariate time-series modeling. Therefore, we have prepared a systematic discussion below, along with additional experiments. We will also organize these discussions into a new section in the appendix to enhance the study.
>
> In general, we discuss the differences between univariate and multivariate modeling from two perspectives:
>
> - Dataset Perspective: Whether the dataset is prepared for univariate or multivariate benchmarking.
> - Model Perspective: How the model handles multivariate data, treating each variable channel independently or not.
>
> **Dataset Perspective**
>
> All datasets listed in Table 1 are typically referred to as multivariate datasets, indicating that there may be strong connections across different variables. Despite this implication, when developing forecasting models, we can treat each variable channel independently, essentially turning a multivariate dataset into a univariate setup. In contrast, some datasets, like M4, M5, and TOURISM listed in Table 15, explicitly serve univariate modeling. We rarely see multivariate models being developed for these univariate cases.
>
> **Model Perspective**
>
> We have observed different preferences for univariate and multivariate modeling. Existing models can be categorized into three groups:
>
> - **Native Univariate Models**
>     - Classical models like N-BEATS and N-HiTS.
>     - Most time-series foundation models, such as TimesFM and Chronos.
> - **Native Multivariate Models**
>     - Most probabilistic models, such as CSDI and TimeGrad.
>     - Some point forecasting models, such as Informer and Autoformer.
> - **Hybrid Models of Univariate and Multivariate Modeling**
>     - Some classical models, such as PatchTST.
>     - Some time-series foundation models, such as MOIRAI.
>
> Please note that native univariate models can also be applied to multivariate datasets by treating them as univariate cases. Similarly, native multivariate models can be applied to univariate datasets by setting the variable dimension to 1. Hybrid models typically include specific modes to activate univariate and multivariate functionalities. For example, in PatchTST, we can use a shared forecasting head for univariate modeling or assign a specific forecasting head for each variable channel to differentiate different variables.
>
> **Additional Experiments**
>
> Below, we include additional experiments comparing univariate and multivariate modeling of PatchTST across different datasets. Our observation is that there is no definitive answer as to which approach is superior; it depends on the nature of the dataset. Some datasets benefit from modeling variable correlations, while others perform better with independent modeling. The performance gaps are not significant.
>
> Similar observations have been reported in MOIRAI, which allows either univariate or multivariate modes by controlling its cross-variate attention masks. When applied to a downstream forecasting scenario, it can search over the validation set to determine which configurations to activate. We believe such a design could serve as a good example of unifying univariate and multvariate.
>
> Table. The comparison of PatchTST on univariate and multivariate modeling.
> | Dataset | Pred. Horizon | PatchTST (Multivariate) | PatchTST (Univariate) |
> | --- | --- | --- | --- |
> | ETTh1 | 96 | 0.3239 | 0.3212 |
> | ETTh1 | 192 |0.3609| **0.3562** |
> | ETTh1 | 336 |0.3763| **0.3737** |
> | ETTh1 | 720 | **0.3882** |0.3909|
> | ETTm1 | 96 | **0.2652** |0.2739|
> | ETTm1 | 192 | **0.2926** |0.2961|
> | ETTm1 | 336 | **0.3101** |0.3188|
> | ETTm1 | 720 | **0.345** |0.3463|
> | Electricity | 96 | **0.0832** |0.0857|
> | Electricity | 192 | **0.0899** |0.0912|
> | Electricity | 336 | **0.0995** |0.1001|
> | Electricity | 720 |0.1183| **0.116** |
> | Exchange | 96 |0.0243| **0.0235** |
> | Exchange | 192 |0.0348| **0.0336** |
> | Exchange | 336 |0.0471| **0.0462** |
> | Exchange | 720 |0.0787| **0.0777** |
> | Weather | 96 |0.0872| **0.0837** |
> | Weather | 192 |0.0924| **0.0858** |
> | Weather | 336 |0.0934| **0.0903** |
> | Weather | 720 |0.0993| **0.0953** |
>
> > -No clear information on the used benchmark datasets in terms of uni/multi-variateness and how the covariates were used;
> >
> As illustrated above, we will supplement discussions about uni/multi variate in the revised manuscript. Besides, regarding covariate usage, we followed each model’s recommended practices in our benchmarking study. For example, TimeGrad mixes covariate information with observation values, while PatchTST does not incorporate covariates. Thank you for your suggestion and we will add a section in the appendix to further explain covariate usage.

---

> ### Author Rebuttal · Authors · 2024-08-17
>
> ## Presentation of figures
>
> > -IMHO, The figures presenting the main results are not the cleanest way of presenting the results and conveying the main message; there are some errors in the figures, like two series presented on a single figure etc..;
> -Fig. 3(a) is hard to read why there are two sequences of markings; the first one is short, in the range 0-75, and the other one in the full range;
> -For the considered metrics (e.g., NMAE, CRPS) mark clearly if smaller / larger values indicate better performance;
> >
>
> We apologize for any confusion caused by the figures. Based on your feedback, we have adjusted the x-axis distribution to improve clarity and have clearly indicated whether smaller or larger values represent better performance for each metric.
>
> Regarding the inclusion of two series in a single figure, our goal was to illustrate the impact of AR and NAR designs across both short and long horizons. For example, Figure 3a was intended to demonstrate how AR models tend to accumulate errors as the forecast horizon extends.
>
> We appreciate your suggestions, which have helped us to present our findings more clearly. We will polished these figures, which have been included in the pdf file attached.
>
> ## Computational budget
>
> > -No information was given on the computational budget used to perform the benchmark;
> >
>
> Thank you for raising the concern about the computational budget. We did not provide computational budget estimates for all baselines, as most of them are relatively efficient.
>
> For the main baselines in this paper, we have included a detailed efficiency analysis in Appendix D6, covering memory usage and time efficiency on long-term forecasting datasets. Specifically, Figure 9a and Table 19 present the computational memory requirements for various models with a forecasting horizon of 96, while Figure 9b and Table 20 compare the inference times relative to the forecasting horizon. Additionally, the benchmarking environment is described in Appendix B3. We apologize for not including this information in the main text and will make sure to highlight it in the revised version.
>
> To provide a clearer perspective, we have summarized the Inference time, memory usage, and model parameter sizes for the main baselines used in our study in the table below. This information will be added to the appendix as a supplement.
>
>  If you have further suggestions regarding the computational budget, we would be happy to adopt them.
>
> Table 19: Summary of model efficiency. The batch size is 1 and the prediction horizon is set to 96.
>
> | Metric | Dataset | DLinear | PatchTST | GRU NVP | TimeGrad | CSDI |
> | --- | --- | --- | --- | --- | --- | --- |
> | NPARAMS (MB) | ETTm1 | 0.075 | 2.145 | 1.079 | 1.233 | 1.720 |
> |  | Electricity-L | 0.076 | 2.146 | 3.680 | 3.472 | 1.370 |
> |  | Traffic-L | 0.078 | 2.149 | 15.926 | 8.298 | 1.390 |
> |  | Weather-L | 0.075 | 2.145 | 3.085 | 0.574 | 1.721 |
> |  | Exchange-L | 0.075 | 0.135 | 1.979 | 0.488 | 1.720 |
> | Max GPU Mem. (GB) | ETTm1 | 0.002 | 0.009 | 0.010 | 0.012 | 0.027 |
> |  | Electricity-L | 0.060 | 0.068 | 0.129 | 0.128 | 1.411 |
> |  | Traffic-L | 0.161 | 0.168 | 0.361 | 0.333 | 9.102 |
> |  | Weather-L | 0.004 | 0.012 | 0.021 | 0.012 | 0.070 |
> |  | Exchange-L | 0.002 | 0.002 | 0.013 | 0.008 | 0.030 |
> | Avg. Infer Time (sec./batch) | ETTm1 | 0.0003 | 0.0003 | 0.0352 | 4.1067 | 16.3280 |
> |  | Electricity-L | 0.0004 | 0.0045 | 0.1783 | 13.8439 | 388.3150 |
> |  | Traffic-L | 0.0010 | 0.0102  | 0.3695 | 31.7644 | - |
> |  | Weather-L | 0.0002 | 0.0004 | 0.0800 | 4.1261 | 37.8984 |
> |  | Exchange-L | 0.0006 | 0.0004 | 0.0284 | 4.1069 | 17.8655 |
> | Avg. Train Time (sec./batch) | ETTm1 | 0.0166 | 0.025 | 0.0219 | 0.0365 | 0.0502 |
> |  | Electricity-L | 0.0242 | 0.0291 | 0.0327 | 0.0399 | 0.0467 |
> |  | Traffic-L | 0.0171 | 0.0296 | 0.0244 | 0.0387 | - |
> |  | Weather-L | 0.01 | 0.0259 | 0.0358 | 0.0367 | 0.0568 |
> |  | Exchange-L | 0.0094 | 0.0259 | 0.0354 | 0.0427 | 0.0506 |
>
> ## Further enhancement
>
> > -Some plots comparing the computed point forecasts against the probabilistic forecasts would be useful;
> >
>
> We totally agree that includes more comparison and analysis will provides more intuitions.
>
> In this paper, we have visualized quantitative model performance in Figure 2, where we compare point and probabilistic forecasts along the dimensions of forecasting horizons and non-Gaussianity of the dataset. In addition, we conducted a case study to intuitively demonstrate the distinct characteristics of point and probabilistic estimations in Appendix D.5.
>
> If there is any other information you would like to learn more about, please let us know, we would be happy to conduct corresponding visualization analysis of that as well.
>
> > -The AR / Non-AR comparison should be extended also to other neural network approaches like LSTM;
> >
>
> Thank you for your suggestion. Per your recommendation, we have included LSTM as a baseline in the AR/Non-AR comparison. The updated experimental results have been included in our updated figures in the attached pdf. We will incorporate these new results into the revised version of the paper to ensure a more comprehensive analysis.
>
> We hope that the above responses have addressed your confusion and concerns. We appreciate your suggestions and will improve all the limitations you mentioned in the revised version.

---

> ### Author Rebuttal · Authors · 2024-08-17
>
> We apologize for the reversed order display of our responses. We have submitted our responses sequentially, but the system displays the last submitted part at the top.
>
> **All References**
>
> [1] Zeng, A., et al. (2023). Are transformers effective for time series forecasting?. *AAAI*.
>
> [2] Nie, Y., et al. (2023). A time series is worth 64 words: Long-term forecasting with transformers. *ICLR.*
>
> [3] Liu, Y., et al. (2024). iTransformer: Inverted transformers are effective for time series forecasting. *ICLR.*
>
> [4] Rasul, K., et al. (2021). Autoregressive denoising diffusion models for multivariate probabilistic time series forecasting. *ICML*.
>
> [5] Tashiro, Y., et al. (2021). CSDI: Conditional score-based diffusion models for probabilistic time series imputation. *NeurIPS*.
>
> [6] Gerald W., et al. (2024). Unified Training of Universal Time Series Forecasting Transformers. *ICML*.
>
> [7] Abhimanyu D., et al. (2024). A decoder-only foundation model for time-series forecasting. *ICML*.
> [8] Abdul Fatir Ansari, et al. (2024). Chronos: Learning the Language of Time Series. *arXiv preprint arXiv:2403.07815*.

---

> ### Author Response · Authors · 2024-08-27
> **A Kind Reminder for the End of Discussion Phase is Approaching**
>
> Dear Reviewer LXN2,
>
> Thank you for your insightful questions and constructive suggestions. As the discussion phase nears its end, we kindly request that you review our responses to your comments.
>
> In response to your suggestions, we have provided additional discussions and analysis on univariate/multivariate forecasting approaches, benchmarking dataset selection, computational budgets, and have also improved the overall presentation of the paper.
>
> We hope these responses have addressed your concerns. We appreciate your consideration and look forward to your feedback. Please feel free to reach out if you have any further questions.
>
> Best Regards,
>
> All Authors

---

> ### Author Response · Authors · 2024-08-31
> **A Kind Reminder for Only One Day Left in the Discussion Phase**
>
> Dear Reviewer LXN2,
>
> With only one day remaining in the discussion phase, we would like to confirm whether our responses have addressed your concerns. If you need further clarification or have additional questions, please feel free to let us know, we would be happy to address them.
>
> Looking forward to your reply!
>
> Best regards,
> All Authors

---

> > ### Comment · Reviewer_LXN2 · 2024-08-31
> >
> > Tanks for the detailed answers and performing additional experiments, I view the paper more favorably now.

---

> > > ### Author Response · Authors · 2024-09-01
> > >
> > > Dear Reviewer LXN2,
> > >
> > > We’re pleased to hear that your concerns have been fully addressed. We sincerely appreciate your thorough review and constructive suggestions, which have significantly improved our paper.
> > >
> > > If our revisions and additional experiments meet your expectations, we would be grateful if you could consider reflecting this in your score.
> > >
> > > Thank you again for your time and support.
> > >
> > > Best regards,
> > >
> > > All Authors

---

### Decision · Program_Chairs · 2024-09-26

**Decision:**

Accept (Poster)

**Comment:**

The reviewers see several strengths in the paper:
- s1. systematic combination of short-/long-term and point/distribution forecast scenarios.
- s2. thorough selection of forecasting models, including foundation models.
- s3. interesting aspects in the analysis, e.g., non-gaussianity.

But they also discussed some weaknesses:
- w1. benchmarks covers only a limited number of datasets.
- w2. performance differences between univariate and multivariate settings
  not clear.
- w3. long- vs short-term forecasting not clearly defined.

In their rebuttal the authors answered w1 "limited datasets" by stating
that they use a superset of what is in use by state-of-the-art model papers
currently, as well as w3 "missing definition of short- vs long-term
forecasting" by using strong seasonalities of the dataset as threshold.
For w2 "missing differences between univariate and multivariate settings"
they provide many details and additional experiments, but it really is
a major aspect of time series forecasting problems, esp. when including
foundational models, that comes only as an afterthought to this paper.

Based on the mostly positive ratings of the reviewers
I overall recommend to accept the paper.